# Weisfeiler and Leman Go Loopy: A New Hierarchy for Graph Representational Learning

**Raffaele Paolino**[*,1,2]    **Sohir Maskey**[*,1]    **Pascal Welke**[3]    **Gitta Kutyniok**[1,2,4,5]

[1]Department of Mathematics, LMU Munich
[2]Munich Center for Machine Learning (MCML)
[3]Faculty of Computer Science, TU Wien
[4]Institute for Robotics and Mechatronics, DLR-German Aerospace Center
[5]Department of Physics and Technology, University of Tromsø

## Abstract

We introduce $r$-loopy Weisfeiler-Leman ($r$-$\ell$WL), a novel hierarchy of graph isomorphism tests and a corresponding GNN framework, $r$-$\ell$MPNN, that can count cycles up to length $r+2$. Most notably, we show that $r$-$\ell$WL can count homomorphisms of cactus graphs. This extends 1-WL, which can only count homomorphisms of trees and, in fact, we prove that $r$-$\ell$WL is incomparable to $k$-WL for any fixed $k$. We empirically validate the expressive and counting power of $r$-$\ell$MPNN on several synthetic datasets and demonstrate the scalability and strong performance on various real-world datasets, particularly on sparse graphs. Our code is available on GitHub.

## 1   Introduction

Graph Neural Networks (GNNs) (Scarselli et al., 2009; Bronstein et al., 2017) have become a prevalent architecture for processing graph-structured data, contributing significantly to various applied sciences, such as drug discovery (Stokes et al., 2020), recommender systems (Fan et al., 2019), and fake news detection (Monti et al., 2019).

Among various architectures, Message Passing Neural Networks (MPNNs) (Gilmer et al., 2017) are widely used in practice, as they encompass only local computation, leading to fast and scalable models. Despite their success, the representational power of MPNNs is bounded by the Weisfeiler-Leman (WL) test, a classical algorithm for graph isomorphism testing (Xu et al., 2019; Morris et al., 2019). This limitation hinders MPNNs from recognizing basic substructures like cycles (Chen et al., 2020). However, specific substructures can be crucial in many applications. For example, in organic chemistry, the presence of cycles can impact various chemical properties of the underlying molecules (Deshpande et al., 2002; Koyutürk et al., 2004). Therefore, it is crucial to investigate whether GNNs can count certain substructures and to design architectures that surpass the limited power of MPNNs.

Several models have been proposed to surpass the limitations of WL. Many of these models draw inspiration from higher-order WL variants (Morris et al., 2019), enabling them to count a broader range of substructures. For instance, GNNs emulating 3-WL can count cycles up to length 7. However, this increased expressivity comes at a high computational cost, as 3-WL does not respect the sparsity of real-world graphs, posing serious scalability issues. Hence, there is a critical need to design expressive GNNs that respect the inherent sparsity of real-world graphs (Morris et al., 2023).

---

*Equal contribution.
Corresponding authors: paolino@math.lmu.de, maskey@math.lmu.de.

38th Conference on Neural Information Processing Systems (NeurIPS 2024).

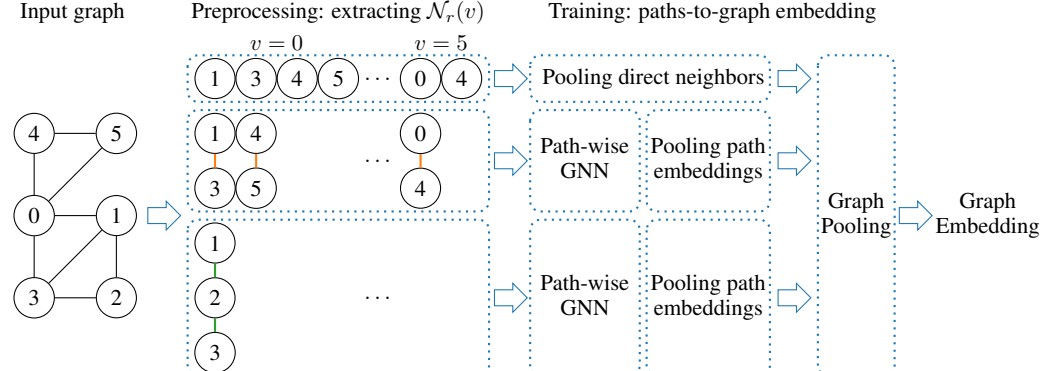

Figure 1: Visual depiction of $r$-$\ell$GIN: During preprocessing, we calculate the path neighborhoods $\mathcal{N}_r(v)$ for each node $v$ in the graph $G$. Paths of varying lengths are processed separately using simple GINs, and their embeddings are pooled to obtain the final graph embedding. The forward complexity scales linearly with the sizes of $\mathcal{N}_r(v)$, enabling efficient computation on sparse graphs.

**Main Contributions.** We introduce a novel class of color refinement algorithms called *r-loopy Weisfeiler-Leman test (r-ℓWL)* and a corresponding class of GNNs named *r-loopy Graph Isomorphism Networks (r-ℓGIN)*. The key idea is to collect messages not only from neighboring nodes but also from the paths connecting any two distinct neighboring nodes, as illustrated in Figure 1. This approach enhances the resulting GNNs' expressivity beyond 1-WL. In particular, $r$-$\ell$WL can count cycles up to length $r+2$, even surpassing the $k$-WL hierarchy.

Furthermore, we prove that $r$-$\ell$WL can homomorphism-count any cactus graph with cycles up to length $r+2$. Cactus graphs are valuable due to their structural properties and simplicity, making them useful for modeling in areas such as electrical engineering (Nishi et al., 1986) and computational biology (Paten et al., 2011). For instance, aromatic compounds often form cactus graphs, where the molecular core, usually a cycle, is coonected to functional groups (e.g., carboxyl groups) that can significantly impact the properties of the molecule. Thus, the ability to homomorphism-count cactus graphs can enhance model performance, and it allows us to compare the expressive power of $r$-$\ell$WL with other popular GNNs in a quantitative manner (Barceló et al., 2021; B. Zhang et al., 2024). Specifically, we show that $r$-$\ell$WL is more expressive than GNNs that include explicit homomorphism counts of cycle graphs, known as $\mathcal{F}$-Hom-GNNs (Barceló et al., 2021). Additionally, 1-$\ell$WL can already separate infinitely many graphs that Subgraph $k$-GNNs (Frasca et al., 2022; Qian et al., 2022) cannot (see, e.g., Figure 7). The higher expressivity, paired with the local computations, highlights the enhanced potential of $r$-$\ell$GIN, showing its competitive performance and the efficiency of its forward pass on real-world datasets, see Section 7.

## 2 Related Work

The notion of expressivity in standard neural networks is linked to the ability to approximate any continuous function (Cybenko, 1989; Hornik et al., 1989). In contrast, GNN expressivity is measured by the ability to distinguish non-isomorphic graphs. According to the Stone-Weierstrass theorem, these criteria are equivalent (Chen et al., 2019; Dasoulas et al., 2021): a network that can distinguish all graphs can approximate any continuous function. Therefore, research often focuses on determining which graphs a GNN can distinguish (Morris et al., 2023).

Xu et al. (2019) and Morris et al. (2019) proved that the expressive power of MPNNs is bounded by 1-WL. Subsequent works (Maron et al., 2018; Morris et al., 2019, 2020) introduced higher-order GNNs that have the same expressive power as $k$-WL or its local variants (Geerts et al., 2022). Although these networks are universal (Maron et al., 2019b; Keriven et al., 2019), their exponential time and space complexity in $k$ renders them impractical. Abboud et al. (2022) proposed $k$-hop GNNs which aggregate information from $k$-hop neighbors, thus, enhancing expressivity beyond 1-WL but within 3-WL (Feng et al., 2022). Michel et al. (2023) and Graziani et al. (2024) construct GNNs that process paths emanating from each node to overcome 1-WL. Subgraph GNNs (Bevilacqua et al., 2021; You et al., 2021; Frasca et al., 2022; Huang et al., 2022) surpass 1-WL by decomposing the initial input

graph into a bag of subgraphs. However, subgraph GNNs are upper-bounded by 3-WL (Frasca et al., 2022). A different line of work leverages positional encoding through unique node identifiers (Vignac et al., 2020), random features (Abboud et al., 2021; Sato et al., 2021) or eigenvectors (Lim et al., 2022; Maskey et al., 2022) to augment the expressive power of MPNNs.

While the predominant approach for gauging the expressive power of GNNs is within the $k$-WL hierarchy, such a measure is inherently qualitative, as it cannot shed light on substructures a particular GNN can encode. Lovász (1967) showed that *homomorphism counts* is a *complete graph invariant*, meaning two graphs are isomorphic if and only if their homomorphism counts are identical. Building on this result, B. Zhang et al. (2024) advocate for homomorphism-count as a quantitative measure of expressivity, as GNN architectures can homomorphism-count particular families of motifs. Tinhofer (1986, 1991) established that $1$-WL is equivalent to counting homomorphisms from graphs with tree-width one, while Dell et al. (2018) proved the equivalence between $k$-WL and the ability to count homomorphisms from graphs with tree-width $k$. Nguyen et al. (2020), Barceló et al. (2021), Welke et al. (2023), and Jin et al. (2024) used homomorphism counts to develop expressive GNNs.

Manually augmenting node features with homomorphism counts can be disadvantageous as performance depends on the chosen substructures. This can be alleviated by designing domain-agnostic GNNs that can learn structural information suitable for the task at hand. For instance, higher-order GNNs can count a large class of substructures as homomorphisms (B. Zhang et al., 2024), but they suffer from scalability issues. We propose $r$-$\ell$WL and $r$-$\ell$GIN, which can count homomorphisms of cactus graphs without adding explicit substructure counts. Our method is scalable to large datasets, particularly when the graphs in these datasets are sparse.

## 3 Preliminaries

Let $\mathcal{G}$ be the set of all simple and undirected graphs, and let $G \in \mathcal{G}$. We denote the set of nodes by $V(G)$ and the set of edges by $E(G)$. The *direct neighborhood* of a node $v \in V(G)$ is defined as $\mathcal{N}(v) := \{u \in V(G) \mid \{v, u\} \in E(G)\}$.

**Definition 1.** *Let $F, G \in \mathcal{G}$. A* homomorphism *from $F$ to $G$ is a map $h : V(F) \to V(G)$ such that $\{u, v\} \in E(F)$ implies $\{h(u), h(v)\} \in E(G)$. A* subgraph isomorphism *is an injective homomorphism.*

Intuitively, a homomorphism from $F$ to $G$ is an edge-preserving map. A subgraph isomorphism ensures that $F$ actually occurs as a subgraph of $G$. Consequently, it also maps distinct edges to distinct edges. A visual explanation can be found in Figure 5. We denote by $\mathrm{Hom}(F, G)$ the set of homomorphisms from $F$ to $G$ and by $\mathrm{hom}(F, G)$ its cardinality. Similarly, we denote by $\mathrm{Sub}(F, G)$ the set of subgraph isomorphisms from $F$ to $G$ and by and $\mathrm{sub}(F, G)$ its cardinality.

### 3.1 Graph Invariants

In order to unify different expressivity measures, we recall the definition of graph invariants.

**Definition 2.** *Let $P$ be a designated set, referred to as the* palette. *A graph invariant is a function $\zeta : \mathcal{G} \to P$ such that $\zeta(G) = \zeta(H)$ for all isomorphic pairs $G, H \in \mathcal{G}$. $\zeta$ is a* complete graph invariant *if $\zeta(G) \neq \zeta(F)$ for all non-isomorphic pairs $G, F \in \mathcal{G}$.*

Complete graph invariants have maximal expressive power. However, no polynomial-time algorithm to compute a complete graph invariant is known. To compare the expressive power of different graph invariants, such as graph colorings and GNN architectures, we introduce the following definition.

**Definition 3.** *Let $\gamma, \zeta$ be two graph invariants. We say that $\gamma$ is* more powerful *than $\zeta$ ($\gamma \sqsubseteq \zeta$) if for every pair $G, H \in \mathcal{G}$, $\gamma(G) = \gamma(H)$ implies $\zeta(G) = \zeta(H)$. We say that $\gamma$ is* strictly more powerful *than $\zeta$ if $\gamma \sqsubseteq \zeta$ and there exists a pair $F, G \in \mathcal{G}$ such that $\gamma(G) \neq \gamma(H)$ and $\zeta(G) = \zeta(H)$.*

### 3.2 Message Passing Neural Networks and Weisfeiler-Leman

Message passing is an iterative algorithm that updates the *colors* of each node $v \in V(G)$ as

$$c^{(t+1)}(v) \leftarrow f^{(t+1)} \left( c^{(t)}(v), g^{(t+1)} \left( \left\{\!\!\left\{ c^{(t)}(u) \mid u \in \mathcal{N}(v) \right\}\!\!\right\} \right) \right). \tag{1}$$

The graph output after $t$ iterations is given by

$$c^{(t)}(G) \coloneqq h\left(\left\{\left\{c^{(t)}(v) \mid v \in V(G)\right\}\right\}\right).$$

Here, $g^{(t)}, h$ are functions on the domain of multisets and $f^{(t)}$ is a function on the domain of tuples. For each $t$, the colorings $c^{(t)}$ are graph invariants. When the subsets of nodes with the same colors cannot be further split into different color groups, the algorithm terminates; the stable coloring after convergence is denoted by $c(G)$.

Choosing injective functions for all $f^{(t)}$ and setting $g^{(t)}$ and $h$ as the identity function results in 1-WL (Weisfeiler et al., 1968). If $f^{(t)}, g^{(t)}, h$ are chosen as suitable neural networks, one obtains a Message Passing Neural Network (MPNN). Xu et al. (2019) proved that MPNNs are as powerful as 1-WL if the functions $f^{(t)}, g^{(t)}$, and $h$ are injective on their respective domains. The $k$-WL algorithms uplift the expressive power of 1-WL by considering interactions between $k$-tuples of nodes. This results in a hierarchy of strictly more powerful graph invariants (see Appendix B.1 for a formal definition).

### 3.3 Homomorphism and Subgraph Counting Expressivity

A more nuanced graph invariant can be built by considering the occurrences of a motif $F$.

**Definition 4.** *Let $F \in \mathcal{G}$. A graph invariant $\zeta$ can* homomorphism-count *$F$ if for all pairs $G, H \in \mathcal{G}$ $\zeta(G) = \zeta(H)$ implies $\hom(F, G) = \hom(F, H)$. By analogy, $\zeta$ can* subgraph-count *$F$ if for all pairs $G, H \in \mathcal{G}$, $\zeta(G) = \zeta(H)$ implies $\mathrm{sub}(F, G) = \mathrm{sub}(F, H)$.*

If $\mathcal{F}$ is a family of graphs, we say that $\zeta$ can homomorphism-count $\mathcal{F}$ if $\zeta$ can homomorphism-count every $F \in \mathcal{F}$; we denote the vector of homomorphism-count by $\hom(\mathcal{F}, G) \coloneqq (\hom(F, G))_{F \in \mathcal{F}}$. Interpreting $\hom(\mathcal{F}, \cdot)$ as a graph invariant, given by $G \mapsto \hom(\mathcal{F}, G)$, another graph invariant $\zeta$ can homomorphism-count $\mathcal{F}$ if and only if $\zeta \sqsubseteq \hom(\mathcal{F}, \cdot)$.

The ability of a graph invariant to count homomorphisms is highly relevant because $\hom(\mathcal{G}, \cdot)$ is a complete graph invariant. Conversely, if $\zeta$ is a complete graph invariant, then $\zeta$ can homomorphism-count all graphs (Lovász, 1967). Additionally, homomorphism-counting serves as a quantitative expressivity measure to compare different WL variants and GNNs, such as $k$-WL, Subgraph GNNs, and other methods (Lanzinger et al., 2024; B. Zhang et al., 2024), and allows for relating them to our proposed $r$-$\ell$WL variant, as detailed in Corollary 2.

## 4 Loopy Weisfeiler-Leman Algorithm

In this section, we introduce a new graph invariant by enhancing the direct neighborhood of nodes with *simple paths* between neighbors.

**Definition 5.** *Let $G \in \mathcal{G}$. A simple path of length $r$ is a collection $\mathbf{p} = \{p_i\}_{i=1}^{r+1}$ of $r+1$ nodes such that $\{p_i, p_{i+1}\} \in E(G)$ and $i \neq j \implies p_i \neq p_j$ for every $i, j \in \{1, \ldots, r\}$,.*

Simple paths are the building blocks of $r$-neighborhoods, which in turn are the backbone of our $r$-$\ell$WL algorithm. The following definition is inspired by (Cantwell et al., 2019; Kirkley et al., 2021).

**Definition 6.** *Let $G \in \mathcal{G}$ and $r \in \mathbb{N} \setminus \{0\}$, we define the $r$-neighborhood $\mathcal{N}_r(v)$ of $v \in V(G)$ as*

$$\mathcal{N}_r(v) \coloneqq \{\mathbf{p} \mid \mathbf{p} \text{ simple path of length } r, p_1, p_{r+1} \in \mathcal{N}(v), v \notin \mathbf{p}\}.$$

For consistency, we set $\mathcal{N}_0(v) \coloneqq \mathcal{N}(v)$. An example of the construction of $r$-neighborhood is shown in Figure 2, where different $r$-neighborhoods of node $v$ are represented with different colors.

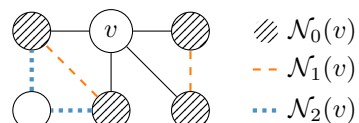

We generalize 1-WL in (1) as follows.

Figure 2: Example of $r$-neighborhoods.

**Definition 7.** *We define the $r$-loopy Weisfeiler-Leman ($r$-$\ell$WL) test by the following color update:*

$$c_r^{(t+1)}(v) \leftarrow \mathrm{HASH}_r\left(c_r^{(t)}(v), \left\{\left\{c_r^{(t)}(\mathbf{p}) \mid \mathbf{p} \in \mathcal{N}_0(v)\right\}\right\}, \ldots, \left\{\left\{c_r^{(t)}(\mathbf{p}) \mid \mathbf{p} \in \mathcal{N}_r(v)\right\}\right\}\right), \quad (2)$$

*where $c_r^{(t)}(\mathbf{p}) \coloneqq \left(c_r^{(t)}(p_1), c_r^{(t)}(p_2), \ldots, c_r^{(t)}(p_{r+1})\right)$ is the sequence of colors of nodes in the path.*

We denote by $c_r^{(t)}(G)$ the final graph output after $t$ iterations of $r$-$\ell$WL, i.e.,

$$c_r^{(t)}(G) = \text{HASH}_r\left(\left\{\left\{c_r^{(t)}(v) \mid v \in V(G)\right\}\right\}\right),$$

and by $c_r(G)$ the stable coloring after convergence. The stable coloring $c_r$ serves as graph invariant and will be referred to as $r$-$\ell$WL.

# 5  Expressivity of $r$-$\ell$WL

We analyze the expressivity of $r$-$\ell$WL in terms of its ability to distinguish non-isomorphic graphs, subgraph-count, and homomorphism-count motifs. The proofs for all statements are in Appendix D.

## 5.1  Isomorphism Expressivity

It is straightforward to check that 0-$\ell$WL corresponds to 1-WL, since $\mathcal{N}_0(v) = \mathcal{N}(v)$ for all nodes $v$. However, increasing $r$ leads to a strict increase in expressivity.

**Proposition 1.** *Let $0 \leq q < r$. Then, $r$-$\ell$WL is strictly more powerful than $q$-$\ell$WL. In particular, every $r$-$\ell$WL is strictly more powerful than 1-WL.*

This shows that the number of graphs we can distinguish monotonically increases with $r$. We empirically verify this fact on several synthetic datasets in Section 7.

## 5.2  Subgraph Expressivity

Recent studies highlight limitations in the ability of certain graph invariants to subgraph-count cycles. For instance, 1-WL cannot subgraph-count cycles (Chen et al., 2020, Theorem 3.3), while 3-WL can only subgraph-count cycles of length up to 7 (Arvind et al., 2020, Theorem 3.5). Similarly, Subgraph GNNs have limited cycle-counting ability (Huang et al., 2022, Proposition 3.1). In contrast, $r$-$\ell$WL can count cycles of arbitrary length, as shown in the following statement.

**Theorem 1.** *For any $r \geq 1$, $r$-$\ell$WL can subgraph-count all cycles with at most $r + 2$ nodes.*

Since 3-WL cannot subgraph-count any cycle with more than 7 nodes, Theorem 1 implies that 6-$\ell$WL is not less powerful than 3-WL. This observation generalizes to any $k$-WL, as shown next.

**Corollary 1.** *Let $k \in \mathbb{N}$. There exists $r \in \mathbb{N}$, such that $r$-$\ell$WL is not less powerful than $k$-WL. Specifically, $r \in \mathcal{O}(k^2)$, with $r \leq \frac{k(k+1)}{2} - 2$ for even $k$ and $r \leq \frac{(k+1)^2}{2} - 2$ for odd $k$.*

The $r$-$\ell$WL color refinement algorithm surpasses the limits of the $k$-WL hierarchy while only using local computation. This is particularly important since already 3-WL is computationally infeasible, whereas our method can scale efficiently to higher orders if the graphs are sparse, which is commonly the case in real-world applications.

## 5.3  Homomorphism Expressivity

The following section unveils a close connection between the expressivity of $r$-$\ell$WL and cactus graphs (Harary et al., 1953), a significant class between trees and graphs with tree-width 2.

**Definition 8.** *A cactus graph is a graph where every edge lies on at most one simple cycle. For $r \geq 2$, an $r$-cactus graph is a cactus where every simple cycle has at most $r$ vertices. We denote by $\mathcal{M}$ the set of all cactus graphs, and by $\mathcal{M}^r$ the set of all $q$-cactus graphs for $q \leq r$.*

Figure 6 shows two examples of cactus graphs. From the expressivity perspective, the ability to homomorphism-count cactus graphs establishes a lower bound strictly between the homomorphism-counting capabilities of 1-WL and 3-WL (Neuen, 2024), as cactus graphs are a strict superset of all trees and a strict subset of all graphs of treewidth two. With this in mind, we are now ready to present our significant result on the homomorphism expressivity of our $r$-$\ell$WL algorithm.

**Theorem 2.** *Let $r \geq 0$. Then, $r$-$\ell$WL can homomorphism-count $\mathcal{M}^{r+2}$.*

We refer to Appendix G for a detailed proof of Theorem 2, which is fairly involved and requires defining canonical tree decompositions of cactus graphs and unfolding trees of $r$-$\ell$WL. Demonstrating their strong connection, we then follow the approach in (Dell et al., 2018; B. Zhang et al., 2024) to decompose homomorphism counts of cactus graphs. In fact, we prove a more general result, showing that $r$-$\ell$WL can count all *fan-cactus graphs*, see Appendix G for more details.

The class $\mathcal{M}^2$ contains only forests; hence, Theorem 2 implies the standard results on the ability of 1-WL to count forests. Since forests are the only class of graphs 1-WL can count, Theorem 2 implies that $r$-$\ell$WL is always strictly more powerful than 1-WL, corroborating the claim in Proposition 1.

The implications of Theorem 2 are profound: it establishes that $r$-$\ell$WL can homomorphism-count a large class of graphs. Specifically, Theorem 2 provides a quantitative expressivity measure that enables comparison of $r$-$\ell$WL's expressivity with other WL variants and GNNs. This comparison is achieved by examining the range of graphs that $r$-$\ell$WL can homomorphism-count against those countable by other models, as detailed in works by Barceló et al. (2021) and B. Zhang et al. (2024). For instance, B. Zhang et al. (2024) showed that Subgraph GNNs (Bevilacqua et al., 2021; You et al., 2021; Frasca et al., 2022; Huang et al., 2022) are limited to homomorphism-count graphs with end-point shared NED. Hence, Subgraph GNNs can not homomorphism-count $F = \{\,$ ⬡⬤ $\,\}$, while 1-$\ell$WL can. Based on this, we can identify pairs of graphs that 1-$\ell$WL can distinguish but Subgraph GNNs cannot. We summarize these and other implications of Theorem 2 in the following corollary.

**Corollary 2.** *Let $r \in \mathbb{N} \setminus \{0\}$. Then,*

   i) *$r$-$\ell$WL is more powerful than $\mathcal{F}$-Hom-GNNs, where $\mathcal{F} = \{C_3, \ldots, C_{r+2}\}$.*

   ii) *1-$\ell$WL is not less powerful than Subgraph GNNs. In particular, any $r$-$\ell$WL can separate infinitely many graphs that Subgraph GNNs fail to distinguish.*

   iii) *For any $k > 0$, 1-$\ell$WL is not less powerful than Subgraph $k$-GNNs. In particular, any $r$-$\ell$WL can separate infinitely many graphs that Subgraph $k$-GNNs fail to distinguish.*

   iv) *$r$-$\ell$WL can subgraph-count all graphs $F$ such that $\mathrm{spasm}(F) \subset \mathcal{M}^{r+2}$, where $\mathrm{spasm}(F) \coloneqq \{H \in \mathcal{G} \mid \exists \text{ surjective } h \in \mathrm{Hom}(F, H)\}$. In particular, if $1 \le r \le 4$, then $r$-$\ell$WL can subgraph-count all paths up to length $r + 3$.*

A detailed explanation of Subgraph ($k$-)GNNs, $\mathcal{F}$-Hom-GNNs, along with the proofs of Corollary 2, can be found in Appendix H. Finally, we note that Theorem 2 states a loose lower bound on the homomorphism expressivity of $r$-$\ell$WL. This observation opens the avenue for future research to explore tight lower bounds, or upper bounds, on the homomorphism expressivity of $r$-$\ell$WL.

## 6 Loopy Message Passing

In this section, we build a GNN emulating $r$-$\ell$WL.

**Definition 9.** *For $t \in \{0, \ldots, T-1\}$ and $k \in \{0, \ldots, r\}$, $r$-$\ell$MPNN applies the following message, update and readout functions:*

$$
\begin{aligned}
m_k^{(t+1)}(v) &= f_k^{(t+1)}\left(\left\{\left\{c_k^{(t)}(\mathbf{p}) \mid \mathbf{p} \in \mathcal{N}_k(v)\right\}\right\}\right), \\
c_r^{(t+1)}(v) &= g^{(t+1)}\left(c_r^{(t)}(v),\, m_0^{(t+1)}(v), \ldots, m_r^{(t+1)}(v)\right),
\end{aligned}
\tag{3}
$$

*and final readout layer $c_r^{(T)}(G) = h\left(\left\{\left\{c_r^{(T)}(v) \mid v \in V(G)\right\}\right\}\right)$.*

In the following statement, we link the expressive power of $r$-$\ell$MPNN and $r$-$\ell$WL.

**Theorem 3.** *For fixed $t, r \ge 0$, $t$ iterations of $r$-$\ell$WL are more powerful than $r$-$\ell$MPNN with $t$ layers. Conversely, $r$-$\ell$MPNN is more powerful than $r$-$\ell$WL if the functions $f^{(t)}, g^{(t)}$ in (3) are injective.*

The previous result derives conditions under which $r$-$\ell$MPNN is as expressive as $r$-$\ell$WL. To implement $r$-$\ell$MPNN in practice, we choose suitable neural layers for $f_k^{(t)}, g^{(t)}$, and $h$ in Definition 9. As a consequence of (Xu et al., 2019, Lemma 5), the aggregation function in (3) can be written as

$$
f_k^{(t+1)}\left(\left\{\left\{c_k^{(t)}(\mathbf{p}) \mid \mathbf{p} \in \mathcal{N}_k(v)\right\}\right\}\right) := f\left(\sum_{\mathbf{p} \in \mathcal{N}_k(v)} g(\mathbf{p})\right),
$$

for suitable functions $f, g$. Since 1-WL is injective on forests (Arvind et al., 2015), hence on paths, and since GIN can approximate 1-WL (Xu et al., 2019), we choose $f = \text{MLP}$ and $g = \text{GIN}$. Hence, $r$-$\ell$GIN is defined as an $r$-$\ell$MPNN that updates node features via

$$x_r^{(t+1)}(v) := \text{MLP}\left(x_r^{(t)}(v) + (1 + \varepsilon_0) \sum_{u \in \mathcal{N}_0(v)} x_r^{(t)}(u) + \sum_{k=1}^{r}(1 + \varepsilon_k) \sum_{\mathbf{p} \in \mathcal{N}_k(v)} \text{GIN}_k(\mathbf{p})\right). \quad (4)$$

To reduce the number of learnable parameters in (4), the $\text{GIN}_k$ can be shared among all $k$. Nothing prevents from choosing a different path-processing layer; we opted for GIN because it is simple yet maximally expressive on paths. We refer to Figure 1 for a visual depiction of $r$-$\ell$GIN.

**Computational Complexity**    The complexity of $r$-$\ell$GIN is $\mathcal{O}(|E| + \sum_{v \in V(G)} \sum_{k=1}^{r} 2k|\mathcal{N}_k(v)|)$. The former addend is the standard message complexity, while the latter arises from applying GIN to paths of length $k \le r$. This implies that our model's complexity scales linearly with the number of edges, and with the number of paths within $\mathcal{N}_k(v)$. The number of such paths is typically less than the number of edges. For example, ZINC12K has overall 598K edges while only containing 374K paths in $\mathcal{N}_r(v)$ for $1 \le r \le 5$. Hence, the runtime overhead is small in practice. Compared to 3-WLGNN (Dwivedi et al., 2022a), which has the same cycle-counting expressivity, our model requires ca. 10 seconds/epoch while 3-WLGNN takes ca. $329.49$ seconds/epoch on ZINC12K. Our runtime is comparable to that of GAT, MoNet, or GatedGCN (see Table 10 for a thorough comparison).

**Comparison with (Michel et al., 2023)**    PathNN updates node features by computing all possible paths starting from each node. In contrast, our approach selects paths between distinct neighbors, potentially resulting in fewer paths. For instance, a tree's $r$-neighborhoods ($r \ge 1$) are empty, while counts of paths between nodes are quadratic. Notably, Michel et al. (2023) do not explore the impact of increasing the path length on architecture expressiveness, a consideration we address (see, e.g., Proposition 1 and Corollary 1). Another significant contribution of our work, which we assert does not hold (at least not trivially) for PathNN, is the provable ability to subgraph-count cycle graphs (see, e.g., Theorem 1) and homomorphism-count cactus graphs (see, e.g., Theorem 2).

# 7    Experiments

All instructions to reproduce the experiments are available on GitHub (MIT license). Additional information on the training and test details can be found in Appendix C.

**Expressive Power.**    We showcase the expressive power of $r$-$\ell$GIN on synthetic datasets:

- *GRAPH8C* (Balcilar et al., 2021) comprises $11\,117$ connected non-isomorphic simple graphs on 8 nodes; 312 pairs are 1-WL equivalent but none is 3-WL equivalent.

- *EXP_ISO* (Abboud et al., 2022) comprises 600 pairs of 1-WL equivalent graphs.

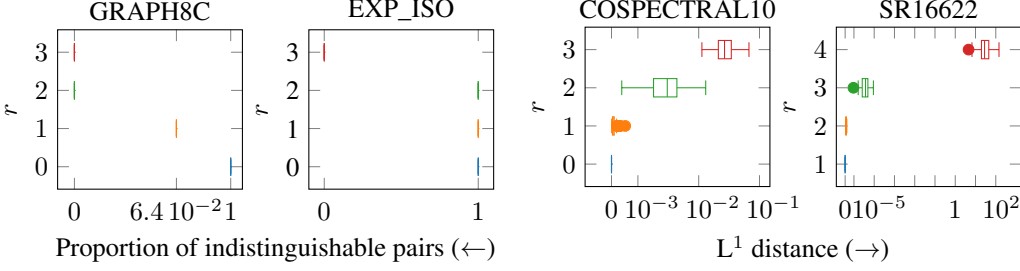

Figure 3: Indistinguishable pairs at initialization, symlog scale. For GRAPH8C and EXP_ISO, we report the proportion of indistinguished pairs: 2 graphs are deemed indistinguishable if the $\text{L}^1$ distance of their embeddings is less than $10^{-3}$. For COSPECTRAL10 and SR16622, we report the $\text{L}^1$ distance between graph embeddings. We report the mean and standard deviation over 100 seeds.

Table 1: Num. of distinguished pairs ($\uparrow$). Results from (Wang et al., 2024).

| Model | Basic (60) | Regular (140) | Extension (100) | CFI (100) |
|---|---|---|---|---|
| 3-WL | 60 | 50 | 100 | 60 |
| PPGN | 60 | 50 | 100 | 23 |
| NestedGNN | 59 | 48 | 59 | 0 |
| GSN | 60 | 99 | 95 | 0 |
| OSAN | 52 | 41 | 82 | 2 |
| 4-$\ell$GIN | 60 | 100 | 95 | 2 |

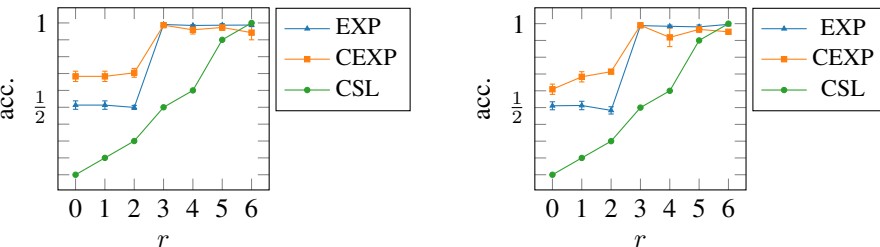

Figure 4: Test accuracy on synthetic classification task: (left) shared and (right) non-shared weights.

- *COSPECTRAL10* (van Dam et al., 2003): the dataset comprises two cospectral 4-regular non-isomorphic graphs on 10 nodes which are 1-WL equivalent (see, e.g., Figure 8a).
- *SR16622* (Michel et al., 2023) comprises two strongly regular graphs on 16 nodes, namely the Shrikhande and the $4 \times 4$ rook graph, which are 3-WL equivalent (see, e.g., Figure 8b).

The goal is to check whether the model can distinguish non-isomorphic pairs at initialization. The results are shown in Figure 3.

Additionally, Table 1 shows the performance on *BREC* (Wang et al., 2024), which includes 400 pairs of non-isomorphic graphs ranging from 1-WL to 4-WL equivalent. The baselines include PPGN, which is 3-WL equivalent and can count up to 7-cycles and homomorphism-count all graphs of tree-width 2; NestedGNN which is between 1-WL and 3-WL; GSN which is more powerful than 1-WL but whose expressive power depends on the chosen pattern.

Finally, Figure 4 reports the performance on synthetic classification tasks:

- *EXP, CEXP* (Abboud et al., 2021) require expressive power beyond 1-WL.
- *CSL* (Murphy et al., 2019) comprises 150 cycle graphs with skip links (see, e.g., Figure 8c). The task is to predict the length of the skip link.

**Counting Power.** Following (B. Zhang et al., 2024), we use the SUBGRAPHCOUNT dataset (Chen et al., 2020) to test the ability to homomorphism- and subgraphs-count exemplary motifs.

Table 2: Test MAE for homomorphism- and subgraph-counts. Results from (B. Zhang et al., 2024).

| Model | $\hom(F, G)$ | | | $\mathrm{sub}(F, G)$ | | | | | |
|---|---|---|---|---|---|---|---|---|---|
| MPNN | 0.300 | 0.233 | 0.254 | 0.358 | 0.208 | 0.188 | 0.146 | 0.261 | 0.205 |
| Subgraph GNN | 0.011 | 0.015 | 0.012 | 0.010 | 0.020 | 0.024 | 0.046 | 0.007 | 0.027 |
| Local 2-GNN | 0.008 | 0.008 | 0.010 | 0.008 | 0.011 | 0.017 | 0.034 | 0.007 | 0.016 |
| Local 2-FGNN | 0.003 | 0.005 | 0.004 | 0.003 | 0.004 | 0.010 | 0.020 | 0.003 | 0.010 |
| $r$-$\ell$GIN | 0.001 (r=2) | 0.006 (r=3) | 0.009 (r=3) | 0.0005 (r=1) | 0.0005 (r=2) | 0.0003 (r=3) | 0.0003 (r=4) | 0.001 (r=2) | 0.0004 (r=3) |

There is a strict hierarchy in the expressive power of the baselines: MPNN ⊑ Subgraph GNN ⊑ local 2-GNN ⊑ local 2-FGNN. These variants, apart from MPNNs, are more expressive than 1-WL and can subgraph-count up to 7-cycles.

**Real-World Datasets.** We experimented with three benchmark datasets: ZINC250K (Irwin et al., 2012), ZINC12K (Dwivedi et al., 2022a), and QM9 (Wu et al., 2018) which consist of 250 000, 12 000, and 130 831 molecular graphs, respectively. We report the mean and standard deviation over 4 random seeds.

For ZINC250K and ZINC12K, we selected as baseline models standard MPNNs (GIN, GCN, GAT), Subgraph GNNs (NestedGNN, GNNAK+, SUN), domain-agnostic GNNs fed with substructure counts (GSN, CIN), a GNN processing paths (PathNN), and expressive GNNs with provable cycle counting power (HIMP, SignNet, I2-GNN, DRFWL). Following the standard procedure, we kept the number of parameters under 500K (Dwivedi et al., 2022a) for ZINC12K. The results are detailed in Table 3.

For the QM9 dataset, we followed the setup of (Huang et al., 2022; Zhou et al., 2023). Specifically, the test MAE is multiplied by the standard deviation of the target and divided by the corresponding conversion unit. The baseline results and models were obtained from (Zhou et al., 2023), including expressive GNNs with provable cycle counting power. We omit methods that use additional geometric features to focus on the model's expressive power. The results are presented in Table 4.

Table 3: Test MAE ($\downarrow$) on ZINC dataset.

| Model | ZINC12K | ZINC250K |
|---|---|---|
| GIN | $0.163 \pm 0.004$ | $0.088 \pm 0.002$ |
| GCN | $0.321 \pm 0.009$ | - |
| GAT | $0.384 \pm 0.007$ | - |
| GSN | $0.115 \pm 0.012$ | - |
| CIN | $\underline{0.079 \pm 0.006}$ | $0.022 \pm 0.002$ |
| NestedGNN | $0.111 \pm 0.003$ | $0.029 \pm 0.001$ |
| SUN | $0.083 \pm 0.003$ | - |
| GNNAK+ | $0.080 \pm 0.001$ | - |
| I2-GNN | $0.083 \pm 0.001$ | $0.023 \pm 0.001$ |
| DRFWL | $0.077 \pm 0.002$ | $0.025 \pm 0.003$ |
| SignNet | $0.084 \pm 0.004$ | $\underline{0.024 \pm 0.003}$ |
| HIMP | $0.151 \pm 0.006$ | $0.036 \pm 0.002$ |
| PathNN | $0.090 \pm 0.004$ | - |
| 5-$\ell$GIN | $0.072 \pm 0.002$ | $0.022 \pm 0.001$ |

Table 4: Normalized test MAE ($\downarrow$) on QM9 dataset. Top three models as 1st, 2nd, 3rd.

| | | | | | Model | | | |
|---|---|---|---|---|---|---|---|---|
| Target | 1-GNN | 1-2-3-GNN | DTNN | Deep LRP | NestedGNN | I2-GNN | DRFWL | 5-$\ell$GIN |
| $\mu$ | 0.493 | 0.476 | 0.244 | 0.364 | 0.428 | 0.428 | 0.346 | $0.350 \pm 0.011$ |
| $\alpha$ | 0.78 | 0.27 | 0.95 | 0.298 | 0.290 | 0.230 | 0.222 | $0.217 \pm 0.025$ |
| $\varepsilon_{\text{homo}}$ | 0.00321 | 0.00337 | 0.00388 | 0.00254 | 0.00265 | 0.00261 | 0.00226 | $0.00205 \pm 0.00005$ |
| $\varepsilon_{\text{lumo}}$ | 0.00355 | 0.00351 | 0.00512 | 0.00277 | 0.00297 | 0.00267 | 0.00225 | $0.00216 \pm 0.00004$ |
| $\Delta(\varepsilon)$ | 0.0049 | 0.0048 | 0.0112 | 0.00353 | 0.0038 | 0.0038 | 0.00324 | $0.00321 \pm 0.00014$ |
| $R^2$ | 34.1 | 22.9 | 17.0 | 19.3 | 20.5 | 18.64 | 15.04 | $13.21 \pm 0.19$ |
| ZVPE | 0.00124 | 0.00019 | 0.00172 | 0.00055 | 0.0002 | 0.00014 | 0.00017 | $0.000127 \pm 0.000003$ |
| $U_0$ | 2.32 | 0.0427 | 2.43 | 0.413 | 0.295 | 0.211 | 0.156 | $0.0418 \pm 0.0520$ |
| $U$ | 2.08 | 0.111 | 2.43 | 0.413 | 0.361 | 0.206 | 0.153 | $0.023 \pm 0.023$ |
| $H$ | 2.23 | 0.0419 | 2.43 | 0.413 | 0.305 | 0.269 | 0.145 | $0.0352 \pm 0.0304$ |
| $G$ | 1.94 | 0.0469 | 2.43 | 0.413 | 0.489 | 0.261 | 0.156 | $0.0118 \pm 0.0015$ |
| $C_v$ | 0.27 | 0.0944 | 2.43 | 0.129 | 0.174 | 0.0730 | 0.0901 | $0.0702 \pm 0.0024$ |

**Discussion of Results** The results in Figures 3 and 4 and Table 1 constitute a strong empirical validation of our theory: increasing $r$ leads to more expressive $r$-$\ell$MPNN. Albeit 6-$\ell$WL is not less powerful than 3-WL (see, e.g., Section 5.2), in practice, smaller values of $r$ can already distinguish pair of graphs that are 3-WL equivalent, such as the Shrikhande and the ($4 \times 4$) rook graphs. In the BREC dataset, 4-$\ell$GIN distinguishes all pairs of strongly regular graphs, significantly outperforming 3-WL (0/50 graphs). Notably, 4-$\ell$GIN can already distinguish 257 out of 400 total pairs of graphs, surpassing other expressive GNNs like PPGN (233/400), theoretically equivalent to 3-WL, and NestedGNN (166/400). Refer to (Wang et al., 2024, Table 2) for detailed baseline results.

The results in Table 2 further substantiate our theory, as $r$-$\ell$WL can effectively count cycles of length $r+2$ (see, e.g., Theorem 1).

On molecular datasets, we observe that $r$-$\ell$GIN, although designed for subgraph-counting cycles and homomorphism-counting cactus graphs, is highly competitive. Notably, we outperform the baseline $0$-$\ell$GIN by 226% on ZINC12K and 400% on ZINC250K and surpass domain-agnostic methods such as CIN or GSN. We conjecture that this is attributed to straightforward optimization, driven by the simplicity of the architecture (see, e.g., Figure 1) and its inductive bias towards counting cycles.

**Limitations**   Path calculations can become infeasible for dense graphs due to $\mathcal{O}(N\,d^r)$ complexity, where $N$ is the number of nodes and $d$ is the average degree. However, for sparse graphs, the runtime remains reasonably low. For instance, preprocessing ZINC12K for $r = 5$ takes just over a minute.

## 8   Conclusion

In this paper,we introduce a novel hierarchy of color refinement algorithms, denoted as $r$-$\ell$WL, which incorporates an augmented neighborhood mechanism accounting for nearby paths. We establish connections between $r$-$\ell$WL and the classical $k$-WL. We construct a GNN ($r$-$\ell$MPNN) designed to emulate and match the expressive powerof $r$-$\ell$WL. Theoretical and empirical evidence support the claim that $r$-$\ell$MPNN can effectively subgraph-count cycles and homomorphism-count cactus graphs.

Future research could focus on precisely characterizing the expressivity of $r$-$\ell$WL tests by identifying the maximal class of graphs that $r$-$\ell$WL can homomorphism-count. This would facilitate comparisons by constructing pairs of graphs that $r$-$\ell$WL cannot separate, but other WL variants can. Another promising direction involves exploring the generalization capabilities of GNNs with provable homomorphism-counting properties. The ability to homomorphism-count certain motifs could provide a mathematical framework to support the intuitive notion that the capacity to count relevant features may improve generalization. We observed this improved generalization experimentally in our ablation study on ZINC12K (see, e.g., Table 8).

## Acknowledgements

R.P. is funded by the Munich Center for Machine Learning (MCML).

S.M. is funded by the NSF-Simons Research Collaboration on the Mathematical and Scientific Foundations of Deep Learning (MoDL) (NSF DMS 2031985) and DFG SPP 1798, KU 1446/27-2.

P.W. is funded by the Vienna Science and Technology Fund (WWTF) project StruDL (ICT22-059).

G.K. acknowledges partial support by the Konrad Zuse School of Excellence in Reliable AI (DAAD), the Munich Center for Machine Learning (BMBF) as well as the German Research Foundation under Grants DFG-SPP-2298, KU 1446/31-1 and KU 1446/32-1. Furthermore, G.K. acknowledges support from the Bavarian State Ministry for Science and the Arts as well as by the Hightech Agenda Bavaria.

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

# Appendix

## Table of Contents

# A    Additional Figures

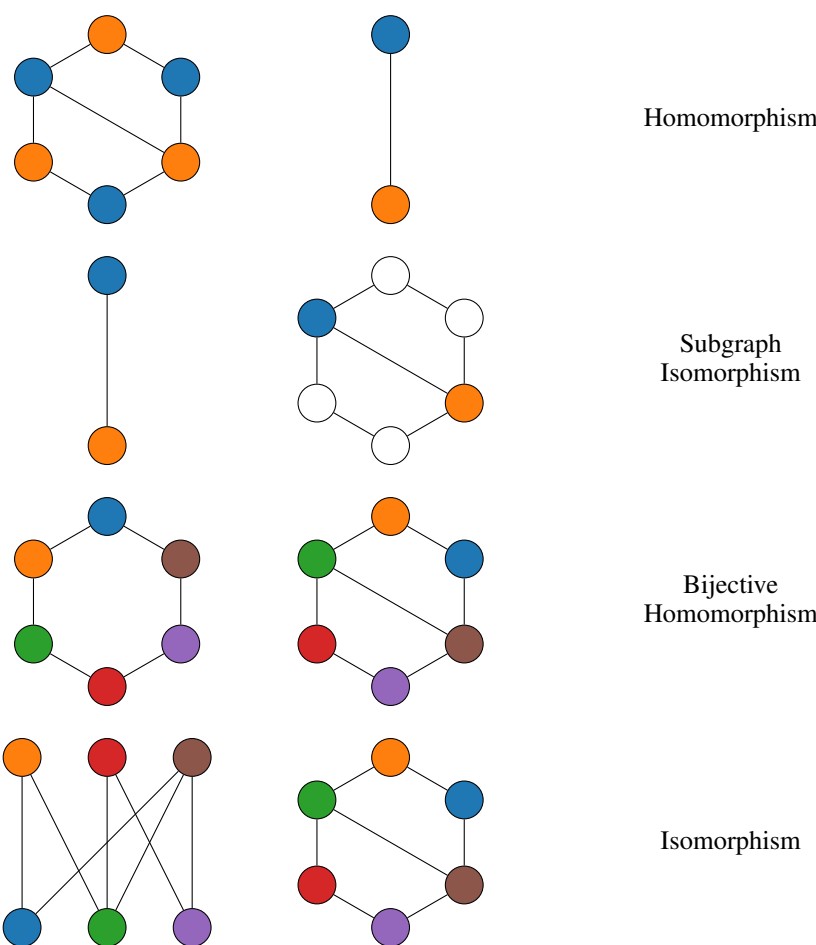

Homomorphism

Subgraph
Isomorphism

Bijective
Homomorphism

Isomorphism

Figure 5: Examples of non-injective homomorphism (row 1), subgraph isomorphism (row 2), bijective homomorphism with non-homomorphic inverse (row 3), and isomorphism (row 4). For better clarity, the mappings $h : V(F) \rightarrow V(G)$ are visually represented with colors, where $F$ is consistently on the left, and $G$ is on the right in each row.

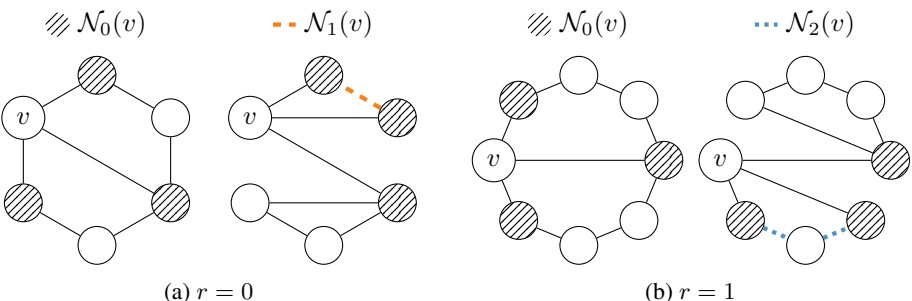

(a) $r = 0$

(b) $r = 1$

Figure 6: Example of two non-isomorphic graphs that are $r$-$\ell$WL equivalent but not $(r+1)$-$\ell$WL equivalent: a chordal cycle (left) and a cactus graph (right).

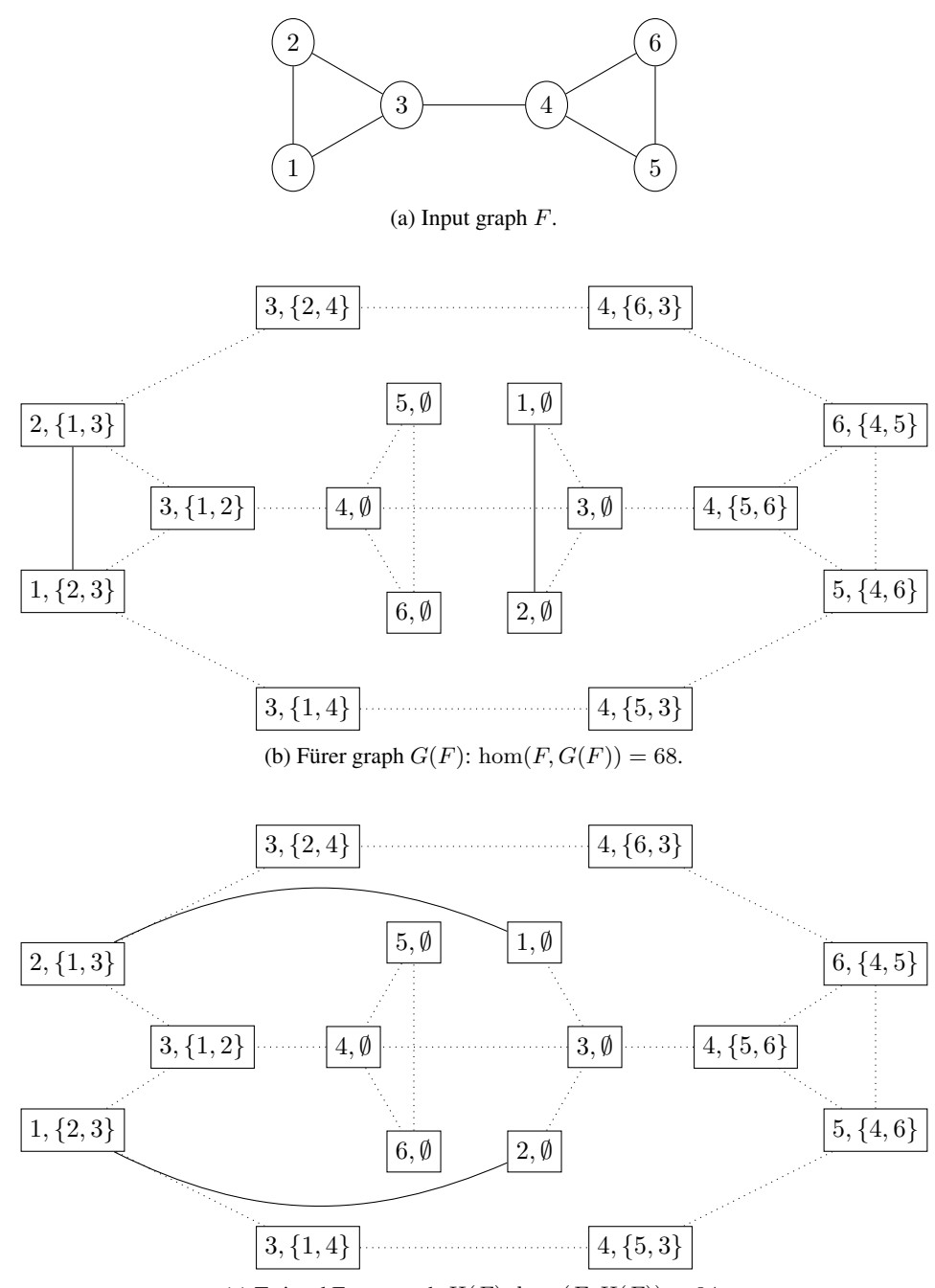

(a) Input graph $F$.

(b) Fürer graph $G(F)$: $\hom(F, G(F)) = 68$.

(c) Twisted Fürer graph $H(F)$: $\hom(F, H(F)) = 34$.

Figure 7: Example of graphs that Subgraph GNNs cannot separate but 1-$\ell$WL can: Subgraph GNNs cannot separate $G(F)$ and $H(F)$. However, since $\hom(F, G(F)) \neq \hom(F, H(F))$ and $F$ is a cactus graph, 1-$\ell$WL can separate $G(F)$ and $H(F)$ by Theorem 2.

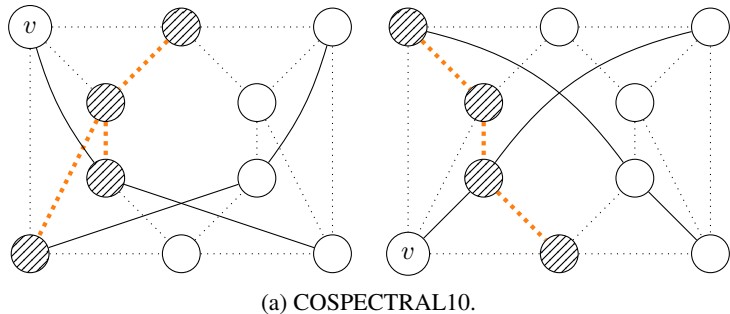

(a) COSPECTRAL10.

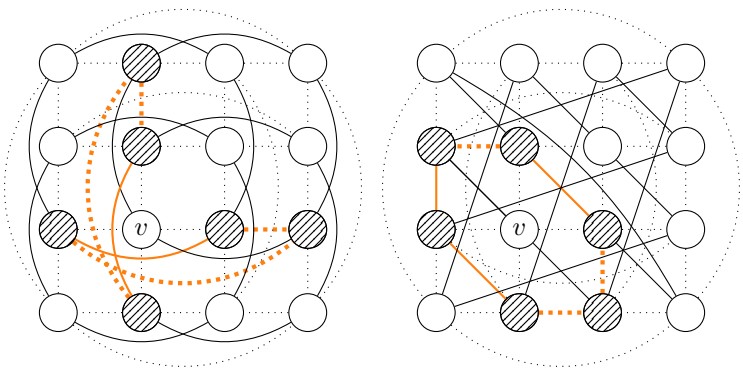

(b) SR16622.

(c) CSL example, skip length 2 (left) and 3 (right).

Figure 8: Some synthetic datasets. The dotted lines are the common edges. The orange edges identifies $\mathcal{N}_1(v)$.

# B  Additional Notions

## B.1  Higher-Order Weisfeiler-Leman Tests

It is possible to uplift the expressive power of WL by considering higher-order interactions. The simplest higher-order variant of WL is the $k$-dimensional Weisfeiler-Leman test, denoted by $k$-WL. Given a graph $G$ with nodes $V(G)$ and edges $E(G)$, the algorithm generates a new graph $H$ where each node is a $k$-tuple of elements of $V(G)$

$$V(H) = \left\{ \mathbf{v} = \{v_i\}_{i=1}^k \mid v_i \in V(G) \right\} = V(G)^k,$$

and edges $E(H)$ are built among those $k$-tuples that differ in one entry only

$$E(H) = \{\{\mathbf{v}, \mathbf{u}\} \mid d_H(\mathbf{v}, \mathbf{u}) = 1, \ \mathbf{u}, \mathbf{v} \in V(H)\}$$

where $d_H$ is the Hamming distance. The algorithm assigns to each node $\mathbf{v} \in V(H)$ an initial color depending on the isomorphic type of the induced subgraph $G[\mathbf{v}]$. The color refinements scheme is

exactly (1) applied to $H$. While $H$ can be generated by a simple algorithm, the approach quickly becomes impractical as the number of nodes and edges grows exponentially in $k$.

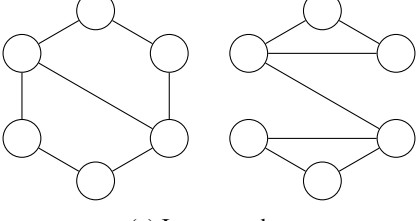

(a) Input graphs.

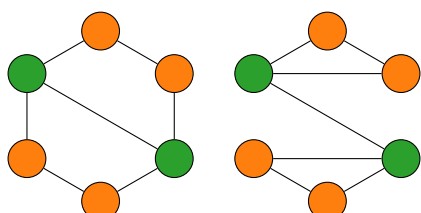

(b) 1-WL after one iteration.

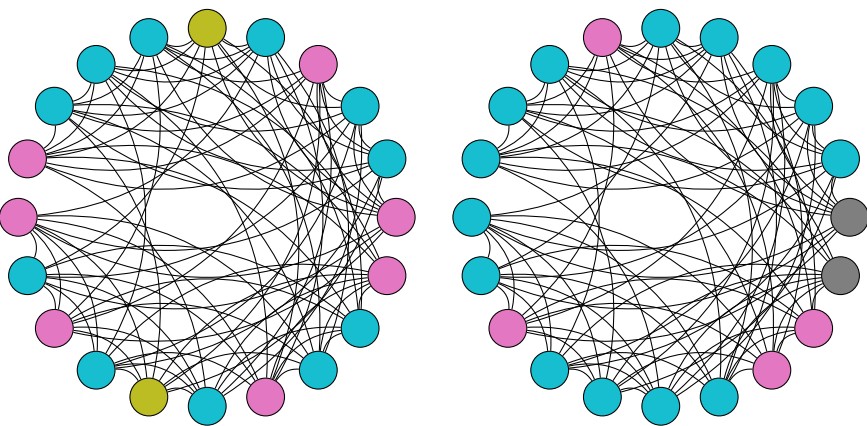

(c) 3-WL at initialization.

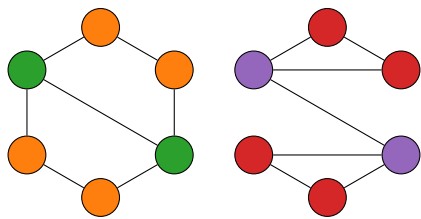

(d) 1-$\ell$WL after one iteration.

Figure 9: The input graphs cannot be distinguished by 1-WL, since the color distribution after convergence of the algorithm is equal. 3-WL can distinguish them at the cost of creating new dense graphs. Our proposed 1-$\ell$WL can distinguish the two graphs heeding the original graph sparsity.

## C   Experimental Details

Our model is implemented in `PyTorch` (BSD-3 license) (Paszke et al., 2019), using `PyTorch Geometric` (MIT license) (Fey et al., 2019). The $r$-neighborhoods are computed with `NetworkX` (Creative Commons Zero v1.0 Universal) (Hagberg et al., 2008) as preprocessing. Hyperparameters

on real-world datasets were tuned using grid search; for synthetic experiments, we fixed one configuration of hyperparameters. All experiments were run on an internal cluster with Intel Xeon CPUs (28 cores, 192GB RAM) and GeForce RTX 3090 Ti GPUs (4 units, 24GB memory each), as well as Intel Xeon CPUs (32 cores, 192GB RAM) and NVIDIA RTX A6000 GPUs (3 units, 48GB memory each). All models are trained with Adam optimizer (Kingma et al., 2015).

## C.1 Synthetic Datasets

The SR16622 dataset is retrieved from the official PATHNN repository (MIT license) (Michel et al., 2023). The GRAPH8C dataset is downloaded from Australian National University webpage (Creative Commons Attribution 4.0 International (CC BY 4.0) license). The EXP, EXP_ISO, and CEXP datasets are downloaded from GNN-RNI official repository (GPL-3.0 license) (Abboud et al., 2021), while the corresponding splits are generated via Stratified 5-fold cross-validation. The CSL dataset is provided by `torch_geometric`, while the corresponding splits are taken from PathNN official repository. The SUBGRAPHCOUNT dataset is taken from the official repository (MIT license) of (Zhao et al., 2022). The BREC dataset is downloaded from its official repository (MIT license) (Wang et al., 2024). The configuration of hyperparameters can be found in Table 5. For the synthetic datasets, we fixed one configuration and studied the effect of increasing $r$ on the expressive and counting power of the architecture.

For the SR16622, GRAPH8C, EXP_ISO, and COSPECTRAL10 datasets, we report the mean and standard deviation over 100 random seeds. For the EXP, CEXP, and CSL datasets, we report the mean and standard deviation of 5-fold cross-validation. For BREC, we follow the original setup and perform an $\alpha$-level Hotellings T-square test; see (Wang et al., 2024) for more details. For the SUBGRAPHCOUNT dataset, we report the mean and standard deviation over 4 random seeds, using the original splits from (Zhao et al., 2022).

Table 5: Hyperparameter configuration for synthetic experiments.

| | COSPECTRA10 | GRAPH8C | SR16622 | EXP_ISO | EXP | CEXP | CSL | SUBGRAPHCOUNT | BREC |
|---|---|---|---|---|---|---|---|---|---|
| Epochs | - | - | - | - | $10^3$ | $10^3$ | $10^3$ | $1.2\,10^3$ | 40 |
| Learning Rate | - | - | - | - | $10^{-3}$ | $10^{-3}$ | $10^{-3}$ | $10^{-3}$ | $10^{-4}$ |
| Early Stopping | - | - | - | - | lr $< 10^{-5}$ | lr $< 10^{-5}$ | lr $< 10^{-5}$ | - | lr $< 10^{-5}$ |
| Scheduler | - | - | - | - | $\{50, 0.5\}$ | $\{50, 0.5\}$ | $\{50, 0.5\}$ | $\{10, 0.9\}$ | $\{50, 0.5\}$ |
| Hidden Size | 64 | 64 | 64 | 64 | 64 | 64 | 64 | 64 | 32 |
| Num. Layers | 3 | 3 | 3 | 3 | 3 | 3 | 3 | 5 | 5 |
| Num. Encoder Layers | 2 | 2 | 2 | 2 | 2 | 2 | 2 | 2 | 2 |
| Num. Decoder Layers | 2 | 2 | 2 | 2 | 2 | 2 | 2 | 2 | 2 |
| Batch Size | 64 | 64 | 64 | 64 | 64 | 64 | 64 | 128 | 64 |
| Dropout | 0 | 0 | 0 | 0 | 0 | 0 | 0 | 0 | 0 |
| Readout | sum | sum | sum | sum | sum | sum | sum | sum | sum |

## C.2 Real-World Datasets

All real-world datasets are provided by `torch_geometric`. The splits for both ZINC datasets are also provided by `torch_geometric`. For QM9, we follow the set-up of (Zhou et al., 2023) and use random $80/10/10$ splits. Details for the datasets are provided in Table 6.

Hyperparameters were tuned using grid search. For ZINC12K, the grid was defined by Hidden Size $\in \{64, 128\}$ and Num. Layers $\in \{3, 4, 5\}$. For ZINC250K, the grid was defined by Hidden Size $\in \{128, 256\}$ and Num. Layers $\in \{4\}$. For the QM9 tasks, the grid was defined by Hidden Size $\in \{64, 128\}$ and Num. Layers $\in \{3, 4, 5\}$. For the QM9 tasks, we followed the training set-up of (Zhou et al., 2023), training for $400$ epochs with a `ReduceLROnPlateau` scheduler, reducing the learning rate by a factor of $0.9$ if the validation metric did not decrease for 10 epochs. The exact hyperparameters are given in Table 7.

All real-world datasets come with edge features. We use an encoder layer, followed by a linear layer to encode node, edge features, and atomic types before passing them to the $r$-$\ell$GIN. Within the $r$-$\ell$GIN layers, we process the edge features via a 2-layered learnable MLP, and replace the GIN in (4) by GINE layers (Hu* et al., 2020a). After $t$ rounds of $r$-$\ell$GIN layer, we apply a two-layered MLP as decoder layer. In all experiments, `BatchNorm1D` (Ioffe et al., 2015) is used in the MLP layers. We refer to Figure 1 for a depiction of the architecture.

Table 6: Statistics of real-world datasets.

| Dataset | Number of graphs | Average number of nodes | Average number of edges |
|---|---|---|---|
| QM9 | 130 831 | 18.0 | 18.7 |
| ZINC12K | 12 000 | 23.2 | 24.9 |
| ZINC250K | 249 456 | 23.2 | 24.9 |

Table 7: Hyperparameters configuration for real-world experiments.

| | ZINC12K | ZINC250K | QM9 ($\mu$) | QM9 ($\alpha$) | QM9 ($\varepsilon_{\mathrm{homo}}$) |
|---|---|---|---|---|---|
| Epochs | 1000 | 2000 | 400 | 400 | 400 |
| Learning Rate | 0.001 | 0.001 | 0.001 | 0.001 | 0.001 |
| Early Stopping | lr $< 10^{-5}$ | lr $< 10^{-6}$ | lr $< 10^{-5}$ | lr $< 10^{-5}$ | lr $< 10^{-5}$ |
| Scheduler | $\{50, 0.5\}$ | $\{50, 0.5\}$ | $\{10, 0.9\}$ | $\{10, 0.9\}$ | $\{10, 0.9\}$ |
| r | 5 | 5 | 5 | 5 | 5 |
| Hidden Size | 64 | 256 | 64 | 64 | 64 |
| Depth | 3 | 4 | 4 | 5 | 5 |
| Batch Size | 64 | 128 | 64 | 64 | 64 |
| Dropout | 0 | 0 | 0 | 0 | 0 |
| Readout | sum | sum | sum | sum | sum |
| # Parameters | 452 633 | 2 379 041 | 418 481 | 519 677 | 519 677 |
| Preprocessing Time [sec] | 77.4 | 1278.5 | 427.5 | 425.9 | 517.4 |
| Run Time per Seed [h] | 2.8 | 45.6 | 13.6 | 15.9 | 21.1 |

Table 8: Ablation study on the effect of $r$ in $r$-$\ell$GIN, ZINC12K.

| | MAE ($\downarrow$) | |
|---|---|---|
| Model | Train | Test |
| 0-$\ell$GIN | $0.060 \pm 0.009$ | $0.209 \pm 0.007$ |
| 1-$\ell$GIN | $0.060 \pm 0.012$ | $0.201 \pm 0.004$ |
| 2-$\ell$GIN | $0.068 \pm 0.011$ | $0.198 \pm 0.008$ |
| 3-$\ell$GIN | $0.056 \pm 0.013$ | $0.184 \pm 0.007$ |
| 4-$\ell$GIN | $0.0203 \pm 0.0002$ | $0.077 \pm 0.001$ |
| 5-$\ell$GIN | $0.022 \pm 0.004$ | $0.072 \pm 0.002$ |
| 6-$\ell$GIN | $0.028 \pm 0.000$ | $0.077 \pm 0.000$ |

Table 9: Test metrics on long-range graph benchmark datasets (Dwivedi et al., 2022b). The baseline results are obtained from (Dwivedi et al., 2022b, Table 4). Our method is able to enhance performance over standard baselines.

| Model | PEPTIDES | |
| | STRUCT (MAE ↓) | FUNC (AP ↑) |
|---|---|---|
| GCN | $0.3496 \pm 0.0013$ | $59.30 \pm 0.23$ |
| GINE | $0.3496 \pm 0.0013$ | $59.30 \pm 0.23$ |
| GatedGCN | $0.3420 \pm 0.0013$ | $58.64 \pm 0.77$ |
| 7-$\ell$GIN | $0.2513 \pm 0.0021$ | $65.70 \pm 0.60$ |

Table 10: Empirical time complexity for QM9 dataset; results from (Zhou et al., 2023). In parenthesis the size of the dataset after the computation of $r$-neighborhoods.

| Model | Memory usage [GB] | Preprocessing [sec] | Training [sec/epoch] |
|---|---|---|---|
| MPNN | 2.28 | 64 | 45.3 |
| NestedGNN | 13.72 | 2 354 | 107.8 |
| I2-GNN | 19.69 | 5 287 | 209.9 |
| 2-DRFWL | 2.31 | 430 | 141.9 |
| 0-$\ell$GIN | 0.02 (0.39) | 191 | 44.7 |
| 1-$\ell$GIN | 0.03 (0.48) | 370 | 66.4 |
| 2-$\ell$GIN | 0.05 (0.57) | 388 | 84.0 |
| 3-$\ell$GIN | 0.07 (0.66) | 408 | 85.2 |
| 4-$\ell$GIN | 0.10 (0.82) | 427 | 96.9 |
| 5-$\ell$GIN | 0.12 (0.91) | 444 | 130.6 |

# D   Preparation for Proofs

We begin by recalling some core concepts that are relevant for Section 5 and the proofs therein.

**Definition 10.** *A node invariant* $\zeta_{(\cdot)}$ *is a mapping that assigns to each graph* $G \in \mathcal{G}$ *a function* $\zeta_G : V(G) \to P$, *which satisfies*

$$\forall v \in V(G), \zeta_G(v) = \zeta_H(h(v)),$$

*where* $H$ *is any graph isomorphic to* $G$ *and* $h$ *is the corresponding isomorphism from* $H$ *to* $G$.

The following definition enables us to compare the expressive power of different node invariants.

**Definition 11** (Node Invariant Refinement). *Given two node invariants* $\gamma$ *and* $\zeta$. *We say that* $\zeta$ *refines* $\gamma$ *if for every fixed graph* $G$ *and nodes* $u, v \in V(G)$, *it holds* $\zeta_G(u) = \zeta_G(v) \Rightarrow \gamma_G(u) = \gamma_G(v)$. *We write* $\zeta \sqsubseteq \gamma$.

We emphasize that every node invariant $\zeta$ induces a graph invariant $\mathcal{A}[\gamma]$ by collecting the multiset, i.e., $G \mapsto \{\{\zeta_G(v)\}\}_{v \in V(G)}$. We denote the induced graph invariant of a node invariant $\gamma$ as $\mathcal{A}[\gamma]$.

The following lemma establishes a connection between the expressive power of two node invariants (see Definition 11) and that of their induced graph invariants (see Definition 3).

**Lemma 1.** *Let* $\zeta, \gamma$ *be node invariant. If* $\zeta \sqsubseteq \gamma$, *then* $\mathcal{A}[\zeta]$ *is more powerful than* $\mathcal{A}[\gamma]$.

*Proof.* Let $G, H$ be two graphs, and let $P$ be the underlying palette of $\zeta, \gamma$. Consider the function

$$\phi : P \longrightarrow P, \ \zeta(u) \mapsto \gamma(u) \ \forall u \in V(G) \cup V(H).$$

As a consequence of $\zeta \sqsubseteq \gamma$, $\phi$ is well-defined, since

$$\zeta(u) = \zeta(v) \implies (\phi \circ \zeta)(u) = \gamma(u) = \gamma(v) = (\phi \circ \zeta)(v).$$

Assume that $\mathcal{A}[\zeta](G) = \mathcal{A}[\zeta](H)$, i.e,

$$\{\{\zeta(u) \mid u \in V(G)\}\} = \{\{\zeta(v) \mid v \in V(H)\}\}.$$

As $\phi$ is well-defined, we have

$$\{\{\phi \circ \zeta(u) \mid u \in V(G)\}\} = \{\{\phi \circ \zeta(x) \mid v \in V(H)\}\},$$

which leads to $\mathcal{A}[\gamma](G) = \mathcal{A}[\gamma](H)$. $\qquad\square$

# E   Appendix for Section 5.1

In this section, we provide the proof of Proposition 1 from the main paper.

**Proposition 1.** *Let* $0 \le q < r$. *Then,* $r$-*ℓWL is strictly more powerful than* $q$-*ℓWL. In particular, every* $r$-*ℓWL is strictly more powerful than* 1-*WL.*

*Proof of Proposition 1.* Let $r \ge 0$. We aim to prove that $(r+1)$-ℓWL is strictly more powerful than $r$-ℓWL. We begin by demonstrating that $(r+1)$-ℓWL is more powerful than $r$-ℓWL.

To establish this, we rely on Lemma 1. Specifically, we demonstrate that the underlying $(r+1)$-ℓWL node invariant $c_{r+1}$ refines $c_r$. Moreover, we go beyond and show that the node invariant $c_{r+1}^{(t)}$ refines $c_r^{(t)}$ at every iteration $t \ge 0$, which shows that $t$ iterations of $(r+1)$-ℓWL are more powerful than $t$ iterations of $r$-ℓWL.

For this purpose, let $G$ be a graph with node set $V(G)$. For $t = 0$, $c_{r+1}^{(0)} \sqsubseteq c_r^{(0)}$ since both algorithms start with the same labels. By induction, we assume that

$$c_{r+1}^{(t)}(u) = c_{r+1}^{(t)}(v) \implies c_r^{(t)}(u) = c_r^{(t)}(v) \tag{5}$$

holds; we need to prove that (5) implies

$$c_{r+1}^{(t+1)}(u) = c_{r+1}^{(t+1)}(v) \implies c_r^{(t+1)}(u) = c_r^{(t+1)}(v). \tag{6}$$

Since HASH in Definition 7 is injective, $c_{r+1}^{(t)}(u) = c_{r+1}^{(t)}(v)$ in (5) leads to

$$\left\{\left\{ c_{r+1}^{(t)}(\mathbf{p}) \mid \mathbf{p} \in \mathcal{N}_q(u)\right\}\right\} = \left\{\left\{ c_{r+1}^{(t)}(\mathbf{p}) \mid \mathbf{p} \in \mathcal{N}_q(v)\right\}\right\}$$

for all $q \in \{0, \ldots, r\}$. The assumption $c_r^{(t)}(u_{l,k}^q) = c_r^{(t)}(v_{l,k}^q)$ in (5) is satisfied for every path $\mathbf{u}_l^q = \left\{ u_{l,k}^q \right\} \in \mathcal{N}_q(u)$ and $\mathbf{v}_l^q = \left\{ v_{l,k}^q \right\} \in \mathcal{N}_q(v)$ for $q = 0, \ldots, r$, $l = 1, \ldots, |\mathcal{N}_v|$ and $k = 1, \ldots, q+1$. Hence,

$$\left\{\left\{ c_r^{(t)}(\mathbf{p}) \mid \mathbf{p} \in \mathcal{N}_k(u)\right\}\right\} = \left\{\left\{ c_r^{(t)}(\mathbf{p}) \mid \mathbf{p} \in \mathcal{N}_k(v)\right\}\right\}$$

Inputting this into Definition 7, we get (6), i.e, $c_{r+1}^{(t+1)} \sqsubseteq c_r^{(t+1)}$.

The "strictly" can be deduced as follows. The cycle graph on $(2r+6)$ nodes equipped with a chord between nodes $1$ and $r+4$ is $r$-$\ell$WL equivalent to the graph consisting of two $(r+3)$-cycles connected by one edge; however, they are not $(r+1)$-$\ell$WL equivalent (see, e.g., Figure 6). $\qquad\square$

## F  Appendix for Section 5.2

The goal of this subsection is to provide a proof for Theorem 1 and Corollary 1. In fact, we present and prove a more general statement. Specifically, for a graph $G$ and $v \in V(G)$, we introduce the node invariant $\mathrm{sub}(F^x, G^v)$, defined as the count of subgraph isomorphisms from $F$ to $G$ that are rooted, meaning that $x$ is mapped to $v$. Let us denote this node invariant as $\mathrm{sub}(F^x, \cdot)$. Our result establishes that $c_r^{(1)}(\cdot)$ refines $\mathrm{sub}(C^x, \cdot)$ for every cycle graph $C$ with at most $r$ nodes. In simpler terms, $c_r^{(1)}$ can determine how often node $v$ appears in a cycle $C$.

**Lemma 2.** *Let $r \geq 1$. For every cycle graph $C$ with at most $r+2$ nodes and $x \in V(C)$, it holds $c_r^{(1)}(\cdot) \sqsubseteq \mathrm{sub}(C^x, \cdot)$.*

*Proof of Lemma 2.* Let $G$ be any graph, $u, v \in V(G)$, and $q = 1, \ldots, r+2$. Let $C$ be a cycle graph with $q$ nodes. It is important to note that for every $x_1, x_2 \in C$, we have $\mathrm{sub}(C^{x_1}, G^v) = \mathrm{sub}(C^{x_2}, G^v)$ since every node in $C$ is automorphic to each other. Therefore, we can arbitrarily choose any $x \in V(C)$.

We show that

$$\mathrm{sub}\left(C^x, G^u\right) \neq \mathrm{sub}\left(C^x, G^v\right) \implies c_r^{(1)}(u) \neq c_r^{(1)}(v)$$

The number of injective homomorphisms from the $q$-long cycles $C^x$ to $G^v$, i.e., $\mathrm{sub}(C_q^a, G^v)$, is equal to the number of paths of length $(q-2)$ between distinct neighbors of $v$.

The neighborhood $\mathcal{N}_{(q-2)}(v)$ comprises exactly all paths of length $(q-2)$ between any two distinct neighbors of $v$. Therefore,

$$\mathrm{sub}\left(C^x, G^v\right) = \left|\mathcal{N}_{(q-2)}(v)\right|.$$

Thus

$$\mathrm{sub}\left(C^x, G^u\right) \neq \mathrm{sub}\left(C^x, G^v\right) \implies \left|\mathcal{N}_{(q-2)}(u)\right| \neq \left|\mathcal{N}_{(q-2)}(v)\right|,$$

which implies

$$\left\{\left\{ c_{q-2}^{(0)}(\mathbf{p}) : \mathbf{p} \in \mathcal{N}_{(q-2)}(u)\right\}\right\} \neq \left\{\left\{ c_{q-2}^{(0)}(\mathbf{p}) : \mathbf{p} \in \mathcal{N}_{(q-2)}(v)\right\}\right\}.$$

Finally, as HASH in Definition 7 is injective, we get the thesis $c_{q-2}^{(1)}(u) \neq c_{q-2}^{(1)}(v)$. $\qquad\square$

Now, Theorem 1 from the main paper is a simple corollary of Lemma 2.

**Theorem 1.** *For any $r \geq 1$, $r$-$\ell$WL can subgraph-count all cycles with at most $r+2$ nodes.*

*Proof of Theorem 1.* Combining Lemma 2 and Lemma 1, we get that $c_r^{(1)}$ (as a graph invariant) is stronger than the induced graph invariant $\mathcal{A}\left[\mathrm{sub}(C^x, \cdot)\right]$. Now, consider graphs $G, H$, and assume without loss of generality that $|V(G)| = n = |V(H)|$.

If $c_r^{(1)}(G) = c_r^{(1)}(H)$, we have $\mathcal{A}\left[\mathrm{sub}(C^x, \cdot)\right](G) = \mathcal{A}\left[\mathrm{sub}(C^x, \cdot)\right](H)$. Hence, by definition of induced graph invariants,

$$\{\{\mathrm{sub}(C^x, G^v) \,|\, v \in V(G)\}\} = \{\{\mathrm{sub}(C^x, H^w) \,|\, w \in V(H)\}\}.$$

Hence,

$$\frac{1}{n}\sum_{v \in V(G)} \mathrm{sub}(C^x, G^v) = \frac{1}{n}\sum_{w \in V(H)} \mathrm{sub}(C^x, H^w),$$

which is equivalent to $\mathrm{sub}(C, G) = \mathrm{sub}(C, H)$. $\qquad\square$

We proceed to restate Corollary 1 and provide its proof.

**Corollary 1.** *Let $k \in \mathbb{N}$. There exists $r \in \mathbb{N}$, such that $r$-$\ell$WL is not less powerful than $k$-WL. Specifically, $r \in \mathcal{O}(k^2)$, with $r \leq \frac{k(k+1)}{2} - 2$ for even $k$ and $r \leq \frac{(k+1)^2}{2} - 2$ for odd $k$.*

*Proof of Corollary 1.* Let $k > 0$. We need to show that there exist $r_k \in \mathbb{N}$ and a pair of graphs $G, H$, such that $k - \mathrm{WL}(G) = k - \mathrm{WL}(H)$ and $r_k$-$\ell$WL$(G) \neq r_k$-$\ell$WL$(H)$.

The *hereditary treewidth* $\mathrm{hdtw}(F)$ of a graph $F$ is the maximum treewidth of $\varphi(F)$ where $\varphi$ is an edge surjective homomorphism. Neuen (2024) has shown that $k$-WL can subgraph-count a graph $F$ if and only if $\mathrm{hdtw}(F) \leq k$. This directly implies that for $F$ with hereditary treewidth larger than $k$, there exist graphs $G_F, H_F$ with $k - \mathrm{WL}(G_F) = k - \mathrm{WL}(H_F)$ and $\mathrm{sub}(F, G) \neq \mathrm{sub}(F, H)$.

Since the hereditary tree-width of cycle graphs is not uniformly bounded (Arvind et al., 2020), for every $k > 0$ there exists a cycle $C_{c_k}$ of length $c_k \in \mathbb{N}$ with hereditary treewidth larger than $k$. Setting $F = C_{c_k}$ concludes the existence proof with $r_k = c_k - 2$.

To see that $r_k \in O(k^2)$, note that the complete graph $K_n$ on $n$ vertices has treewidth $n - 1$ and exactly $\binom{n}{2}$ edges. For odd $n$, $K_n$ is Eulerian, i.e., there exists an edge surjective homomorphism from a cycle to $K_n$ which uses each edge exactly once, i.e., from $C_{\binom{n}{2}}$. If $n$ is odd, the minimum $T$-join which makes $K_n$ Eulerian contains exactly $\frac{n}{2}$ edges (see, e.g., Korte et al., 2018). As a result, there exists an edge surjective homomorphism from $C_{\binom{n}{2}}$ to $K_n$ if $n$ is odd, and an edge surjective homomorphism from $C_{\binom{n}{2} + \frac{n}{2}}$ to $K_n$ if $n$ is even. This implies that $\mathrm{hdtw}(C_{c_k}) > k$ for $c_k = \binom{k+1}{2} + \lceil \frac{k+1}{2} \rceil$. Hence, $r_k := c_k - 2 \in O(k^2)$. $\qquad\square$

# G  Appendix on Homomorphism Counting and Section 5.3

In this section, we provide background information and all proofs related to homomorphism counts. We begin by introducing additional definitions and notation.

**Definition 12** (Induced Subgraph). *Let $G = (V(G), E(G))$ and $S \subset V(G)$. The* induced subgraph *$G[S]$ of $G$ over $S$ is defined as the graph $G[S]$ with vertices $V(G[S]) = S$ and edges $E(G[S]) = \{\{u, v\} \in E(G) \,|\, u, v \in S\}$.*

The following definition indicates whether a pair of nodes is connected by an edge or not.

**Definition 13** (Atomic Type). *For a tuple of nodes $(u_1, u_2)$, the* atomic type $\mathrm{atp}_G((u_1, u_2))$ *of $G$ over $(u_1, u_2)$ indicates where $\{u_1, u_2\} \in E(G)$, i.e., $\mathrm{atp}_G((u_1, u_2)) = 1$ if $\{u_1, u_2\} \in E(G)$ and zero otherwise.*

We continue by defining *tree graphs*, an important class of graphs closely related to the 1-WL test.

**Definition 14** (Tree Graph). *A graph $T$ is called a* tree (graph) *if it is connected and does not contain cycles. A rooted tree $T^s = (V(T^s), E(T^s))$ is a tree in which a node $s \in V(T^s)$ is singled out. This node is called the* root *of the tree. For each vertex $t \in V(T^s)$, we define its depth $\mathrm{dep}_{T^s}(t) := \mathrm{dist}_{T^s}(t, s)$, where $\mathrm{dist}$ denotes the shortest path distance between $t$ and $s$. The* depth *of $T^s$ is then the maximum depth among all nodes $t \in V(T)$. We define $\mathrm{Desc}_{T^s}(t)$ the set of* descendants *of $t$, i.e., $\mathrm{Desc}_{T^s}(t) = \{t' \in T^s \,|\, \mathrm{dep}_{T^s}(t') = \mathrm{dep}_{T^s}(t) + \mathrm{dist}_{T^s}(t, t')\}$. For each $t \in V(T^s) \setminus \{s\}$, we define the* parent node $\mathrm{pa}_{T^s}(t)$ *of $t$ as the unique node $t' \in \mathcal{N}(t)$ such that $\mathrm{dep}_{T^s}(t) = \mathrm{dep}_{T^s}(t') + 1$. We define the* subtree *of $T^s$ rooted at node $t$ by $T^s[t]$, i.e., $T^s[t] := T^s[\mathrm{Desc}_{T^s}(t)]$.*

The remainder of this section is structured as follows. Appendix G.1 introduces the basics of tree decompositions. In Appendix G.2, we present the class of fan cactus graphs, encompassing all cactus graphs, and develop its canonical tree decomposition. We present an alternative formulation of $r$-$\ell$WL in Appendix G.3 for technical reasons. Subsequently, in Appendix G.4, we define the unfolding tree of $r$-$\ell$WL and illustrate its relation to the $r$-$\ell$WL colors and canonical tree decompositions of fan cactus graphs. Finally, in Appendix G.4.1, we establish the groundwork to conclude the proof of Theorem 2, a simple corollary of all the results in this section.

## G.1 Tree Decomposition Preliminaries

Along with its notation, this subsection closely adheres to the conventions outlined by B. Zhang et al. (2024, Section C). We start with a formal definition of a *tree decomposition* for a graph.

**Definition 15** (Tree Decomposition). *Let $G = (V(G), E(G))$. A* tree decomposition *of $G$ is a tree $T = (V(T), E(T))$ together with a function $\beta_T : V(T) \to 2^{V(G)}$ satisfying the following conditions:*

1. *Each tree node $t \in V(T)$ is mapped to a non-empty subset of vertices $\beta_T(t) \subset V(G)$ in $G$, referred to as a bag. We say tree node $t$ contains vertex $u$ if $u \in \beta_T(t)$.*

2. *For each edge $\{u, v\} \in E(G)$, there exists at least one tree node $t \in V(T)$ such that $\{u, v\} \subset \beta_T(t)$.*

3. *For each vertex $u \in V(G)$, all tree nodes $t$ containing $u$ form a connected subtree, i.e., the induced subgraph $T[\{t \in V(T) : u \in \beta_T(t)\}]$ is connected.*

*If $(T, \beta_T)$ is a tree decomposition of $G$, we refer to the tuple $(G, T, \beta_T)$ as a tree-decomposed graph. The width of the tree decomposition $T$ of $G$ is defined as*

$$\max_{t \in V(T)} |\beta_T(t)| - 1.$$

*If $T$ has root $s$, we also denote it as $(G, T^s, \beta_T)$.*

**Definition 16** (Treewidth). *The* treewidth *of a graph $G$, denoted as $\mathrm{tw}(G)$, is the minimum positive integer $k$ such that there exists a tree decomposition of width $k$.*

## G.2 Cactus Graphs and their Canonical Tree Decomposition

Cactus graphs play a crucial role in graph theory due to their unique structural properties. Before delving into their canonical tree decomposition, we define the concept of a rooted $r$-cactus graph. To simplify the notation, we assume that graphs in this section are connected and that $V(G) \subseteq \mathbb{N}$ for all graphs $G$. Further, we assume that $r \in \mathbb{N}$ throughout this section.

**Definition 17** (Rooted $r$-Cactus Graph). *A* cactus graph *is a graph where every edge lies on at most one simple cycle. An* $r$-cactus graph *is a cactus graph where every simple cycle has at most $r$ vertices. A* rooted cactus (graph) *$G^s$ is a cactus graph $G$ with a root node $s \in V(G)$.*

Now, we introduce the notion of a fan cactus, which is an essential concept for our subsequent discussions on the canonical tree decomposition of these graphs.

**Definition 18** (Fan Cactus). *Let $G^s$ be a rooted $r$-cactus. For every simple cycle $C$ in $G$ let $v_C$ be the unique vertex in $C$ that is closest to $s$. We obtain a* fan $r$-cactus $F^s$ *from a rooted $r$-cactus $G^s$ by adding an arbitrary number of edges $\{v_C, w\}$ to any cycle $C$ with $w \in V(C)$. Let $\mathcal{M}^{r+2}$ be the class of graphs $F$ with $s \in V(F)$ such that $F^s$ is a fan $r$-cactus.*

**Remark 1.** *Every $r$-cactus is a fan $r$-cactus. Every fan $r$-cactus is outerplanar. Every outerplanar graph has tree-width at most 2.*

Figure 10 shows an example of a fan 6-cactus. As fan cacti are outerplanar, graph isomorphism can be decided in linear time. One way to do so is to use a *canonicalization* function, that maps graphs to a unique representative of each set of isomorphic graphs. We denote the set of all such representatives as $\mathcal{M}^{r+2}/\cong \; \subseteq \mathcal{M}^{r+2}$.

**Lemma 3** (Colbourn et al. (1981)). *There exists a function* $\mathrm{canon} : \mathcal{M}^{r+2} \to \mathcal{M}^{r+2}/\cong$ *such that*

1. $G \cong \mathrm{canon}(G)$

2. $G \cong H \iff V(\mathrm{canon}(G)) = V(\mathrm{canon}(H)) \wedge E(\mathrm{canon}(G)) = E(\mathrm{canon}(H))$.

*Moreover, given $G \in \mathcal{M}^{r+2}$, $\mathrm{canon}(G)$ can be computed in linear time.*

For each $G \in \mathcal{M}^{r+2}$ we denote the isomorphism between $G$ and $\mathrm{canon}(G)$ as $\mathrm{canon}_G$. Colbourn et al. (1981) describe a bottom-up algorithm to obtain $\mathrm{canon}(G)$ of a fan $r$-cactus $G$. We will implicitly use the results of this canonicalization to define a canonical *tree decomposition* of fan $r$-cacti. The crucial point in the algorithm is a simple way to decide which "direction" to use when dealing with a cycle in the underlying cactus graph. Each undirected, rooted cycle allows for a choice between two directions when building a tree decomposition. We will first define a tree decomposition for a rooted cycle which depends on a choice of direction and then define a canonical direction of cycles in $G$ based on $\mathrm{canon}_G$.

**Definition 19** (Tree Decomposition of Rooted Cycle). *Let $C_n$ be a cycle graph on $n$ nodes $v_0$ to $v_{n-1}$. The path $T$ on nodes $w_1, \ldots, w_{2n-3}$ with bags $\beta(w_1) = \{v_0, v_1\}$ and for $i \geq 2$*

$$\beta(w_i) = \begin{cases} \beta(w_{i-1}) \cup \{v_{i/2+1}\} & \text{if } i \text{ is even} \\ \beta(w_{i-1}) \setminus \{v_{(i-1)/2}\} & \text{if } i \text{ is odd} \end{cases}$$

*is a tree decomposition of $C_n$. We say that $v_0$ and $v_1$ correspond to $w_1$ and $v_i$ corresponds to $w_{2i-1}$ for $i \geq 2$.*

A depiction of the tree decomposition $T^0$ (right) of $C_6$ (left) is shown below. Note that we have to choose one of two possible orientations of the undirected cycle to construct $T^0$. We address this choice in the next definition.

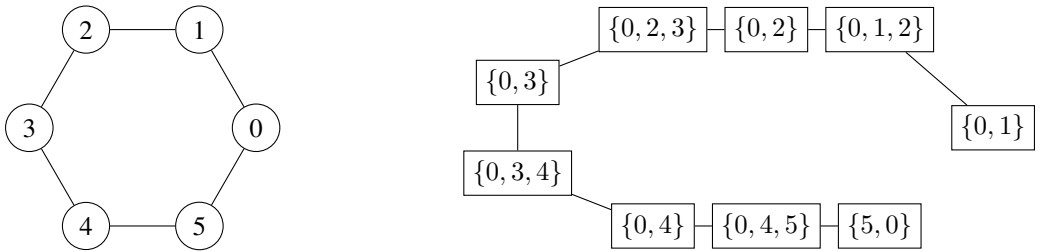

**Definition 20** (Canonical Tree Decomposition of Undirected Rooted Cycle). *Let $F^s$ be a fan $r$-cactus and $C$ be a simple cycle in the underlying cactus $G$. Let $v_C, v_1, \ldots, v_{n-1}$ and $v_C, v_{n-1}, \ldots, v_1$ be the two directions of $C$ rooted at $v_C$. We define the canonical tree decomposition of $C$ in $G$ as the tree decomposition of the smaller of the two orientations $\mathrm{canon}_F(v_C), \mathrm{canon}_F(v_1), \ldots, \mathrm{canon}_F(v_{n-1})$ and $\mathrm{canon}_F(v_C), \mathrm{canon}_F(v_{n-1}), \ldots, \mathrm{canon}_F(v_1)$.*

The choice of "smaller" does not matter as long as it defines a total order. One can, for example, use a lexicographical order. Based on Definition 20, we now define a canonical tree decomposition of fan cactus graphs, in the sense that any two isomorphic fan cactus graphs will have isomorphic tree decompositions.

**Definition 21** (Canonical Tree Decomposition of Fan $r$-Cactus Graphs). *Let $F^s$ be a fan $r$-cactus and $G^s$ its underlying $r$-cactus. We define the canonical tree decomposition $T^{\tilde{s}}$ of $F$ rooted at $\tilde{s}$ as follows*

1. ***Node Gadget:** For all $v \in V(F)$ add a node $t$ to $V(T)$ and set $\beta(t) = \{v\}$. We choose $\tilde{s}$ such that $\beta(\tilde{s}) = \{s\}$.*

2. ***Tree Edge Gadget:** For all $\{v, w\} \in E(G)$ that are not on a simple cycle in $F$ add a node $x_{\{v,w\}}$ to $V(T)$ with $\beta(x_{\{v,w\}}) = \{v, w\}$ and edges $\{v, x_{\{v,w\}}\}$ and $\{w, x_{\{v,w\}}\}$ to $E(T)$*

3. ***Cycle Gadget:** For each (undirected) cycle $C$ in the underlying cactus $G$, add a copy of its canonical tree decomposition $T_C^{v_C}$ of $C$ rooted at $v_C$ to $T$ and connect nodes in it to the corresponding node gadgets.*

See Figure 10 for an illustration. For the discussions in subsequent sections, we introduce the following definition.

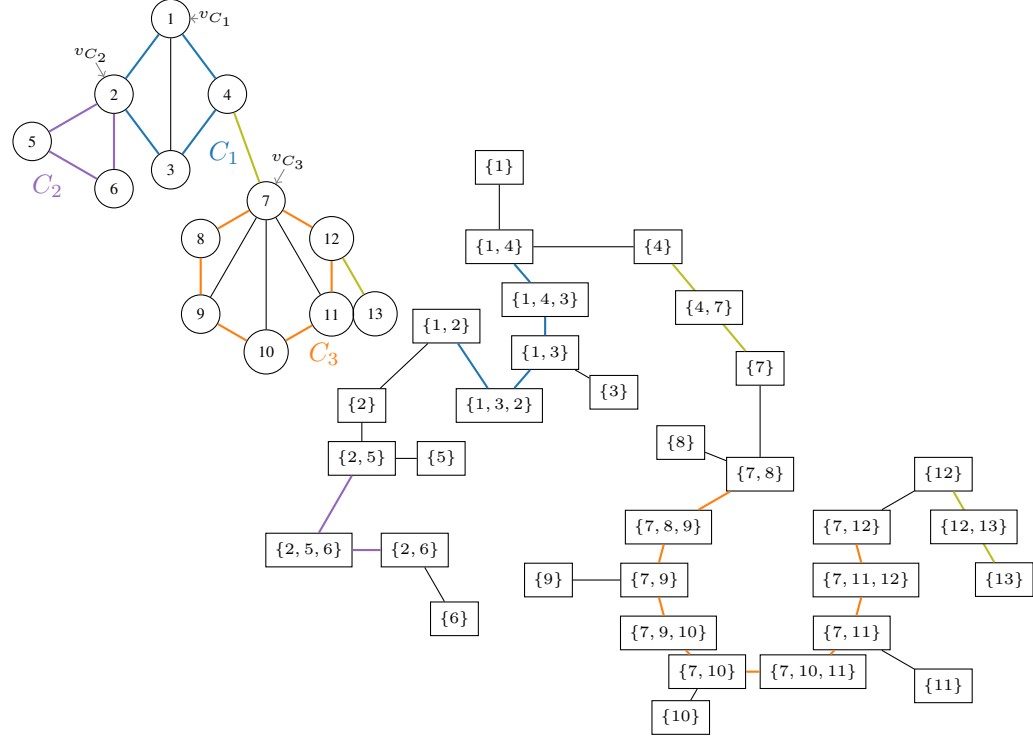

Figure 10: Example of a fan 6-cactus $F^1$ (left) and its canonical tree decomposition $(T, 1)$. The underlying rooted 6-cactus $G^1$ (on colored, thick edges) of $F^1$ contains three simple cycles $C_1, C_2, C_3$. Additional diagonal edges must have $v_{C_i}$ as one endpoint.

**Definition 22** (Depth in the Canonical Tree Decomposition of Fan $r$-Cactus Graphs). *Let $(F, T^s)$ be a canonical tree decomposition of a fan $r$-cactus. We define the* depth $\operatorname{dep}(t)$ *of $t \in V(T)$ recursively as follows:*

1. $\operatorname{dep}(s) = 0$

2. *For $v \in V(T)$ with parent node $p$:* $\operatorname{dep}(v) = \begin{cases} \operatorname{dep}(p) + 1 & \text{if } |\beta(v)| = 1 \text{ or } |\beta(p)| = 1 \\ \operatorname{dep}(p) & \text{otherwise} \end{cases}$

*The depth of $(F, T^s)$ is then the maximum depth of any node $t \in V(T^s)$.*

Intuitively, for a given fan $r$-cactus graph $F$ with its canonical tree decomposition $T^s$, Definition 22 captures the depth (see Definition 15) of the tree $T^s$, if cycles in $F$ and the corresponding bags in $T^s$ were replaced by single edges.

**Lemma 4.** *Let $F^s$ be a fan $r$-cactus. The canonical tree decomposition $(F, T^{\tilde{s}})$ is a tree decomposition of $F^s$.*

*Proof.* We need to show that (1) $T$ is a tree, (2) for every edge $e \in E(F)$ there exists some bag $\beta(v)$ with $e \subseteq \beta(v)$, and (3) $T[\{t \in V(T) : u \in \beta(t)\}]$ is connected.

To see that $T$ does not contain cycles, note that we replace each cycle with its cycle gadget, which is a path. It is easy to see that $T$ is connected as $G$ is connected.

For (2), note that tree edges $e \in V(F)$ have their own gadget node in $x_e$ with $\beta(x_e) = e$. Similarly, each edge $e$ on a simple cycle $C$ of the underlying cactus $F$ of $G$ is contained in some bag within the cycle gadget of $C$. Finally, for diagonal edges $\{v_C, w\} \in E(F) \setminus E(G)$, $v_C$ is contained in any bag of the cycle gadget of $C$. As a result, $\{v_C, v\}$ is contained in the bag of the corresponding node of $v$.

For (3), note that in the tree edge gadget, nodes $t$ with $v \in \beta(t)$ are connected to the node gadget of $v$. In the cycle gadget, any node $t$ with $w \in \beta(t)$ is either directly or via its neighbor connected to the node gadget of $w$ if $w \neq v_C$. As the cycle gadget is connected and $v_C$ is in any bag of the gadget, a path to the node gadget of $v_C$ exists where every bag contains $v_C$. $\square$

We conclude this subsection with a formal definition of when two canonical tree decompositions are isomorphic and prove the main result of this section, i.e., that canonical tree decompositions of fan $r$-cacti $G^s, H^t$ are isomorphic whenever $G^s, H^t$ are isomorphic.

**Definition 23** (Isomorphism between canonical tree-decomposed graphs). *Given two canonical tree-decomposed graphs $(G, T^s)$ and $(\tilde{G}, \tilde{T}^{\tilde{s}})$, a pair of mappings $(\rho, \tau)$ is called an* isomorphism *between $(G, T^s)$ and $(\tilde{G}, \tilde{T}^{\tilde{s}})$, denoted by $(G, T^s) \cong (\tilde{G}, \tilde{T}^{\tilde{s}})$, if the following holds:*

- *$\rho$ is an isomorphism between $G$ and $\tilde{G}$,*

- *$\tau$ is an isomorphism between $T^s$ and $\tilde{T}^{\tilde{s}}$,*

- *For any $t \in T^s$, we have $\rho(\beta_T(t)) = \beta_{\tilde{T}}(\tau(t))$.*

**Lemma 5.** *Let $G^s \cong H^t$ be rooted $r$-fan cacti. Then $(G^s, T[G^s]) \cong (H^t, T[H^t])$.*

*Proof.* Let $\rho$ be a root preserving isomorphism between $G^s$ and $H^t$. According to Lemma 3 then there exist isomorphisms $\mathrm{canon}_G$ and $\mathrm{canon}_H$ with $\rho = \mathrm{canon}_G \circ \mathrm{canon}_H^{-1}$. We construct $\tau : V(T[G^s]) \to V(T[H^t])$ from $\rho$ as follows: It is easy to see that $\rho$ induces a bijective mapping $\tau$ between the nodes of $T[G^s]$ and $T[H^t]$ that assigns each gadget node $v \in V(T[G^s])$ to the unique gadget node $\tau(v) \in V(T[H^t])$ with $\beta(\tau(v)) = \rho(\beta(v))$. By the same argument, $\tau$ maps the root of $T[G^s]$ to the root of $T[H^t]$.

Now assume by contradiction that $\tau$ is not an isomorphism between $T[G^s]$ and $T[H^t]$. That means that w.l.o.g. there exists $\{v, w\} \in E(T[G^s])$ with $\{\tau(v), \tau(w)\} \notin E(T[H^t])$. However, for the bags of $v, w$ it holds $\mathrm{canon}_G(\beta(v)) = \mathrm{canon}_H(\beta(\tau(v)))$ and $\mathrm{canon}_G(\beta(w)) = \mathrm{canon}_H(\beta(\tau(w)))$. This cannot happen, as the addition of edges in Definition 21 depends only on the images of the bags under $\mathrm{canon}$. $\square$

### G.3 Alternative $r$-$\ell$WL

In this subsection, we define slightly modified versions of 1-WL and $r$-$\ell$WL that we consider in the subsequent sections.

**Definition 24** (Alternative 1-WL and $r$-$\ell$WL). *The* alternative 1-WL test *refines vertices' colors as*

$$c^{(t+1)}(v) \leftarrow \mathrm{HASH}\left(c^{(t)}(v), \left\{\left\{\left(\mathrm{atp}(v, u), c^{(t)}(u)\right) \mid u \in V(G)\right\}\right\}\right).$$

*Equivalently, we define the* alternative $r$-$\ell$WL *via*

$$c_r^{(t+1)}(v) \leftarrow \mathrm{HASH}_r\left(c_r^{(t)}(v), \left\{\left\{\left(\mathrm{atp}(v, u), c^{(t)}(u)\right) \mid u \in V(G)\right\}\right\},\right.$$
$$\left\{\left\{c_r^{(t)}(\mathbf{p}) \mid \mathbf{p} \in \mathcal{N}_1(v)\right\}\right\},$$
$$\vdots$$
$$\left.\left\{\left\{c_r^{(t)}(\mathbf{p}) \mid \mathbf{p} \in \mathcal{N}_r(v)\right\}\right\}\right),$$

It is well-known that both the alternative 1-WL test and the standard 1-WL test are equally powerful (in terms of their expressive power). Similarly, the alternative $k$-WL test and the standard $k$-WL test are equally powerful. For the sake of simplicity in the subsequent discussion, we will refer to both the alternative 1-WL and $k$-WL tests simply as the 1-WL and $k$-WL tests, respectively. Although this practice may seem like a slight abuse of notation, it is justified because the expressive power of these tests remains unaffected.

Finally, as noted in Section 6, we alter the $r$-$\ell$WL algorithm slightly by incorporating atomic types into the path representation. Hence, we update node features according to

$$
c_r^{(t+1)}(v) \leftarrow \mathrm{HASH}_r \left( c_r^{(t)}(v), \left\{\left\{ \left( \mathrm{atp}(v, u), c^{(t)}(u) \right) \mid u \in V(G) \right\}\right\}, \right.
$$

$$
\left\{\left\{ \left( \mathrm{atp}(v, \mathbf{p}), c_r^{(t)}(\mathbf{p}) \right) \mid \mathbf{p} \in \mathcal{N}_1(v) \right\}\right\},
$$

$$
\vdots \tag{7}
$$

$$
\left. \left\{\left\{ \left( \mathrm{atp}(v, \mathbf{p}), c_r^{(t)}(\mathbf{p}) \right) \mid \mathbf{p} \in \mathcal{N}_r(v) \right\}\right\} \right),
$$

where $\left( \mathrm{atp}(v, \mathbf{p}), c_r^{(t)}(\mathbf{p}) \right) := \left( \left( \mathrm{atp}(v, p_1), c_r^{(t)}(p_1) \right), \ldots, \left( \mathrm{atp}(v, p_{q+1}), c_r^{(t)}(p_{q+1}) \right) \right)$ for $\mathbf{p} = \{p_i\}_{i=1}^{q+1} \in \mathcal{N}_q(v)$. The definition of atomic types $\mathrm{atp}(\cdot, \cdot)$ is given in Definition 13. Clearly this version of $r$-$\ell$WL is more powerful than the standard version, according to Definition 3. However, it is unclear whether $r$-$\ell$WL with atomic types is strictly more powerful than standard $r$-$\ell$WL.

### G.4   The Unfolding Tree of $r$-$\ell$WL

Given Definition 20, we assume, for the remainder of this appendix, that every fan cactus graph has a unique labeling function, allowing us to select a unique orientation for every cycle in the graph. We call this orientation the *canonical orientation*. If not otherwise mentioned, we consider the canonical orientation of cycle graphs.

We begin this section by introducing a critical concept known as *bag isomorphism* (Dell et al., 2018; B. Zhang et al., 2024).

**Definition 25** (Bag Isomorphism). *Let $(F, T^s)$ be a tree-decomposed graph, and $G$ be a graph. A homomorphism $f$ from $F$ to $G$ is called a* bag isomorphism *from $(F, T^s)$ to $G$ if, for all $t \in V(T^s)$, the mapping $f$ is an isomorphism from $F[\beta_{T^s}(t)]$ to $G[f(\beta_{T^s}(t))]$. We denote by $\mathrm{BIso}((F, T^s), G)$ the set of all bag isomorphisms from $(F, T^s)$ to $G$, and set $\mathrm{bIso}((F, T^s), G) = |\mathrm{BIso}((F, T^s), G)|$.*

Moving forward, we proceed to define $r$-$\ell$WL unfolding trees, which intuitively construct, for a given graph and a node in the graph, the computational graph of the $r$-$\ell$WL algorithm and its canonical tree decomposition.

**Definition 26** (Unfolding tree of $r$-$\ell$WL). *Given a graph $G$, vertex $v \in V(G)$ and a non-negative $D \in \mathbb{Z}$, the* depth-$2D$ $r$-$\ell$WL unfolding tree *of a graph $G \in \mathcal{M}^{r+2}$ at node $v$, denoted as $\left( F^{(D)}(v), T^{(D)}(v) \right)$, is a tree-decomposition $(F, T^s)$ constructed in the following way:*

1. ***Initialization**: $V(F) = \{v\}$ without edges, and $T^s$ only has a root node $s$ with $\beta_{T^s}(s) = \{v\}$. Define a mapping $V(F) \to V(G)$ as $\pi(v) = v$.*

2. ***Introduce nodes**: For each leaf node $t$ with $|\beta_{T^s}(t)| = 1$ in $T^s$, do the following procedure:*

   *Let $\beta_T(t) = \{g\}$. For each $w \in V(G)$ do the following:*

   a) *Add a fresh child $t_w$ to $t$ in $T^s$.*
   b) *Add a fresh vertex $f$ to $F$ and extend $\pi$ with $[f \mapsto w]$.*
   c) *Define the bag of $t_w$ by $\beta_{T^s}(t_w) = \beta_{T^s}(t) \cup \{f\}$.*
   d) *Add an edge between $f$ and $g$ if $\{\pi(f), \pi(g)\} \in E(G)$.*

3. ***Introduce paths**: For each $q = 1, \ldots, r$, do:*
   *For each length $q$ path with canonical orientation $\mathbf{p} = \{p_i\}_{i=1}^{q+1} \in \mathcal{N}_q(g)$, do the following:*

   a) *Add a fresh path $\mathbf{t_p} = \{t_{\{p_1\}}, t_{\{p_1, p_2\}}, t_{\{p_2\}}, \ldots, t_{\{p_q\}}, t_{\{p_q, p_{q+1}\}}, t_{\{p_{q+1}\}}\}$ to $t$ in $T^s$.*
   b) *Add $q + 1$ fresh vertices $f_1, \ldots, f_{q+1}$ to $F$ and extend $\pi$ with $[f_i \mapsto p_i]$ for every $i = 1, \ldots, q + 1$.*

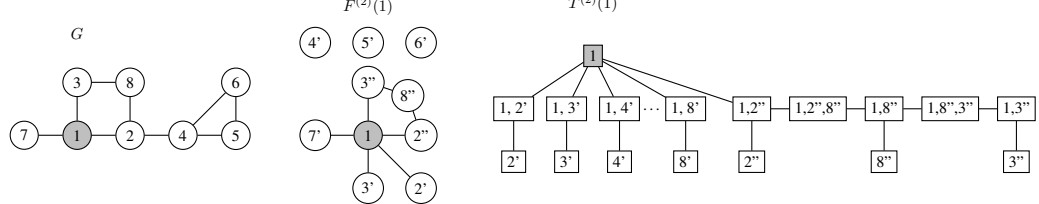

Figure 11: The depth-2 unfolding tree of graph $G$ at vertex 1 for 2-$\ell$WL.

c) For $i = 1, \ldots, q$, let the bag of $t_{\{p_i, p_{i+1}\}}$ be defined via $\beta_{T^s}(t_{\{p_i, p_{i+1}\}}) = \beta_{T^s}(t) \cup \{f_i, f_{i+1}\}$.

d) For $i = 1, \ldots, q+1$, let the bag of $t_{\{p_i\}}$ be defined via $\beta_{T^s}(t_{\{p_i\}}) = \beta_{T^s}(t) \cup \{f_i\}$.

e) For $i = 1, \ldots, q+1$, add edges between $f_i$ and $f_{i+1}$.

f) Add edges between $g$ and $f_1, \ldots, f_{q+1}$ such that for every $i = 1, \ldots, q$, we have $F[\beta_{T^s}(t_{\{p_i, p_{i+1}\}})] = F[\{f_i, f_{i+1}, g\}] \cong G[\pi(\beta_{T^s}(t_{\{p_i, p_{i+1}\}}))]$, i.e., add edges between $g$ and $f_i$ if and only if there is an edge between $\{\pi(g), \pi(f_i)\} \in E(G)$.

4. **Forget nodes**: If $t$ is a leaf node of $T^s$ with $|\beta_{T^s}(t)| = 2$ and parent $t'$ with $|\beta_{T^s}(t)| = 1$, do the following:

   a) Add a fresh child $t_1$ of $t$ to $T^s$.

   b) Let $f$ be that vertex introduced at $t$, that is, we have $\beta_{T^s}(t) \setminus \beta_{T^s}(t') = \{f\}$.

   c) We set $\beta_{T^s}(t_1) = \{f\}$.

5. **Forget paths**: If $\mathbf{t_p} = \{t_{\{p_1\}}, t_{\{p_1, p_2\}}, t_{\{p_2\}}, \ldots, t_{\{p_q\}}, t_{\{p_q, p_{q+1}\}}, t_{\{p_{q+1}\}}\}$ is a leaf path of $T^s$ with parent $t'$ of $t_{\{p_1\}}$, do the following:

   a) For $i = 2, \ldots, q+1$, add a fresh child $\tilde{t}_{\{p_i\}}$ to $t_{\{p_i\}}$.

   b) Let $f_2, \ldots, f_{q+1}$ be the vertices introduced at $\mathbf{t_p}$, that is, we have $\beta_{T^s}(t_{\{p_i\}}) \setminus \beta_{T^s}(t') = \{f_i\}$.

   c) For $i = 2, \ldots, q+1$, we set $\beta_{T^s}(\tilde{t}_{\{p_i\}}) = \{f_i\}$.

We refer to Figure 11 for the depth-2 2-$\ell$WL unfolding tree of an example graph.

**Theorem 4.** Let $r \geq 1$. For any graph $G$, any vertex $v \in V(G)$, and any non-negative integer $D$, let $\left(F^{(D)}(v), T^{(D)}(v)\right)$ be its depth-$2D$ $r$-$\ell$WL unfolding tree at node $v$. Then, $F^{(D)}(v)$ is a fan $r$-cactus graph, and $T^{(D)}(v)$ is an $r$-canonical tree decomposition of $F^{(D)}(v)$. Moreover, the constructed mapping $\pi$ in Definition 26 is a bag isomorphism from $\left(F^{(D)}(v), T^{(D)}(v)\right)$ to the graph $G$.

*Proof.* Clear by the definition of the depth-$2D$ unfolding tree of $r$-$\ell$WL. $\qquad\square$

We present the following results that fully characterize when two graphs and their respective nodes have the same $r$-$\ell$WL colors in terms of their $r$-$\ell$WL unfolding trees.

**Theorem 5.** Let $r \in \mathbb{N}$. For any two connected graphs $G, H$, any vertices $v \in V(G)$ and $x \in V(H)$ and any $D \in \mathbb{N}$, it holds: $c_r^{(D)}(v) = c_r^{(D)}(x)$ if and only if there exists a root preserving isomorphism between $\left(F^{(D)}(v), T^{(D)}(v)\right)$ and $\left(F^{(D)}(x), T^{(D)}(x)\right)$.

*Proof of "$\Longrightarrow$".* The proof is based on induction over $D$. When $D = 0$, the theorem obviously holds. Assume that the theorem holds for $D \leq d$, and consider $D = d + 1$. We show that if $c_r^{(d+1)}(v) = c_r^{(d+1)}(x)$, then there exists an isomorphism $(\rho, \tau)$ from $\left(F^{(d+1)}(v), T^{(d+1)}(v)\right)$ to $\left(F^{(d+1)}(x), T^{(d+1)}(x)\right)$ such that $\rho(v) = x$.

If $c_r^{(d+1)}(v) = c_r^{(d+1)}(x)$, then

$$\left\{\!\!\left\{\left(\mathrm{atp}(v, u), c_r^{(d)}(u)\right) \mid u \in V(G)\right\}\!\!\right\} = \left\{\!\!\left\{\left(\mathrm{atp}(x, y), c_r^{(d)}(y)\right) \mid y \in V(H)\right\}\!\!\right\},$$

i.e., $|V(G)| = |V(H)|$, and we set $n = |V(G)|$. We enumerate $V(G) = \{w_1, \ldots, w_n\}$ and $V(H) = \{z_1, \ldots, z_n\}$ such that

$$c_r^{(d)}(w_i) = c_r^{(d)}(z_i) \tag{8}$$

for all $i = 1, \ldots, n$. Also, again since $c_r^{(d+1)}(v) = c_r^{(d+1)}(x)$, we have for every $q = 1, \ldots, r$,

$$\left\{\left\{\left(\left(\mathrm{atp}(v, u_1), c_r^{(d)}(u_1)\right), \ldots, \left(\mathrm{atp}(v, u_{q+1}), c_r^{(d)}(u_{q+1})\right)\right) \mid \{u_1, \ldots, u_{q+1}\} = \mathbf{u} \in \mathcal{N}_q(v)\right\}\right\}$$
$$= \left\{\left\{\left(\left(\mathrm{atp}(x, y_1), c_r^{(d)}(y_1)\right), \ldots, \left(\mathrm{atp}(x, y_{q+1}), c_r^{(d)}(y_{q+1})\right)\right) \mid \{y_1, \ldots, y_{q+1}\} = \mathbf{y} \in \mathcal{N}_q(x)\right\}\right\}.$$

In particular, $|\mathcal{N}_q(v)| = |\mathcal{N}_q(x)|$ and we can enumerate the paths in $\mathcal{N}_q(v)$ and $\mathcal{N}_q(x)$ such that

$$c_r^{(d)}(\mathbf{u}_l^q) = c_r^{(d)}(\mathbf{y}_l^q) \quad \text{and} \quad \mathrm{atp}(v, \mathbf{u}_l^q) = \mathrm{atp}(v, \mathbf{y}_l^q) \tag{9}$$

for every $l = 1, \ldots, |\mathcal{N}_q(v)|$.

Now, by definition of the $r$-$\ell$WL unfolding tree, the graph $F^{(d+1)}(v)$ is isomorphic to the union of: a) all graphs $F^{(d)}(w_i)$ for $i = 1, \ldots, n$, plus additional edges between $w_i$ to $v$ if $\{w_i, v\} \in E(G)$, and b) all graphs $F^{(d)}(p_{l,k}^q)$ for $q = 1, \ldots, r$, $l = 1, \ldots, |\mathcal{N}_q(v)|$ for any path $\mathbf{p}_l^q = \left\{p_{l,1}^q, \ldots, p_{l,q+1}^q\right\} \in \mathcal{N}_q(v)$. And adding, for $k = 1, \ldots q$, edges between $p_{l,k}^q$ and $p_{l,k+1}^q$. And adding, for $k = 1, \ldots q+1$, edges $p_{l,k}^q$ and $v$ if there is one in $G$, i.e., if $\left\{p_{l,k}^q, v\right\} \in E(G)$.

Similarly, the tree $T^{(d+1)}(v)$ is isomorphic to the disjoint union of all trees $T^{(d)}(w_i)$ (for $i = 1, \ldots, n$) and $T^{(d)}(p_{l,k}^q)$ (for $q = 1, \ldots, r$, $k = 1, \ldots, q+1$ and $l = 1, \ldots, |\mathcal{N}(v)|$). Plus adding the following fresh tree nodes and edges: a root node $s$, nodes $t_{w_i}$ (for $i = 1, \ldots, n$) that connects to $s$ and the root of $T^{(d)}(w_i)$. And for $q = 1, \ldots, r$, $l = 1, \ldots, |\mathcal{N}_q(v)|$ for any path $\mathbf{p}_l^q \in \mathcal{N}_q(v)$ a path of length $2q$, given by $\mathbf{t}_{\mathbf{p}_l^q} = \left\{t_{\left\{p_{l,1}^q\right\}}, t_{\left\{p_{l,1}^q, p_{l,2}^q\right\}}, \ldots, t_{\left\{p_{l,q}^q, p_{l,q+1}^q\right\}}, t_{\left\{p_{l,q+1}^q\right\}}\right\}$, where $s$ is attached to $t_{\left\{p_{l,1}^q\right\}}$. And finally, connecting the trees $T^{(d)}(p_{l,k}^q)$ at root node $p_{l,k}^q$ to $t_{\left\{p_{l,k}^q\right\}}$.

By (8) and induction, there exist isomorphisms $(\rho_i, \tau_i)$ from $(F^{(d)}(w_i), T^{(d)}(w_i))$ to $(F^{(d)}(z_i), T^{(d)}(z_i))$ such that $\rho_i(w_i) = z_i$ for $i = 1, \ldots, n$. By (9) and induction, there exist isomorphisms $(\rho_{l,k}^q, \tau_{l,k}^q)$ from $(F^{(d)}(u_{l,k}^q), T^{(d)}(u_{l,k}^q))$ to $(F^{(d)}(y_{l,k}^q), T^{(d)}(y_{l,k}^q))$ such $\rho_i(u_{l,k}^q) = y_{l,k}^q$ for $q = 1, \ldots, r$, $k = 1, \ldots, q+1$ and $l = 1, \ldots, |\mathcal{N}_q(v)|$.

We now construct $\rho$ by merging all $\rho_i$ and $\rho_{l,k}^q$, and construct $\tau$ by merging all $\tau_i$ and $\tau_{l,k}^q$. We finally specify an appropriate mapping for the tree root, its direct children and the paths attached to the tree root. Then, it is easy to see that $(\rho, \tau)$ is well-defined and an isomorphism between $\left(F^{(d+1)}(v), T^{(d+1)}(v)\right)$ and $\left(F^{(d+1)}(x), T^{(d+1)}(x)\right)$ such that $\rho(v) = x$. $\qquad \square$

*Proof of "$\Longleftarrow$".* We now prove the other direction, again via induction over $D$. When $D = 0$ the assertion obviously holds. Assume that the assertion holds for $D \leq d$. Now, assume that there exists an isomorphism $(\rho, \tau)$ between $\left(F^{(d+1)}(v), T^{(d+1)}(v)\right)$ and $\left(F^{(d+1)}(x), T^{(d+1)}(x)\right)$ such that $\rho(v) = x$. We show that $c_r^{(d+1)}(v) = c_r^{(d+1)}(x)$.

We begin our proof by establishing the equality of two multisets: $\left\{\left\{(c_r^{(d)}(w), \mathrm{atp}(v, w)) \mid w \in V(G)\right\}\right\}$ and $\left\{\left\{(c_r^{(d)}(z), \mathrm{atp}(v, z)) \mid z \in V(H)\right\}\right\}$. The proof of this equivalence closely mirrors the argument presented in the proof of B. Zhang et al. (2024, Lemma C.14). Since $\tau$ is an isomorphism it maps all tree nodes $T^{(d+1)}(v)$ of depth 2 with 1 element in their bag to the corresponding tree nodes in $T^{(d+1)}(x)$. Let $s_1, \ldots, s_n$ and $t_1, \ldots, t_n$ be the nodes in $T^{(d+1)}(v)$ and $T^{(d+1)}(x)$ of depth 2 with 1 element in their bag, respectively. For $i = 1, \ldots, n$, let $s_i'$ and $t_i'$ the parents of $s_i$ and $t_i$, respectively. We then choose the order such that the following holds for all $i = 1, \ldots, n$

1. Let $\beta_{T^{(d+1)}(v)}(s_i') = \{v, \tilde{w}_i\}$ and $\beta_{T^{(d+1)}(x)}(t_i') = \{x, \tilde{z}_i\}$. Then, $\rho(v) = x$ and $\rho(\tilde{w}_i) = \tilde{z}_i$ and thus, per assumption, $\{v, \tilde{w}_i\} \in E(F^{(d+1)}(v))$ if and only if $\{x, \tilde{z}_i\} \in E(F^{(d+1)}(x))$.

2. $\tau$ is an isomorphism from the subtree rooted at $s_i$ in $T^{(d+1)}(v)$, i.e., $T^{(d+1)}(v)[s_i]$, the subtree rooted at $t_i$ in $T^{(d+1)}(x)$, i.e., $T^{(d+1)}(v)[t_i]$.

3. For all $s \in \mathrm{Desc}_{T^{(d+1)}(v)}(s_i)$, it holds $\rho(\beta_{T^{(d+1)}(v)}(s)) = \beta_{T^{(d+1)}(x)}(\tau(s))$.

4. By the definition of the unfolding tree, $\rho$ is an isomorphism from the induced subgraph $F^{(d+1)}(v)\left[T^{(d+1)}(v)[s_i]\right]$ and the induced subgraph $F^{(d+1)}(x)\left[T^{(d+1)}(x)[t_i]\right]$.

By the last three items, we get that $\left(F^{(d+1)}(v)\left[T^{(d+1)}(v)[s_i]\right], T^{(d+1)}(v)[s_i]\right)$ and $\left(F^{(d+1)}(x)\left[T^{(d+1)}(x)[t_i]\right], T^{(d+1)}(x)[t_i]\right)$ are isomorphic. By definition of the $r$-$\ell$WL unfolding tree, $\left(F^{(d+1)}(v)\left[T^{(d+1)}(v)[s_i]\right], T^{(d+1)}(v)[s_i]\right)$ is isomorphic to $(F^{(d)}(w_i), T^{(d)}(w_i))$ for some $w_i \in V(G)$ that satisfies $\{\tilde{w}_i, v\} \in E_{F^{(d+1)}(v)}$ if and only if $\{w_i, v\} \in E(G)$. And $\left(F^{(d+1)}(x)\left[T^{(d+1)}(x)[t_i]\right], T^{(d+1)}(x)[t_i]\right)$ is isomorphic to $(F^{(d)}(z_i), F^{(d)}(z_i))$ for some $z_i \in V(H)$ that satisfies $\{\tilde{z}_i, x\} \in E(F^{(d+1)}(x))$ if and only if $\{z_i, x\} \in E(G)$. Hence, by induction, we have $\mathrm{atp}(v, w_i) = \mathrm{atp}(x, z_i)$ and $c_r^{(d)}(w_i) = c_r^{(d)}(z_i)$ for all $i = 1, \ldots, n$.

It remains to show that, for every $q = 1, \ldots, r$,

$$\left\{\!\left\{\left(\left(\mathrm{atp}(v, u_1), c_r^{(d)}(u_1)\right), \ldots, \left(\mathrm{atp}(v, u_{q+1}), c_r^{(d)}(u_{q+1})\right)\right) \mid \{u_1, \ldots, u_{q+1}\} = \mathbf{u} \in \mathcal{N}_q(v)\right\}\!\right\}$$
$$= \left\{\!\left\{\left(\left(\mathrm{atp}(x, y_1), c_r^{(d)}(y_1)\right), \ldots, \left(\mathrm{atp}(x, y_{q+1}), c_r^{(d)}(y_{q+1})\right)\right) \mid \{y_1, \ldots, y_{q+1}\} = \mathbf{y} \in \mathcal{N}_q(x)\right\}\!\right\}.$$

Fix $q = 1, \ldots, r$. Since $\tau$ is an isomorphism it maps all paths of length $q$ in $T^{(d+1)}(v)$ connected to $v$ to paths of length $q$ in $T^{(d+1)}(x)$ connected to $x$.

By construction of the $r$-$\ell$WL unfolding tree and since $(\rho, \tau)$ is an isomorphism, it holds $|\mathcal{N}_q(v)| = |\mathcal{N}_q(x)|$. Denote the relevant bags at depth 2 by $s_{l,k}'^q$ for $l = 1, \ldots, |\mathcal{N}_q(v)|$ and $k = 1, \ldots, q+1$. Denote by $s_{l,k}^q$ and $t_{l,k}'^q$ the parents of $s_{l,k}'^q$ and $t_{l,k}'^q$, respectively. We then choose the order $l = 1, \ldots, |\mathcal{N}_q(v)|$ and $k = 1, \ldots, q+1$ such that it holds

1. Let $\beta_{T^{(d+1)}}(s_{l,k}'^q) = \left\{v, \tilde{w}_{l,k}^q\right\}$ and $\beta_{T^{(d+1)}}(t_{l,k}'^q) = \left\{x, \tilde{z}_{l,k}^q\right\}$. Then, $\rho(\tilde{w}_{l,k}^q) = \tilde{z}_{l,k}^q$ and thus, per assumption, $\left\{v, \tilde{w}_{l,k}^q\right\} \in E(F^{(d+1)}(v))$ if and only if $\left\{x, \tilde{z}_{l,k}^q\right\} \in E(F^{(d+1)}(x))$

2. $\tau$ is an isomorphism from the subtree rooted at $s_{l,k}^q$ in $T^{(d+1)}(v)$, i.e., $T^{(d+1)}(v)[s_{l,k}^q]$, to the subtree rooted at $t_{l,k}^q$ in $T^{(d+1)}(x)$, i.e., $T^{(d+1)}(x)[t_{l,k}^q]$.

3. For all $s \in \mathrm{Desc}_{T^{(d+1)}(v)}(s_{l,k}^q)$, it holds $\rho(\beta_{T^{(d+1)}(v)}(s)) = \beta_{T^{(d+1)}(x)}(\tau(s))$.

4. By the definition of the unfolding tree, $\rho$ is an isomorphism from the induced subgraph $F^{(d+1)}(v)\left[T^{(d+1)}(v)[s_{l,k}^q]\right]$ and the induced subgraph $F^{(d+1)}(x)\left[T^{(d+1)}(x)[t_{l,k}^q]\right]$.

By the last three items, we get that $\left(F^{(d+1)}(v)\left[T^{(d+1)}(v)[s_{l,k}^q]\right], T^{(d+1)}(v)[s_{l,k}^q]\right)$ and $\left(F^{(d+1)}(x)\left[T^{(d+1)}(x)[t_{l,k}^q]\right], T^{(d+1)}(x)[t_{l,k}^q]\right)$ are isomorphic. By definition of the $r$-$\ell$WL unfolding tree, $\left(F^{(d+1)}(v)\left[T^{(d+1)}(v)[s_{l,k}^q]\right], T^{(d+1)}(v)[s_{l,k}^q]\right)$ is isomorphic to $(F^{(d)}(w_{l,k}^q), T^{(d)}(w_{l,k}^q))$ for $w_{l,k}^q \in V(G)$ that satisfies $\left\{\tilde{w}_{l,k}^q, v\right\} \in E_{F^{(d+1)}(v)}$ if and only if $\left\{w_{l,k}^q, v\right\} \in E(G)$.

And $\left(F^{(d+1)}(x)\left[T^{(d+1)}(x)[t_{l,k}^q]\right], T^{(d+1)}(x)[t_{l,k}^q]\right)$ is isomorphic to $(F^{(d)}(z_{l,k}^q), T^{(d)}(z_{l,k}^q))$ for $z_{l,k}^q \in V(G)$ that satisfies $\left\{\tilde{z}_{l,k}^q, v\right\} \in E(F^{(d+1)}(v))$ if and only if $\left\{z_{l,k}^q, v\right\} \in E(G)$. Hence, by induction, we have $c_r^{(d)}(w_{l,k}^q) = c_r^{(d)}(z_{l,k}^q)$ for all indices. By Item 1, we then have $c_r^{(d)}(\mathbf{w_l^q}) = c_r^{(d)}(\mathbf{z_l^q})$ and $\mathrm{atp}(v, \mathbf{w}_l^q) = \mathrm{atp}(v, \mathbf{z}_l^q)$ for every $q = 1, \ldots, r$ and $l = 1, \ldots, |\mathcal{N}_q(v)|$. $\qquad \square$

We introduce the following definition that provides a similarity measure between a graph and a tree-decomposed graph.

**Definition 27.** *Given a graph $G$ and a tree-decomposed graph $(F, T^s)$, define*

$$\mathrm{cnt}\left((F, T^s), G\right) = \left|\left\{ v \in V \mid \exists D \in \mathbb{N} \text{ s.t. } (F^{(D)}(v), T^{(D)}(v)) \cong (F, T^s)\right\}\right|,$$

*where $(F^{(D)}(v), T^{(D)}(v))$ is the depth-$2D$ $r$-$\ell$WL unfolding tree of $G$ at $v$.*

The counting function $\mathrm{cnt}\left((F, T^s), G\right)$ serves as a key metric, allowing us to draw connections between $r$-$\ell$WL colorings of two different graphs.

**Corollary 3.** *Let $r \in \mathbb{N}$. Let $G$ and $H$ be two graphs. Then, $c_r(G) = c_r(H)$ if and only if $\mathrm{cnt}\left((F, T^s), G\right) = \mathrm{cnt}\left((F, T^s), H\right)$ holds for all graphs $(F, T^s) \in \mathcal{M}^{r+2}$.*

*Proof of "$\Longrightarrow$".* Let $c_r(G) = c_r(H)$, i.e.,

$$\{\{c_r(v) \mid v \in V(G)\}\} = \{\{c_r(x) \mid x \in V(H)\}\}.$$

Assume, by contradiction, that there exists a tuple $(F, T^s) \in \mathcal{M}^{r+2}$ such that $\mathrm{cnt}\left((F, T^s), G\right) \neq \mathrm{cnt}\left((F, T^s), H\right)$. Let $c_1, \ldots, c_k$ be the final colors of nodes in $V(G)$ and $V(H)$. Then, define for $i = 1, \ldots, k$

$$\mathrm{cnt}\left((F, T^s), G[c_i]\right) := \left|\left\{ v \in V(G) \mid c_r(v) = c_i \text{ and } \exists D \in \mathbb{N} \text{ s.t. } (F^{(D)}(v), T^{(D)}(v)) \cong (F, T^s)\right\}\right|.$$

We have

$$\mathrm{cnt}\left((F, T^s), G\right) = \sum_{i=1}^{k} \mathrm{cnt}\left((F, T^s), G[c_i]\right),$$

and

$$\mathrm{cnt}\left((F, T^s), H\right) = \sum_{i=1}^{k} \mathrm{cnt}\left((F, T^s), H[c_i]\right).$$

Since $\mathrm{cnt}\left((F, T^s), G\right) \neq \mathrm{cnt}\left((F, T^s), H\right)$, there exist an index $i = 1, \ldots, k$ such that

$$\mathrm{cnt}\left((F, T^s), G[c_i]\right) \neq \mathrm{cnt}\left((F, T^s), H[c_i]\right). \tag{10}$$

Furthermore, there exists $i_n \in \mathbb{N}$ such that there are exactly $n$ nodes $v_1, \ldots, v_n$ and $x_1, \ldots, x_n$ such that

$$c_r(v_1) = \ldots = c_r(v_{i_n}) = c_i \text{ and } c_r(x_1) = \ldots = c_r(x_{i_n}) = c_i.$$

Hence, as $c_r$ refines $c_r^{(D)}$, we have

$$c_r^{(D)}(v_1) = \ldots = c_r^{(D)}(v_{i_n}) = c_r^{(D)}(x_1) = \ldots = c_r^{(D)}(x_{i_n}).$$

By (10), there exists some $D \in \mathbb{N}$ such that (without loss of generality) $(F^{(D)}(v_1), T^{(D)}(v_1)) \cong (F, T^s)$. Then, by Theorem 5, we have

$$(F^{(D)}(v_1), T^{(D)}(v_1)) \cong \ldots \cong (F^{(D)}(v_{i_n}), T^{(D)}(v_{i_n})) \cong (F^{(D)}(x_{i_n}), T^{(D)}(x_{i_n}))$$
$$\cong \ldots \cong (F^{(D)}(x_1), T^{(D)}(x_1)).$$

There does not exist any other node $w$ with $c_r(w) = c_1$ such that the corresponding unfolding tree is isomorphic to $(F^{(D)}(v_1), T^{(D)}(v_1))$. Hence, $\mathrm{cnt}((F, T^s), G[c_i]) = \mathrm{cnt}((F, T^s), H[c_i])$, which is a contradiction.

*Proof of "$\Longleftarrow$".* Suppose that $\mathrm{cnt}((F, T^s), G) = \mathrm{cnt}((F, T^s), H)$ for all $(F, T^s) \in \mathcal{M}^{r+2}$. Let $c_1, \ldots, c_{k_G}$ with multiplicities $m_1, \ldots, m_{k_G}$ and $\tilde{c}_1, \ldots, \tilde{c}_{k_H}$ with multiplicities $\tilde{m}_1, \ldots, \tilde{m}_{k_G}$ be the final colors of $r$-$\ell$WL applied to $G$ and $H$, respectively. Consider some $v \in V(G)$ such that $c_r(v) = c_1$. Let $D$ be sufficiently large (any $D$ after convergence of $r$-$\ell$WL), then $\mathrm{cnt}((F^{(D)}(v), T^{(D)}(v)), G) = \mathrm{cnt}((F^{(D)}(v), T^{(D)}(v)), H)$ since $(F^{(D)}(v), T^{(D)}(v)) \in \mathcal{M}^{r+2}$. Hence, without loss of generality, $c_1 = \tilde{c}_1$ and $m_1 = \tilde{m}_1$. Repeating this argument for all colors finishes the proof. $\square$

### G.4.1 Proof of Theorem 2

In this section, we employ techniques adapted from the works of Dell et al. (2018) and B. Zhang et al. (2024) to derive a proof for Theorem 2 from the established result in Corollary 3.

**Definition 28** (Definition 20 in (Dell et al., 2018)). *Let $(F, T^t)$ and $(\tilde{F}, \tilde{T}^s)$ be two tree-decomposed graphs. A pair of mappings $(\rho, \tau)$ is said to be a* bag isomorphism homomorphism *from $(F, T^t)$ to $(\tilde{F}, \tilde{T}^s)$ if it satisfies the following conditions*

1. *$\rho$ is a homomorphism from $F$ to $\tilde{F}$.*

2. *$\tau$ is a homomorphism from $T^t$ to $\tilde{T}^s$.*

3. *$\tau$ is depth-surjective, i.e., the image of $T^t$ under $\tau$ contains vertices at every depth present in $\tilde{T}^s$.*

4. *For all $t' \in T^t$, we have $\mathrm{dep}_{T^t}(t') = \mathrm{dep}_{\tilde{T}^s}(\tau(t'))$ and $F[\beta_{T^t}(t')] \cong \tilde{F}[\beta_{\tilde{T}^s}(\tau(t'))]$.*

5. *For all $t' \in T^t$, the set equality $\rho(\beta_{T^t}(t')) = \beta_{\tilde{T}^s}(\tau(t'))$ holds.*

6. *The depth of $T^t$ and $\tilde{T}^s$ is equal.*

*We denote the set of bag isomorphism homomorphisms from $(F, T^t)$ to $(\tilde{F}, \tilde{T}^s)$ by $\mathrm{BIsoHom}\left((F, T^t), (\tilde{F}, \tilde{T}^s)\right)$ and set $\mathrm{bIsoHom}\left((F, T^t), (\tilde{F}, \tilde{T}^s)\right) = |\mathrm{BIsoHom}\left((F, T^t), (\tilde{F}, \tilde{T}^s)\right)|$.*

We continue with the following lemma that shows a linear relation between the number of bag isomorphisms and the output of the counting function in Definition 27.

**Lemma 6.** *Let $r \in \mathbb{N}$. For any tree-decomposed graph $(F, T^s) \in \mathcal{M}^{r+2}$ and any graph $G$, it holds*

$$\mathrm{bIso}\left((F, T^s), G\right) = \sum_{(\tilde{F}, \tilde{T}^t) \in \mathcal{M}^{r+2}} \mathrm{bIsoHom}\left((F, T^s), \left(\tilde{F}, \tilde{T}^t\right)\right) \cdot \mathrm{cnt}\left(\left(\tilde{F}, \tilde{T}^t\right), G\right). \quad (11)$$

*Proof.* Let $(F, T^s)$ be a tree-decomposed graph such that $T^s$ has depth $2D$. The sum is over all isomorphism types $(\tilde{F}, \tilde{T}^t)$ of tree-decomposed graphs. This sum is finite and thus well-defined as $\mathrm{bIsoHom}\left((F, T^s), \left(\tilde{F}, \tilde{T}^t\right)\right) = 0$ holds if $\tilde{T}^t$ has depth unequal to $2D$ or nodes with at least $(r+1) \cdot (|V(G)| - 1)$ children.

Assume that for the root bag of $(F, T^s)$ it holds $\beta_{T^s}(s) = \{v\}$. Let $x \in V(G)$ be any vertex in $G$, and denote by $(F^{(D)}(x), T^{(D)}(x))$ the depth-$2D$ $r$-$\ell$WL-unfolding tree at node $x$. Define the following two sets,

$$S_1(x) = \{h \in \mathrm{BIso}((F, T^s), G) \mid h(v) = x\},$$
$$S_2(x) = \left\{(\rho, \tau) \in \mathrm{BIsoHom}\left((F, T^s), \left(F^{(D)}(x), T^{(D)}(x)\right)\right) \mid \rho(v) = x\right\}.$$

We prove that $|S_1(x)| = |S_2(x)|$ for every $x \in V(G)$, which is equivalent to (11). For this, we show for any bag isomorphism $h$ from $(F, T^s)$ to $G$ with $h(v) = x$, there exists a *unique* bag isomorphism homomorphism $\sigma$ from $(F, T^s)$ to $(F^{(D)}(x), T^{(D)}(x))$ with $\sigma(v) = x$ such that $h = \pi \circ \sigma$, where $\pi$ is the bag isomorphism from $(F^{(D)}(x), T^{(D)}(x))$ to $G$, defined in Definition 26 and Theorem 4, respectively. To visualize this proof idea, see Figure 12.

First, define $\rho(v) := x$. Let $v_1, \ldots, v_n \in V(F)$ be nodes that correspond to bags in $T^s$ of depth 2 with one element inside the bag and their parents having two elements in their bag, i.e., $\{v_i\}$ are the corresponding bags. Similarly, set $x_1, \ldots, x_m \in V(F^{(D)}(x))$ nodes that correspond to bags of depth 2 in $T^{(D)}(x)$, with one element inside the bag and their parents having two elements in their bag. Since $h$ is a bag isomorphism and $\pi$ as well, for every $i = 1, \ldots, n$ there exists a $j_i$ such that $h(v_i) = \tilde{x}_{j_i} = \pi(x_{j_i})$, where $\tilde{x}_{j_i} \in V(G)$ and $x_{j_i} \in V\left(F^{(D)}(x)\right)$. Since $\pi$ and $h$ are bag isomorphisms, we have

$$F[\{\{v, v_i\}\}] \cong G[\{\{x, \tilde{x}_{j_i}\}\}] \cong F^{(D)}(x)[\{\{x, x_{j_i}\}\}]. \quad (12)$$

Now, set $\rho(v_i) = x_{j_i}$ for every $i = 1, \ldots, n$. Based on (12), we can easily define $\tau$ such that $\tau$ satisfies Definition 28 with respect to bags that are of depth 1 and 2.

For $q = 1, \ldots, r$ and $l = 1, \ldots, |\mathcal{N}_q(v)|$, let $\mathbf{p}_l^q$ be a path of length $2q$ starting from the root node $s$ in $T^s$. Every such path $\mathbf{p}_l^q$ in $T^s$ corresponds to unique path $\mathbf{v}_l^q$, that is in $\mathcal{N}_q(x)$, of length $q$ in $F$. We represent the path by $\left\{ v_{l,1}^q, v_{l,2}^q, \ldots, v_{l,q+1}^q \right\}$, where every consecutive node is connected to each other and for $k = 1, \ldots q + 1$, we have $\left\{ v, v_{l,k}^q \right\} \in E(F)$ iff $\left\{ h(v), h(v_{l,k}^q) \right\} \in E(G)$ as $h$ is a bag isomorphism. Further for every node $k = 1, \ldots, q + 1$ there exists a $j_{l,k}^q$ such that $h(v_{l,k}^q) = \tilde{x}_{j_{l,k}^q} = \pi(x_{j_{l,k}^q})$, where $\tilde{x}_{j_{l,k}^q} \in V(G)$, $\left\{ \tilde{x}_{j_{l,1}^q}, \ldots, \tilde{x}_{j_{l,q+1}^q} \right\} \in \mathcal{N}_q(x)$ and $\left\{ x_{j_{l,1}^q}, \ldots, x_{j_{l,q+1}^q} \right\} \in \mathcal{N}_q(x)$. We set $\sigma(v_{l,k}^q) = x_{j_{l,k}^q}$ for every $k = 1, \ldots, q + 1$. Clearly, we have

$$F\left[\left\{\{v, v_i, v_{i+1}\}\right\}\right] \cong G\left[\left\{\{x, \tilde{x}_{j_i}, \tilde{x}_{j_{i+1}}\}\right\}\right] \cong F^{(D)}(x)\left[\left\{\{x, x_{j_i}, x_{j_{i+1}}\}\right\}\right]. \tag{13}$$

Now, based on (13), we can easily define $\tau$ such that $\tau$ satisfies Definition 28 with respect to bags that correspond to paths in $\mathcal{N}_q(v)$ for $q = 1, \ldots, r$. Now, following this construction recursively leads to a bag isomorphism $\rho$ such that $h = \pi \circ \rho$.

It remains to show that $(\rho, \tau)$ is unique (up to isomorphism). For this, let $(\rho_1, \tau_1)$ be another bag isomorphism homomorphism between $(F, T^s)$ and $(F^{(D)}(x), T^{(D)}(x))$ such that $\rho_1(v) = x$ and $h = \pi \circ \rho_1$. We show that $\rho = \rho_1$.

We begin by showing that $\rho(v) = \rho_1(v)$ for every $v$ that is not in a cycle. Adopting the previous notations, consider $v_1, \ldots, v_n \in V(F)$ and $x_1, \ldots, x_m \in V(F^{(D)}(x))$. For each $i = 1, \ldots, n$, let $k_i$ and $l_i$ be the indices such that $\rho(v_i) = x_{k_i}$ and $\rho_1(v_i) = x_{l_i}$. Consequently, $\pi(x_{k_i}) = \pi(x_{l_i})$. We note that the image of $h(v_i)$ is not contained in a cycle in $G$, as otherwise, $h$ would not be a bag isomorphism. Similarly, $x_{k_i}$ and $x_{l_i}$ are not contained in a cycle; otherwise, $\rho$ and $\rho_1$ would not be bag isomorphisms. Now, $\pi$ is an injective mapping if the domain is restricted to nodes that are of depth 1 and 2, and not contained in a cycle. Hence, $x_{k_i} = x_{l_i}$.

We continue by showing that for every $w \in V(F)$, that is contained in a cycle, we have $\rho(w) = \rho_1(w)$. This follows a similar argument as the nodes that are not included in any cycle. We summarize the argument shortly: It must hold that $\rho(w)$ and $\rho_1(w)$ are contained in a cycle, and $\pi(\rho(w))$ and $\pi(\rho_1(w))$ as well. Now, $\pi$ is injective if the domain is restricted to nodes that are only contained in cycles. Hence, $\rho_1 = \rho$. $\qquad \square$

We continue this subsection by introducing the concept of a *bag extension* in the context of tree-decomposed graphs. This definition formalizes the notion of one tree-decomposed graph being an extension of another.

**Definition 29** (Definition 20 in (Dell et al., 2018))**.** *Let $(F, T^t)$ be a tree-decomposed graph. A bag extension of $(F, T^t)$ is a graph $(H, T^t)$ with $V(H) = V(F)$ such that for every $t \in V(T^t)$ the induced subgraph $H[\beta_{T^t}(t)]$ is an extension of $F[\beta_{T^t}(T)]$, i.e., if $e \in E(F[\beta_{T^t}(T)])$, then $e \in E(H[\beta_{T^t}(T)])$. We define* $\mathrm{bExt}\left((F, T^t), (\tilde{F}, \tilde{T}^s)\right)$ *as the number of bag extensions of $(F, T^t)$ that are isomorphic to $(\tilde{F}, \tilde{T}^s)$.*

Intuitively, a bag extension of a tree-decomposed graph $(F, T^s)$ can be achieved by adding an arbitrary number of edges to $F$. Each added edge must be contained within a bag that corresponds to a node in the tree $T^s$.

**Definition 30** (Definition C.28 in (B. Zhang et al., 2024))**.** *Given a tree-decomposed graph $(F, T^r)$ and a graph $G$, a* bag-strong homomorphism *from $(F, T^s)$ to $G$ is a homomorphism $f$ from $F$ to $G$ such that, for all $t \in V(T^r)$, $f$ is a strong homomorphism from $F[\beta_{T^s}(t)]$ to $G[f(\beta_{T^s}(t))]$, i.e., $\{u, v\} \in E(F[\beta_{T^s}(t)])$ iff $\{f(u), f(v)\} \in E(G[f(\beta_{T^s}(t))])$. Denote $\mathrm{BStrHom}((F, T^s), G)$ to be the set of all bag-strong homomorphisms from $(F, T^s)$ to $G$, and denote $\mathrm{bStrHom}((F, T^s), G) = |\mathrm{BStrHom}((F, T^s), G)|$.*

We continue with decomposing the number of homomorphism from a fan cactus graph to any graph.

**Lemma 7.** *Let $r \in \mathbb{N}$. For any tree-decomposed graph $(F, T^s) \in \mathcal{M}^{r+2}$ and any graph $G$, it holds*

$$\hom(F, G) = \sum_{(\tilde{F}, \tilde{T}^t) \in \mathcal{M}^{r+2}} \mathrm{bExt}\left((F, T^s), (\tilde{F}, \tilde{T}^t)\right) \cdot \mathrm{bStrHom}\left((\tilde{F}, \tilde{T}^t), G\right). \tag{14}$$

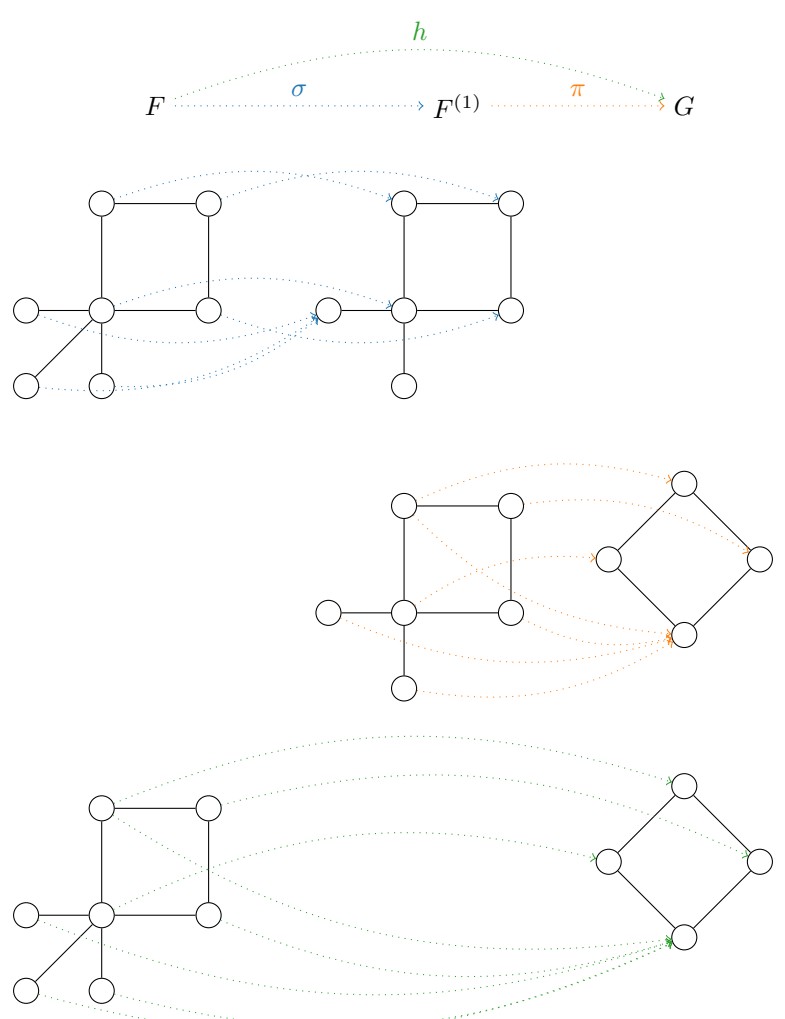

Figure 12: Visualization of proof idea of Lemma 6

*Proof.* The proof follows the lines of Lemma C.29. in (B. Zhang et al., 2024). First, (14) is well-defined as $T^s$ is finite, hence, there can only be finitely many bag extensions of $(F, T^s)$.

Further, consider the set

$$S = \left\{ \left( \left( \tilde{F}, \tilde{T}^t \right), (\rho, \tau), g \right) \mid \left( \tilde{F}, \tilde{T}^t \right) \in \mathcal{M}^{r+2}, (\rho, \tau) \in \mathrm{BExt} \left( (F, T^s), \left( \tilde{F}, \tilde{T}^t \right) \right), \right.$$
$$\left. g \in \mathrm{BstrHom} \left( \left( \tilde{F}, \tilde{T}^t \right), G \right) \right\}.$$

We consider the mapping $\sigma$ from $S$ to $\hom(F, G)$ via $((\rho, \tau), g) \mapsto g \circ \rho$. We show that for every homomorphism $h$ there exists a unique, up to automorphisms, $\left( \tilde{F}, \tilde{T}^t \right) \in \mathcal{M}^{r+2}, (\rho, \tau)$ and $g$ such that $h = g \circ \rho$.

We begin with the existence part. For $h \in \hom(F, G)$, we define $\left( \tilde{F}, \tilde{T}^t \right) \in \mathcal{M}^{r+2}, (\rho, \tau)$ and $g$ as follows.

- We define $\tilde{F}$ by adding the edges given by

$$\{\{u, v\} \mid u, v \in V(F), \exists t \in T^s \text{ s.t. } \{u, v\} \in \beta_{T^s}(t), \{h(u), h(v)\} \in E(G)\}. \quad (15)$$

We define $\tilde{T}^t := T^s$. Clearly, $\left(\tilde{F}, \tilde{T}^t\right) \in \mathcal{M}^{r+2}$ and it is a bag extension as only edges are added that are contained within a bag that corresponds to a node in $T^s$.

- We define $\rho$ and $\tau$ as the identity mappings on their respective domain, leading to $(\rho, \tau) \in$ BExt $\left((F, T^s), \left(\tilde{F}, \tilde{T}^t\right)\right)$.

- We define $g = h$. For $x \in \tilde{T}^t$, we show that $g$ is a strong homomorphism from $\tilde{F}[\beta_{\tilde{T}^t}(x)]$ to $G[g\left(\beta_{\tilde{T}^t}(x)\right)]$. Let $\{u, v\} \in E\left(\tilde{F}[\beta_{\tilde{T}^t}(x)]\right)$, then $\{g(u), g(v)\} \in E\left(G[g\left(\beta_{\tilde{T}^t}(x)\right)]\right)$ as $h$ is a homomorphism with respect to the edges $E(F)$ and in (15) only edge $\{u, v\}$ were added that satisfy $\{h(u), h(v)\} \in E(G)$. On the other hand $\{g(u), g(v)\} \in E\left(G[g\left(\beta_{\tilde{T}^t}(x)\right)]\right)$, but $\{u, v\} \notin E\left(\tilde{F}[\beta_{\tilde{T}^t}(x)]\right)$ would contradict (15) as $u, v$ are contained in the same bag $\beta_{\tilde{T}^t}(x)$. Hence, $g \in$ BstrHom $\left(\left(\tilde{F}, \tilde{T}^t\right), G\right)$.

We finally prove the uniqueness part, i.e., that $\sigma\left((\tilde{F}_1, \tilde{T}_1^{t_1}), (\rho_1, \tau_1), g_1\right) = h$ implies that there exists an isomorphism $(\tilde{\rho}, \tilde{\tau})$ from $\left(\tilde{F}_1, \tilde{T}_1^{t_1}\right)$ to $\left(\tilde{F}, \tilde{T}^t\right)$ such that $\tilde{\rho} \circ \rho_1 = \rho$, $\tilde{\tau} \circ \tau_1 = \tau$. We first prove that $\tilde{F}_1 \cong \tilde{F}$ and $\tilde{T}_1^{t_1} \cong \tilde{T}^t$.

1. For any $u, v \in V(F)$, we obviously have $\rho(u) = \rho(v)$ iff $u = v$ iff $\rho_1(u) = \rho_1(v)$ as $\rho$ and $\rho_1$ are injective mappings.

2. Let $u, v \in V(F)$. Consider $\{\rho_1(u), \rho_1(v)\} \in E(\tilde{F}_1)$, we show that $\{\rho(u), \rho(v)\} \in E(\tilde{F})$. If $\{u, v\} \in E(F)$, then clearly $\{\rho(u), \rho(v)\} \in E(\tilde{F})$ as $\rho$ is a homomorphism. Hence, assume that $\{u, v\} \notin E(F)$. Then, $u, v$ must be contained in the same bag of $T^s$ as $\rho_1$ is a bag extension and only node pairs are added if they are in the same bag. Hence, $\rho(u)$ and $\rho(v)$ are contained in the same bag. As $g_1$ is a homomorphism, we have $\{g_1(\rho_1(u)), g_1(\rho_1(v))\} \in E(G)$. But, then also $\{g(\rho(u)), g(\rho(v))\} \in E(G)$, and as $g$ is a strong homomorphism (with respect to the bag in which $\rho(u)$ and $\rho(v)$ are contained), we have $\{\rho(u), \rho(v)\} \in E(\tilde{F})$. By symmetry of the argument, we have $\{\rho_1(u), \rho_1(v)\} \in E(\tilde{F}_1)$ iff $\{\rho(u), \rho(v)\} \in E(\tilde{F})$.

3. Since $\rho_1$ and $\rho$ are bag extension, they are bijective on their respective domain. Hence, $\tilde{\rho} = \rho \circ \rho_1^{-1}$ defines an isomorphism from $\tilde{F}_1$ to $\tilde{F}$. On the other hand, $\tilde{T}_1^{t_1} \cong \tilde{T}^t$ trivially holds, again with $\tilde{\tau} = \tau \circ \tau_1^{-1}$.

We have $\tilde{\rho} \circ \rho_1 = \rho$, $\tilde{\tau} \circ \tau_1 = \tau$. We show that the tuple $(\tilde{\rho}, \tilde{\tau})$ is an isomorphism, i.e., it remains to show that for any $b \in \tilde{T}_1^{t_1}$, we have $\tilde{\rho}(\beta_{\tilde{T}_1^{t_1}}(b)) = \beta_{\tilde{T}^t}(\tilde{\tau}(b))$. Since $\tau_1$ is surjective, we can choose $a$ such that $\tau_1(a) = b$. Then,

$$\tilde{\rho}(\beta_{\tilde{T}_1^{t_1}}(\tau_1(a))) = \tilde{\rho}(\rho_1(\beta_{T^s}(a))) = \rho(\beta_{T^s}(a)) = \beta_{\tilde{T}^t}(\tau(a)) = \beta_{\tilde{T}^t}(\tilde{\tau} \circ \tau_1(a)) = \beta_{\tilde{T}^t}(\tilde{\tau}(b)).$$

The first and third equalities hold since $(\rho_1, \tau_1)$ and $(\rho, \tau)$ are bag extensions. $\qquad\square$

**Definition 31** (Definition 30 in (B. Zhang et al., 2024)). *Given two tree-decomposed graphs $(F, T^s)$ and $(\tilde{F}, \tilde{T}^t)$, a homomorphism $(\rho, \tau)$ from $(F, T^s)$ to $(\tilde{F}, \tilde{T}^t)$ is called* bag-strong surjective *if $\rho$ is a bag-strong homomorphism from $(F, T^s)$ to $\tilde{F}$ and is surjective on both vertices and edges, and $\tau$ is an isomorphism from $T^s$ to $\tilde{T}^t$ such that for all $x \in V(T^s)$, we have $\rho(\beta_{T^s}(x)) = \beta_{\tilde{T}^t}(\tau(x))$. Denote $\mathrm{BStrSurj}((F, T^s), (\tilde{F}, \tilde{T}^t))$ to be the set of all bag-strong subjective homomorphisms from $(F, T^s)$ to $(\tilde{F}, \tilde{T}^t)$, and denote $\mathrm{bStrSurj}((F, T^s), (\tilde{F}, \tilde{T}^t)) = |\mathrm{BStrSurj}((F, T^s), (\tilde{F}, \tilde{T}^t))|$.*

**Lemma 8.** *Let $r \in \mathbb{N}$. For any tree-decomposed graph $(F, T^s) \in \mathcal{M}^{r+2}$ and any graph $G$, it holds*

$$\mathrm{bStrHom}\left((F, T^s), G\right) = \sum_{(\tilde{F}, \tilde{T}^t) \in \mathcal{M}^{r+2}} \mathrm{bStrSurj}\left((F, T^s), \left(\tilde{F}, \tilde{T}^t\right)\right) \frac{\mathrm{bIso}\left(\left(\tilde{F}, \tilde{T}^t\right), G\right)}{\mathrm{aut}\left(\tilde{F}, \tilde{T}^t\right)}, \tag{16}$$

*where $\mathrm{aut}(\tilde{F}, \tilde{T}^t)$ counts the number of automorphisms of $(\tilde{F}, \tilde{T}^t)$.*

*Proof.* The proof follows the lines of Lemma C.31. in (B. Zhang et al., 2024).

Consider the set

$$S = \left\{ \left( \left( \tilde{F}, \tilde{T}^t \right), (\rho, \tau), g \right) \mid \left( \tilde{F}, \tilde{T}^t \right) \in \mathcal{M}^{r+2}, (\rho, \tau) \in \text{BStrSurj}\left( (F, T^s), \left( \tilde{F}, \tilde{T}^t \right) \right), \right.$$
$$\left. g \in \text{BIso}\left( \left( \tilde{F}, \tilde{T}^t \right), G \right) \right\}.$$

We consider the mapping $\sigma$ from $S$ to $\text{BStrHom}\left( (F, T^s), G \right)$ via $((\rho, \tau), g) \mapsto g \circ \rho$. We show that for every bag-strong homomorphism $h$ there exists a unique, up to automorphisms, $\left( \tilde{F}, \tilde{T}^t \right) \in \mathcal{M}^{r+2}$, bag-strong surjective homomorphism $(\rho, \tau)$ and $g$ such that $h = g \circ \rho$.

We begin with the existence part. For $h \in \text{BStrHom}\left( (F, T^s), G \right)$, we define $\left( \tilde{F}, \tilde{T}^t \right) \in \mathcal{M}^{r+2}, (\rho, \tau)$ and $g$ as follows.

We define $\tilde{F}$ by defining an equivalence relation $\sim$ on $V(F)$: $u \sim v$ if $h(u) = h(v)$ and there exists a path $P$ in $T^s$ with endpoints $t_1, t_2 \in V(T^s)$ such that $u \in \beta_{T^s}(t_1), v \in \beta_{T^s}(t_2)$, and all nodes $t$ on the path $P$ satisfies that $h(u) = h(v) \in h(\beta_{T^s}(t))$. We then define $\rho$ as the quotient map with respect to $\sim$ and set $\tilde{F} = F/\sim$, i.e.,

$$V(\tilde{F}) = \{\rho(u) \mid u \in V(F)\}, E(\tilde{F}) = \{\{\rho(u), \rho(v)\} \mid \{u, v\} \in E(F)\},$$

which is well-defined as $\{u, v\} \in E(F)$ imples $\rho(u) \neq \rho(v)$ since $h$ is a homomorphism. Then, $\rho$ is surjective per construction.

We define the mapping $g : V(\tilde{F}) \to V(G)$ such that $g(\rho(u)) = h(u)$ for all $u \in V(F)$. This mapping $g$ is well-defined since $\rho(u) = \rho(v)$ implies $h(u) = h(v)$, and $\rho : V(F) \to V(\tilde{F})$ is surjective. This leads to the equality $h = g \circ \rho$. To demonstrate that $g$ is a homomorphism, consider any edge $(x, y) \in E(\tilde{F})$. There exists an edge $(u, v) \in E(F)$ such that $\rho(u) = x$ and $\rho(v) = y$, which implies $(h(u), h(v)) \in E(G)$, since $h$ is a homomorphism. Consequently, this means $(g(x), g(y)) \in E(G)$.

We continue by defining the tree $\tilde{T}^t := (V(T), E(T), \beta_{\tilde{T}^t})$. We set $t = s$, and define $\tau$ to be the identity. Furthermore, we have $\beta_{\tilde{T}^t}(x) = \rho(\beta_{T^s}(x))$ for all $x \in V(T)$. It remains to prove that $(\tilde{F}, \tilde{T}^t) \in \mathcal{M}^{r+2}$ is a valid tree decomposition. For this, it suffices to prove that for any vertex $x \in V(\tilde{F})$ the subgraph $B_{\tilde{T}^t}(x)$ is connected. For this, let $x \in V(\tilde{F})$ and $t_1, t_2 \in B_{\tilde{T}^t}(x)$. Then, there exists $u \in \beta_{T^s}(t_1), v \in \beta_{T^s}(t_2)$ such that $\rho(u) = x, \rho(v) = x$. Therefore, $u \sim v$. As such, there exists a path $P \in T^s$ such that all nodes $b$ on $P$ satisfy $h(u) \in h(\beta_{T^s}(b))$. Hence, for every $b \in P$ there exists some $w_b \in \beta_{T^s}(b)$ such that $h(w_b) = h(u)$, and consequently $w_b \sim u$. Finally, $x = \rho(u) = \rho(w_b) \in \rho(\beta_{T^s}(b)) = \beta_{\tilde{T}^t(b)}$ for all $b$ in the path $P$. Hence, $\left( \tilde{F}, \tilde{T}^t \right) \in \mathcal{M}^{r+2}$.

It remains to prove that $\rho$ is a bag-strong surjective homomorphism and $g$ is a bag isomorphism. We begin by showing that $\rho$ is a bag-strong surjective homomorphism. For this, let $t \in V(T^s)$ and $u, v \in \beta_{T^s}(t)$. If $\{u, v\} \notin E(F)$, then $\{h(u), h(v)\} \notin E(G)$ (since $h$ is a bag-strong homomorphism). Therefore, $\{\rho(u), \rho(v)\} \notin E(\tilde{F})$ since $g$ is a homomorphism. Hence, $\rho$ is a bag-strong surjective homomorphism.

We show that $g$ is a bag isomorphism. Let $x \in V(\tilde{T}^t)$, and consider $\tilde{u}, \tilde{v} \in \beta_{\tilde{T}^t}(x)$. Since $\rho$ is surjective, there exist $u, v \in \beta_{T^s}(x)$ such that $\rho(u) = \tilde{u}$ and $\rho(v) = \tilde{v}$. We have $\{\rho(u), \rho(v)\} \notin E(\tilde{F})$ iff $\{h(u), h(v)\} \notin E(G)$, since both $\rho$ and $h$ are bag-strong homomorphisms. Therefore, g is a bag isomorphism.

We finally prove that $\sigma\left( (\tilde{F}_1, \tilde{T}^{t_1}), (\rho_1, \tau_1), g_1 \right) = \sigma\left( (\tilde{F}, \tilde{T}^t), (\rho, \tau), g \right)$ implies there exists an isomorphism $(\tilde{\rho}, \tilde{\tau})$ from $(\tilde{F}_1, \tilde{T}_1^{t_1})$ to $(\tilde{F}, \tilde{T}^t)$ such that $\tilde{\rho} \circ \rho_1 = \rho, \tilde{\tau} \circ \tau_1 = \tau, g_1 = g \circ \tilde{\rho}$. Let $h = g_1 \circ \rho_1 = g \circ \rho$. We will only show that $\tilde{F}_1 \cong \tilde{F}$ since the remaining procedure is almost the same as in previous proofs. It suffices to prove that, for all $u, v \in V(F)$, $\rho_1(u) = \rho_1(v)$ iff

a) $h(u) = h(v)$, and

b) There exists a path $P$ in $T^s$ with endpoints $t_1, t_2 \in V(T)$ such that $u \in \beta_{T^s}(t_1), v \in \beta_{T^s}(t_2)$, and all node $x$ on path $P$ satisfies that $h(u) \in h(\beta_{T^s}(x))$.

We begin by showing the first direction, i.e., $\rho_1(u) = \rho_1(v)$ implies Items a) and b). If $\rho_1(u) = \rho_1(v)$, we clearly have $h(u) = h(v)$ as $g_1$ is well-defined. Also, there exists $x_1 \in B_{T^s}(u), x_2 \in B_{T^s}(v)$, i.e., $u \in \beta_{T^s}(x_1)$ and $v \in \beta_{T^s}(x_2)$. Hence, $\rho(u) \in \rho(\beta_{T^s}(x_1)) \subset \beta_{\tilde{T}_1^{t_1}}(\tau_1(x_1))$ and $\rho_1(u) = \rho_1(v) \in \rho_1(\beta_{T^s}(x_2)) \subset \beta_{\tilde{T}_1^{t_1}}(\tau_1(x_2))$ since $(\rho_1, \tau_1)$ is a homomorphism. Hence, $\tau_1(x_1), \tau_1(x_2) \in B_{\tilde{T}_1^{t_1}}(\rho_1(u))$. Since $\tilde{T}_1^{t_1}[B_{\tilde{T}_1^{t_1}}(\rho_1(u))]$ is connected, there is a path $P$ in $\tilde{T}_1^{t_1}[B_{\tilde{T}_1^{t_1}}(\rho_1(u))]$ with endpoints $\tau_1(x_1), \tau_1(x_2)$ such that all nodes $x$ on $P$ satisfies $\rho_1(u) \in \beta_{\tilde{T}_1^{t_1}}(x) = \beta_{\tilde{T}_1^{t_1}}(\tau \circ \tau^{-1}(x)) = \rho_1\left(\beta_{T^s}(\tau_1^{-1}(x))\right)$. We conclude $h(u) = g_1(\rho_1(u)) \in g_1(\rho_1(\beta_{T^s}(\tau_1^{-1}(x)))) = h(\beta_{T^s}(\tau_1^{-1}(x)))$.

We continue by showing the second direction, i.e., $\rho_1(u) = \rho_1(v)$ if Items a) and b). We prove this by contradiction, i.e., assume $\rho_1(u) \neq \rho_1(v)$ but the above items (a) and (b) hold. We consider two cases. First, assume that $u$ and $v$ are in the same bag of $T^s$. Then, as $(\rho_1, \tau_1)$ is a homomorphism, the nodes $\rho_1(u)$ and $\rho_1(v)$ are in the same bag of $\tilde{T}_1^{t_1}$. Since $g_1$ is a bag isomorphism, we have $g_1(\rho_1(u)) \neq g_1(\rho_1(v))$. This contradicts Item (a) above.

Now, consider the second case. For this, assume that $u$ and $v$ are not in the same bag of $T^s$. Then, there exist two adjacent nodes $x_1, x_2$ on path $P$ such that $u \in \beta_{T^s}(x_1), u \notin \beta_{T^s}(x_2)$. We have $\beta_{T^s}(x_2) \subset \beta_{T^s}(x_1)$ as for every pair of nodes $t_1, t_2$ in a canonical tree decomposition with $\{t_1, t_2\} \in E(T^s)$ we have either $\beta_{T^s}(t_1) \subset \beta_{T^s}(t_2)$ or $\beta_{T^s}(t_2) \subset \beta_{T^s}(t_1)$. Now, item (b) implies that there exists $w \in \beta_{T^s}(x_2)$ such that $w \neq u$ and $h(w) = h(u)$. Then, $\rho_1(w) \in \rho_1\left(\beta_{T^s}(x_2)\right) \subset \rho_1\left(\beta_{T^s}(x_1)\right) \subset \beta_{\tilde{T}_1^{t_1}}(\tau_1(x_1))$. Therefore, $\rho_1(u)$ and $\rho_1(w)$ are two different nodes in $\beta_{\tilde{T}_1^{t_1}}(\tau_1(x_1))$ with $g_1(\rho_1(u)) = h(u) = h(w) = g_1(\rho_1(w))$. This contradicts the condition that $g_1$ is a bag isomorphism. This yields the desired result that $\tilde{F} \cong \tilde{F}_1$. □

Finally, we restate Theorem 2, with its proof now being a straightforward corollary of the preceding results in this section.

**Theorem 2.** *Let $r \geq 0$. Then, $r$-$\ell$WL can homomorphism-count $\mathcal{M}^{r+2}$.*

*Proof.* According to Corollary 3, if $c_r(G) = c_r(H)$, then $\mathrm{cnt}(F, G) = \mathrm{cnt}(F, H)$ for every $F \in \mathcal{M}^{r+2}$. Utilizing Lemma 6, we extend this result to bag isomorphism counts: $\mathrm{bIso}(F, G) = \mathrm{bIso}(F, H)$ holds for every $F \in \mathcal{M}^{r+2}$. Finally, invoking Lemma 7 and Lemma 8, we conclude that $\mathrm{hom}(F, G) = \mathrm{hom}(F, H)$ for all $F \in \mathcal{M}^{r+2}$. □

# H Implications of Theorem 2

In this section, we discuss important implications of Theorem 2 and provide proofs for the results in Corollary 2.

## H.1 Appendix on $\mathcal{F}$-Hom-GNNs and Proof of Corollary 2 i)

Recent work in the domain of MPNNs has explored enhancing the initial node features by incorporating homomorphism counts (Barceló et al., 2021). We summarize this approach in this section and compare it to our $r$-$\ell$WL algorithm.

Define $\mathcal{F} = \{P_1^s, \ldots, P_l^s\}$ as a collection of rooted graphs, termed as patterns. In $\mathcal{F}$-Hom-MPNNs, the initial feature vector of a vertex $v$ in a graph $G$ combines a one-hot encoding of the label $\chi_G(v)$ with homomorphism counts corresponding to each pattern in $\mathcal{F}$. The feature vector for each vertex $v$ is recursively defined over rounds of message passing as follows:

$$
\begin{aligned}
\mathbf{x}_{\mathcal{F},G,v}^{(0)} &= (\chi_G(v), \mathrm{hom}(P_1^s, G^v), \ldots, \mathrm{hom}(P_l^s, G^v)) \\
\mathbf{x}_{\mathcal{F},G,v}^{(t+1)} &= g^{(t+1)}\left(\mathbf{x}_{\mathcal{F},G,v}^{(t)}, f^{(t+1)}\left(\mathbf{x}_{\mathcal{F},G,u}^{(t)} \mid u \in N_G(v)\right)\right)
\end{aligned}
\tag{17}
$$

Here, $g^{(t)}$ and $f^{(t)}$ represent the update and aggregation functions at depth $t$, respectively.

### H.1.1 Expressivity of $\mathcal{F}$-Hom-MPNNs

In this section, we summarize known results about the expressivity of $\mathcal{F}$-Hom-MPNNs. The main result from Barceló et al., 2021 can be summarized as follows.

**Theorem 6.** *For any two graphs $G$ and $H$, it holds $\mathcal{F}$-Hom-MPNNs can separate $G$ and $H$ if and only if $hom(T, G) = hom(T, H)$, for every $\mathcal{F}$-pattern tree.*

To understand the above theorem, we need to define the concept of $\mathcal{F}$-pattern trees. For this, we define the graph join operator $*$ as follows. Given two rooted graphs $G^v$ and $H^w$, the join graph $(G * H)^v$ is obtained by taking the disjoint union of $G^v$ and $H^w$, followed by identifying $w$ with $v$. The root of the join graph is $v$. Further, if $G$ is a graph and $P^r$ is a rooted graph, then joining a vertex $v$ in $G$ with $P^r$ results in the disjoint union of $G$ and $P^r$, where $r$ is identified with $v$.

Let $\mathcal{F} = \{P_1, \ldots, P_l\}$. An $\mathcal{F}$-pattern tree $T^r$ is constructed from a standard rooted tree $S^r = (V, E, \chi)$, which serves as the core structure, or the "backbone", of $T^r$. To form $T^r$, each vertex $s \in V$ of the backbone may be joined to any number of duplicates of any patterns from $\mathcal{F}$. Conceptually, an $\mathcal{F}$-pattern tree is a tree graph enhanced by attaching multiple instances of any pattern from $\mathcal{F}$ to the nodes of the backbone tree. *However, it is important to note that additional patterns may not be attached to any node that already derives from a pattern in $\mathcal{F}$. Our method can homomorphism-count graphs where this is allowed, see Appendix H.1.2.*

Examples of $\mathcal{F}$-pattern trees for $\mathcal{F} = \{\!\!\begin{smallmatrix}\triangle\end{smallmatrix}\!\!\}$ are

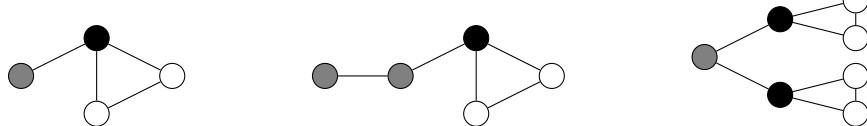

where grey vertices are part of the backbones of the $\mathcal{F}$-pattern trees, black vertices are the joined node and white vertices are part of the attached patterns. We define the set of $\mathcal{F}$-pattern trees by $\mathcal{F}^{\text{Tr}}$.

### H.1.2 Comparison with $r$-$\ell$WL

We compare our proposed $r$-$\ell$GIN against $\mathcal{F}^r$-Hom-MPNNs, where $\mathcal{F}^r = \{C_3, \ldots, C_{r+2}\}$ consists of cycle graphs up to length $r + 2$. Both MPNN variants exhibit equivalent preprocessing complexity. However, after the initial layer, the computational complexity of our method is marginally higher, yet it increases linearly with the number of cycles present in the underlying graph.

According to Theorem 2, our method $r$-$\ell$GIN can homomorphism-count all fan $(r + 2)$-cactus graphs. In particular, $r$-$\ell$GIN can homomorphism-count all $\mathcal{F}^r$-pattern trees. For example, there are infinitely many fan $r$-cactus graphs that cannot be represented as $\mathcal{F}^r$-pattern trees, e.g., for $r = 1$

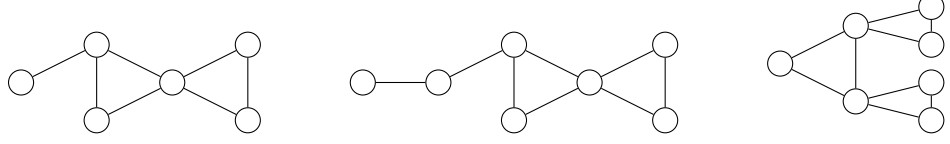

We restate Corollary 2 ii) and give a short proof.

**Corollary 4.** *Let $r \in \mathbb{N} \setminus \{0\}$. Then, $r$-$\ell$WL is more powerful than $\mathcal{F}$-Hom-MPNNs, where $\mathcal{F} = \{C_3, \ldots, C_{r+2}\}$.*

*Proof.* The proof of Corollary 2 ii) can be stated as a summary of all finding of the previous subsection: By Theorem 2, we have $r$-$\ell$WL $\sqsubseteq$ hom$(\mathcal{M}^{r+2}, \cdot)$. By Theorem 6, we have $\mathcal{F}$-Hom-MPNNs $\sqsubseteq$ hom$(\mathcal{F}^{\text{Tr}}, \cdot)$ and hom$(\mathcal{F}^{\text{Tr}}, \cdot) \sqsubseteq \mathcal{F}$-Hom-MPNNs. Clearly, $\mathcal{F}^{\text{Tr}} \subset \mathcal{M}^{r+2}$. Hence, $r$-$\ell$WL $\sqsubseteq$ hom$(\mathcal{M}^{r+2}, \cdot) \sqsubseteq$ hom$(\mathcal{F}^{\text{Tr}}, \cdot)$. Hence, $r$-$\ell$WL is more powerful than $\mathcal{F}$-Hom-MPNNs. $\qquad\square$

### H.2 Appendix on Subgraph GNNs and Proof of Corollary 2 ii)

Subgraph GNNs treat a graph as a collection of graphs $\{G^u \mid u \in N(v)\}$, where $G^u$ is a graph obtained by marking the corresponding node $u$. For every graph $G^u$ it runs an independent WL-

algorithm, i.e.,

$$\mathbf{x}_{\mathrm{Sub},G^u}^{(0)}(v) = (\chi_G(v), \mathbb{1}_{v=u}(v))$$

$$\mathbf{x}_{\mathrm{Sub},G^u}^{(t+1)}(v) = g^{(t+1)}\left(\mathbf{x}_{\mathrm{Sub},G^u}^{(t)}(v), f^{(t+1)}\left(\mathbf{x}_{\mathrm{Sub},G^u}^{(t)}(w) \mid w \in N_G(v)\right)\right). \tag{18}$$

Here, $g^{(t)}$ and $f^{(t)}$ represent the update and aggregation functions at depth $t$, respectively. The final node representations after $t$ rounds are then calculated by

$$\mathbf{x}_{\mathrm{Sub},G}^t(u) = h\left(\mathbf{x}_{\mathrm{Sub},G^u}^{(t)}(v) \mid v \in V(G)\right).$$

### H.2.1 Expressivity of Subgraph GNNs and Comparison with $r$-$\ell$WL

The expressivity of subgraph GNNs is fully characterized by the class

$$\mathcal{F}^{\mathrm{sub}} := \{F \mid \exists u \in V(F) \text{ s.t. } F \setminus \{u\} \text{ is a forest}\},$$

i.e., Subgraph GNNs can separate a pair of graphs $G, H$ if and only if $\hom(\mathcal{F}^{\mathrm{sub}}, G) \neq \hom(\mathcal{F}^{\mathrm{sub}}, H)$. Furthermore, the set $\mathcal{F}^{\mathrm{sub}}$ is the maximal set that satisfies this property (B. Zhang et al., 2024, Theorem 3.4). We restate Corollary 2 ii) and provide a proof.

**Corollary 5.** *1-$\ell$WL is not less powerful than Subgraph GNNs. In particular, any $r$-$\ell$WL can separate infinitely many graphs that Subgraph GNNs fail to distinguish.*

*Proof.* We show that already 1-$\ell$WL can separate infinitely many graphs that Subgraph GNNs fail to distinguish. The other statements then follow as a simple corollary of Proposition 1.

For clarity, we begin by demonstrating that there exists a pair of graphs that 1-$\ell$GIN can separate, but Subgraph GNNs cannot distinguish. Consider the graph $F$ defined as follows: $F = \{\ \triangleright\!\!\circ\!\!-\!\!\circ\!\!\triangleleft\ \}$. It holds that $F \in \mathcal{M}^3 \setminus \mathcal{F}^{\mathrm{sub}}$, where $\mathcal{F}^{\mathrm{sub}}$ is the maximal set that Subgraph GNNs can homomorphism-count. Then, by (B. Zhang et al., 2024, Theorem 3.4), there exists a pair of graphs $G(F)$ and $H(F)$ such that $\hom(F, G(F)) \neq \hom(F, H(F))$ and $\hom(\mathcal{F}^{\mathrm{sub}}, G(F)) = \hom(\mathcal{F}^{\mathrm{sub}}, H(F))$. Hence, Subgraph GNNs cannot separate $G(F)$ and $H(F)$. Since $F \in \mathcal{M}^3$, by $\hom(F, G(F)) \neq \hom(F, H(F))$ and Theorem 2, 1-$\ell$WL can separate $G(F)$ and $H(F)$.

This argument can be repeated for every $F \in \mathcal{M}^3 \setminus \mathcal{F}^{\mathrm{sub}}$. Since there are infinitely many graphs in $\mathcal{M}^3 \setminus \mathcal{F}^{\mathrm{sub}}$, the corollary follows. $\square$

We mention that the construction of the pair of graphs $G(F)$ and $H(F)$ in the previous proof is based on (twisted) Fürer graphs and is largely motivated by the constructions by Fürer (2001) and B. Zhang et al. (2024). More precisely, we can define $G(F)$ as the Fürer graph of $F$ and $H(F)$ as the corresponding twisted Fürer graph. See Figure 7c for a visualization of $F, G(F),$ and $H(F)$.

### H.3 Appendix on Subgraph $k$-GNNs and Proof of Corollary 2 iii)

Qian et al. (2022) introduced a higher-order version of Subgraph GNNs that compute representations for subgraphs made of tuples of nodes. Specifically, Subgraph $k$-GNNs – referred to as vertex-subgraph $k$-OSANs in the original work (Qian et al., 2022) – treat a graph as a collection of graphs $\{G^{\mathbf{u}} \mid \mathbf{u} \in V(G)^k\}$, where $G^{\mathbf{u}}$ is a graph obtained by marking the corresponding nodes $\mathbf{u}$. For every graph $G^{\mathbf{u}}$ it runs an independent WL-algorithm, i.e.,

$$\mathbf{x}_{\mathrm{Sub}(k),G^{\mathbf{u}}}^{(0)}(v) = (\chi_G(v), \mathrm{atp}(\mathbf{u}), \mathbb{1}_{v=u_1}(v), \ldots, \mathbb{1}_{v=u_k}(v))$$

$$\mathbf{x}_{\mathrm{Sub}(k),G^{\mathbf{u}}}^{(t+1)}(v) = g^{(t+1)}\left(\mathbf{x}_{\mathrm{Sub}(k),G^{\mathbf{u}}}^{(t)}(v), f^{(t+1)}\left(\mathbf{x}_{\mathrm{Sub}(k),G^{\mathbf{u}}}^{(t)}(w) \mid w \in N_G(v)\right)\right). \tag{19}$$

Here, $g^{(t)}$ and $f^{(t)}$ represent the update and aggregation functions at depth $t$, respectively. For every $k$-tuple $\mathbf{u}$, the final representations after $t$ rounds are then calculated by

$$\mathbf{x}_{\mathrm{Sub}(k),G}^{(t)}(\mathbf{u}) = h\left(\mathbf{x}_{\mathrm{Sub}(k),G^{\mathbf{u}}}^{(t)}(v) \mid v \in V(G)\right).$$

The final graph representation is then given by

$$\mathbf{x}_{\mathrm{Sub}(k)}^{(t)}(G) = j\left(\mathbf{x}_{\mathrm{Sub}(k),G}^{(t)}(\mathbf{u}) \mid \mathbf{u} \in V(G)^k\right).$$

The homomorphism-expressivity of these GNNs is characterized as follows:

**Theorem 7** (B. Zhang et al., 2024). *The homomorphism-expressivity of Subgraph $k$-GNN is given by $\mathcal{F}^{\mathrm{sub}(k)} = \{F : \exists U \subset V_F \text{ s.t. } |U| \leq k \text{ and } F \setminus U \text{ is a forest }\}$.*

Since the set $\mathcal{F}^{\mathrm{sub}(k)}$ is the *maximal set* of graphs that Subgraph $k$-GNNs can homomorphism-count, we can derive the following corollary, which restates Corollary 2 iii) and provides a proof.

**Corollary 6.** *For any $k \geq 1$, 1-$\ell$WL is not less powerful than Subgraph $k$-GNNs. In particular, any $r$-$\ell$WL can separate infinitely many graphs that Subgraph $k$-GNNs fail to distinguish.*

*Proof.* The proof parallels the proof of Corollary 2. We present it here for completeness.

Let $k \geq 1$. We will show that there exists a pair of graphs that 1-$\ell$GIN can distinguish, but Subgraph $k$-GNNs cannot. Consider the graph $F$, defined as the unique graph with $k$ triangle graphs, all connected by an edge to a single node.

It holds that $F \in \mathcal{M}^3 \setminus \mathcal{F}^{\mathrm{sub}(k)}$, where $\mathcal{F}^{\mathrm{sub}(k)}$ is the maximal set that Subgraph $k$-GNNs can homomorphism-count. Then, by (B. Zhang et al., 2024, Theorem 3.8), there exists a pair of graphs $G(F)$ and $H(F)$ such that $\mathrm{hom}(F, G(F)) \neq \mathrm{hom}(F, H(F))$ and $\mathrm{hom}(\mathcal{F}^{\mathrm{sub}(k)}, G(F)) = \mathrm{hom}(\mathcal{F}^{\mathrm{sub}(k)}, H(F))$. Hence, Subgraph $k$-GNNs cannot separate $G(F)$ and $H(F)$. Since $F \in \mathcal{M}^3$, by $\mathrm{hom}(F, G(F)) \neq \mathrm{hom}(F, H(F))$ and Theorem 2, 1-$\ell$WL can separate $G(F)$ and $H(F)$.

This argument can be repeated for every $F \in \mathcal{M}^3 \setminus \mathcal{F}^{\mathrm{sub}(k)}$. Since there are infinitely many non-isomorphic graphs in $\mathcal{M}^3 \setminus \mathcal{F}^{\mathrm{sub}(k)}$, the corollary follows. $\square$

## H.4 Proof of Corollary 2 iv)

*Proof of Corollary 2 iv).* Given graphs $F$ and $G$, it is well-known (see, e.g., (Neuen, 2024; Curticapean et al., 2017)) that $\mathrm{sub}(F, G)$ can be decomposed as:

$$\mathrm{sub}(F, G) = \sum_{F' \in \mathrm{spasm}(F)/\sim} \alpha(F')\mathrm{hom}(F', G). \tag{20}$$

Here, the sum ranges over all non-isomorphic graphs in $\mathrm{spasm}(F)$. The sum in (20) is finite since the homomorphic image of $F$ has at most $|V(F)|$ nodes. Per assumption, we have $\mathrm{spasm}(F) \subset \mathcal{M}^{r+2}$, i.e., by Theorem 2, $r$-$\ell$WL can homomorphism-count $\mathrm{spasm}(F)$. In particular, if $r$-$\ell$WL cannot separate two graphs $G$ and $H$, we have $\mathrm{hom}(\mathrm{spasm}(F), G) = \mathrm{hom}(\mathrm{spasm}(F), H)$, and hence, $\mathrm{sub}(F, G) = \mathrm{sub}(F, H)$.

The result on subgraph-counting paths follows directly as the homomorphic image of a path $P_{r+3}$ of length $r + 3$ lies in $\mathcal{M}^r$. $\square$

# I Appendix for Section 6

**Theorem 3.** *For fixed $t, r \geq 0$, $t$ iterations of $r$-$\ell$WL are more powerful than $r$-$\ell$MPNN with $t$ layers. Conversely, $r$-$\ell$MPNN is more powerful than $r$-$\ell$WL if the functions $f^{(t)}, g^{(t)}$ in (3) are injective.*

*Proof of Theorem 3.* We begin by proving that $c_r^{(t)} \sqsubseteq h_r^{(t)}$. We argue by induction over $t$ for any fixed $r \geq 0$.

Initially, $c_r^{(0)} = h_r^{(0)}$ as both labeling functions start with the same base labels. Now assume $c_r^{(t+1)}(u) = c_r^{(t+1)}(v)$ for some $u, v \in V(G)$. By definition,

$$\mathrm{HASH}\left(c_r^{(t)}(u), \left\{\left\{c_r^{(t)}(\mathbf{p}) \,|\, \mathbf{p} \in \mathcal{N}_0(u)\right\}\right\}, \ldots\right) = \mathrm{HASH}\left(c_r^{(t)}(v), \left\{\left\{c_r^{(t)}(\mathbf{p}) \,|\, \mathbf{p} \in \mathcal{N}_0(v)\right\}\right\}, \ldots\right).$$

This implies $c_r^{(t)}(u) = c_r^{(t)}(v)$ and

$$\left\{\left\{c_r^{(t)}(\mathbf{p}) \,|\, \mathbf{p} \in \mathcal{N}_k(u)\right\}\right\} = \left\{\left\{c_r^{(t)}(\mathbf{p}) \,|\, \mathbf{p} \in \mathcal{N}_k(v)\right\}\right\}, \quad \forall k \in \{0, \ldots, r\},$$

as HASH is an injective function.

By induction hypothesis, we hence have $h_r^{(t)}(u) = h_r^{(t)}(v)$ and

$$\left\{\left\{h_r^{(t)}(\mathbf{p}) \,|\, \mathbf{p} \in \mathcal{N}_k(u)\right\}\right\} = \left\{\left\{h_r^{(t)}(\mathbf{p}) \,|\, \mathbf{p} \in \mathcal{N}_k(v)\right\}\right\}, \quad \forall k \in \{0, \ldots, r\},$$

which implies that any function, in particular $f_k^{(t+1)}$ and $g^{(t+1)}$ have to return the same result. Therefore, we have $h_r^{(t+1)}(u) = h_r^{(t+1)}(v)$.

We proceed to prove $h_r^{(t)} \sqsubseteq c_r^{(t)}$ if all message, update, and readout functions are injective in Definition 9. For this, we show that for each $t \geq 0$ there exists an injective function $\phi$ such that $h_r^{(t)} = \phi \circ c_r^{(t)}$. For $t = 0$, we can choose $\phi$ to be the identity function. Assume that for $t - 1$ there exists an injective function $\phi$ such that $h_r^{(t-1)}(v) = \phi \circ c_r^{(t-1)}(v)$. Then, we can write

$$h_r^{(t)}(v) = g^{(t)}\left(h^{(t-1)}(v), m_0^{(t)}(v), \ldots, m_r^{(t)}(v)\right)$$

$$= g^{(t)}\left(\phi \circ c_r^{(t-1)}(v), \phi \circ m_0^{(t)}(v), \ldots, \phi \circ m_r^{(t)}(v)\right),$$

where for every $q = 0, \ldots, r$, we set $\phi \circ m_q^{(t)}(v) := \left\{\left\{(\phi \circ c_r^{(t-1)}(\mathbf{p})) \mid \mathbf{p} \in \mathcal{N}_q(v)\right\}\right\}$ and $(\phi \circ c_r^{(t-1)}(\mathbf{p})) = (\phi \circ c_r^{(t-1)}(p_1), \ldots, \phi \circ c_r^{(t-1)}(p_{q+1}))$ for $\mathbf{p} = \{p_i\}_{i=1}^{q+1} \in \mathcal{N}_q(v)$. By assumption, all message, update, and readout functions are injective in Definition 9. Since the concatenation of injective functions is injective, there exists an injective function $\psi$ such that

$$h_r^{(t)}(v) = \psi\left(c_r^{(t-1)}(v), \left\{\left\{c_r^{(t-1)}(\mathbf{p}) \mid \mathbf{p} \in \mathcal{N}_0(v)\right\}\right\},\right.$$

$$\left\{\left\{c_r^{(t-1)}(\mathbf{p}) \mid \mathbf{p} \in \mathcal{N}_1(v)\right\}\right\},$$

$$\vdots$$

$$\left.\left\{\left\{c_r^{(t-1)}(\mathbf{p}) \mid \mathbf{p} \in \mathcal{N}_r(v)\right\}\right\}\right).$$

As HASH in Definition 7 is injective, the inverse $\text{HASH}^{-1}$ exists and is also injective. Hence,

$$h_r^{(t)}(v) = \psi \circ \text{HASH}^{-1} \circ \text{HASH}\left(c_r^{(t-1)}(v), \left\{\left\{c_r^{(t-1)}(\mathbf{p}) \mid p \in \mathcal{N}_0(v)\right\}\right\},\right.$$

$$\left\{\left\{(c_r^{(t-1)}(\mathbf{p}) \mid \mathbf{p} \in \mathcal{N}_1(v)\right\}\right\},$$

$$\vdots$$

$$\left.\left\{\left\{(c_r^{(t-1)}(\mathbf{p}) \mid \mathbf{p} \in \mathcal{N}_r(v)\right\}\right\}\right)$$

$$= \psi \circ \text{HASH}^{-1}\left(c_r^{(t)}(v)\right).$$

Choosing $\phi = \psi \circ \text{HASH}^{-1}$ finishes the proof. $\qquad\square$

We conclude this section with the following lemma that justifies our architectural choice in (4).

**Lemma 9.** *Let $x \in \mathbb{Q}^r$. Then there exist $\varepsilon \in \mathbb{R}^r$ such that*

$$\varphi(x) = \sum_{k=0}^{r} \varepsilon_k x_k \tag{21}$$

*is an injective function.*

*Proof.* We prove this claim by induction. For $r = 0$, any $x \neq 0 \in \mathbb{R}$ fulfills the claim. Now, let $\varepsilon \in \mathbb{R}^r$ such that $\varphi(x) : \mathbb{Q}^r \to \mathbb{R}$ is injective. The set $\mathbb{Q}[\varepsilon_1, \ldots, \varepsilon_r] = \{\sum_{k=0}^{r} \varepsilon_k x_k \mid x \in \mathbb{Q}^r\}$ is

countable and hence a proper subset of $\mathbb{R}$. It follows that there exists $\varepsilon_{r+1} \in \mathbb{R}$ with $\varepsilon_{r+1} \notin \mathbb{Q}[\varepsilon]$. Note that $0 \in \mathbb{Q}$ and hence $\varepsilon_{r+1} \neq 0$. We now prove our claim by contradiction.

Assume there exist $x \neq x' \in \mathbb{Q}^{r+1}$ with $\sum_{k=0}^{r+1} \varepsilon_k x_k = \sum_{k=0}^{r+1} \varepsilon_k x'_k$. We distinguish two cases:

$x_i = x'_i$ for all $i \leq r$ and $x_{r+1} \neq x'_{r+1}$: Then immediately

$$
\begin{aligned}
& x_{r+1} \neq x'_{r+1} \\
\Rightarrow \quad & \varepsilon_{r+1} x_{r+1} \neq \varepsilon_{r+1} x'_{r+1} \\
\Rightarrow \quad & \sum_{k=0}^{r} \varepsilon_k x_k + \varepsilon_{r+1} x_{r+1} \neq \sum_{k=0}^{r} \varepsilon_k x_k + \varepsilon_{r+1} x'_{r+1} \\
\Rightarrow \quad & \sum_{k=0}^{r+1} \varepsilon_k x_k \neq \sum_{k=0}^{r+1} \varepsilon_k x_k \,.
\end{aligned}
$$

$\sum_{k=0}^{r} \varepsilon_k x_k \neq \sum_{k=0}^{r} \varepsilon_k x'_k$: But then

$$
\begin{aligned}
& \sum_{k=0}^{r} \varepsilon_k x_k + \varepsilon_{r+1} x_{r+1} = \sum_{k=0}^{r} \varepsilon_k x'_k + \varepsilon_{r+1} x'_{r+1} \\
\Leftrightarrow \quad & \sum_{k=0}^{r} \varepsilon_k x_k - \sum_{k=0}^{r} \varepsilon_k x'_k = \varepsilon_{r+1} (x'_{r+1} - x_{r+1})
\end{aligned}
$$

The left hand side is an element of $\mathbb{Q}[\varepsilon_1, \ldots, \varepsilon_r]$. However, $\varepsilon_{r+1}(x'_{r+1} - x_{r+1}) \notin \mathbb{Q}[\varepsilon_1, \ldots, \varepsilon_r]$ by choice of $\varepsilon_{r+1}$, leading to a contradiction. $\square$

