# OpenReview forum: "Weisfeiler and Leman Go Loopy: A New Hierarchy for Graph Representational Learning"
_NeurIPS.cc/2024/Conference — NeurIPS 2024 oral_

### Official Review · Reviewer_1SHr · 2024-06-22

**Soundness:** 3
**Presentation:** 4
**Contribution:** 3
**Rating:** 8
**Confidence:** 4

**Summary:**

The paper introduces the $r$-loopy Weisfeiler-Leman ($r$-$l$WL) test, an innovative hierarchy of graph isomorphism tests, and the corresponding GNN framework, $r$-$l$MPNN. This new approach extends the counting capabilities of previous algorithms, specifically allowing the counting of cycles up to length $r+2$ and homomorphisms of cactus graphs. Empirical validation demonstrates the expressiveness and performance of $r$-$l$MPNN on both synthetic and real-world datasets.

**Strengths:**

The paper's strengths are rooted in its originality, introducing a novel algorithm ($r$-$l$WL) and corresponding GNNs ($r$-$l$GIN) that significantly enhance the expressivity of graph neural networks. These contributions are supported by rigorous theoretical proofs and empirical validation. Specifically, $r$-$l$WL demonstrates the ability to count cycles up to length $r+2$ and homomorphisms of cactus graphs, substantiated with detailed mathematical proofs. The experiments use several synthetic datasets to validate the counting power and expressiveness of $r$-$l$MPNN effectively. Furthermore, the paper contextualizes its contributions within prior work, highlighting the limitations of existing methods and demonstrating how $r$-$l$WL and $r$-$l$MPNN address these gaps. Overall, the claims are well-supported by theoretical proofs and empirical results, indicating a clear improvement over existing methods.

**Weaknesses:**

Some of the mathematical proofs are complex and may be difficult for readers without a strong background in graph theory and GNNs. Providing additional intuitive explanations or examples could improve accessibility. While the empirical validation is strong, it could be expanded to include a broader range of real-world datasets to further demonstrate the robustness and generalizability of the approach.

**Questions:**

The paper is generally well-written and clear, although some sections could benefit from additional explanations or examples to aid understanding. I have a few questions:

1. How does the computational complexity of $r$-$l$WL compared to existing higher-order WL variants like $3$-WL in practice, especially when dealing with large and dense graphs?

2. What are the limitations of $r$-$l$WL in terms of scalability and memory usage, particularly when applied to real-world datasets with varying degrees of sparsity?

3. In Table 1, why didn't your method perform well on the Extension (100) and CFI (100) datasets compared to 3-WL and PPGN?

**Limitations:**

The authors have addressed the limitations related to the complexity of higher-order GNNs and the scalability issues associated with $k$-WL. However, a more detailed discussion on the limitations of $r$-$l$WL in terms of computational overhead and potential impact on large-scale applications would be beneficial.

---

> ### Author Rebuttal · Authors · 2024-08-06
>
> We thank the reviewer for their thorough review, for acknowledging the originality and rigor of our paper, and for voting to accept. We address each point individually below. “W/Q” numbers the weakness or question, followed by our response.
>
> ---
> > **W1**: “Some of the mathematical proofs are complex and may be difficult for readers without a strong background in graph theory and GNNs. Providing additional intuitive explanations or examples could improve accessibility.”
>
> **A1**: We agree that some of the mathematical proofs are difficult for readers without a strong background in graph theory and GNNs. To improve accessibility, we have included visualizations of some proofs and counterexamples in the appendix, see, e.g., Figure 10,11,12 or Section H. Following Reviewer 1sHr’s suggestion, we add additional intuitive explanations before every proof in the respective section. Thank you for the suggestion!
>
> > **W2**: “While the empirical validation is strong, it could be expanded to include a broader range of real-world datasets to further demonstrate the robustness and generalizability of the approach.”
>
> A2:  Thanks for recognizing our experiments! We expand the experiments by including the peptides-functional and peptides-struct datasets from the LRGB paper [1]. Our first preliminary results without any sweeping or positional encodings suggest improved performance over standard baselines:
>
> |  | Peptides Structural (MAE $\downarrow$) | Peptides Functional (AP $\uparrow$) |
> |---------------|-----------------|---------------------|
> | GCN	| 0.3496 ± 0.0013 | 59.30 ± 0.23 |
> | GINE	| 0.3547 ± 0.004 | 54.98 ± 0.79 |
> | GatedGCN	| 0.3420 ± 0.0013 | 58.64 ± 0.77 |
> | $7$-$\ell{}$GIN	| 0.2513 ± 0.0021 | 65.70 ± 0.60 |
>
> [1] Vijay Prakash Dwivedi, Ladislav Rampásek, Michael Galkin, Ali Parviz, Guy Wolf, Anh Tuan Luu, Dominique Beaini: Long Range Graph Benchmark. NeurIPS 2022.
>
> > **Q1**: “How does the computational complexity of $r$-$\ell{}$WL compared to existing higher-order $k$-WL variants like $3$-WL in practice, especially when dealing with large and dense graphs?
>
> **A1**: When dealing with dense graphs, i.e., graphs with $n^2$ edges the computational complexity of $r$-$\ell{}$WL is comparable to the complexity of $k$-WL. In the worst-case scenario, assuming a complete graph, every node has $n$ neighbors. Hence, the preprocessing step requires $n*n^{r+2}$ operations and the number of forward operations is also exponential in $r$. This is a limitation for large and dense graphs. We noted this in our manuscript, see Page 9 under Limitations.  From a practical perspective, our method currently scales well to the Peptides dataset, even when using $r=7$, which contains 15k graphs with on average 151 nodes and 307 edges.
>
> > **Q2**: “What are the limitations of $r$-$\ell{}$WL in terms of scalability and memory usage, particularly when applied to real-world datasets with varying degrees of sparsity?”
>
> **A2**: Our $r$-$\ell{}$WL algorithm is particularly well-suited for sparser graphs, as demonstrated in our experiments. This focus allows us to efficiently handle a wide range of real-world datasets that exhibit sparsity. While there are scalability and memory challenges when dealing with denser graphs, we see this as an exciting opportunity for future enhancements. To increase scalability and manage memory usage more effectively, one potential future direction includes random subsampling methods to subsample paths per node. We are actively researching ways to optimize our method to better manage dense graphs, thereby broadening the applicability of our algorithm across diverse datasets.
>
> > **Q3**: “In Table 1, why didn't your method perform well on the Extension (100) and CFI (100) datasets compared to 3-WL and PPGN?”
>
> **A3**: Wang et al. [1] note that 3-WL and PPGN “surpasses most GNNs in CFI graphs due to k-WL’s global receptive field.” Whereas local WL variants like our $r$-$\ell{}$WL exhibit lower performance due to their limited receptive fields.
> We also mention that CFI graphs are particularly challenging; specifically, 60 pairs are distinguishable only by 3-WL, 20 by 4-WL, and 20 remain indistinguishable even by 4-WL. To improve performance, we would need to increase $r$. However, this runs into memory issues given our current resources, as CFI graphs contain many paths in the $r$-neighborhoods, often having a high number of edges (up to 742).
>
> Our method successfully distinguished 95 out of 100 pairs of extension graphs, which aligns closely with the performance of the baseline models as detailed in Table 2. This demonstrates that our algorithm is competitive on the extension graphs.
> We are actively exploring ways to optimize our method to handle such challenging graphs more effectively in future work.
>
> [1] Wang, Yanbo, et al. "An Empirical Study of Realized GNN Expressiveness." International Conference on Machine Learning. PMLR, 2024.
>
> ---
>
> Thank you again for your valuable feedback, which we will incorporate into a potential camera-ready version, making our work even more accessible!

---

> > ### Comment · Reviewer_1SHr · 2024-08-10
> > **Satisfied with the answers**
> >
> > Thank you to the authors for addressing my comments and for their thoughtful responses to the other reviewers' feedback. I continue to believe this is a good paper and will maintain my score.

---

### Official Review · Reviewer_rfpS · 2024-07-09

**Soundness:** 3
**Presentation:** 3
**Contribution:** 3
**Rating:** 7
**Confidence:** 4

**Summary:**

The paper proposed a hierarchy of graph isomorphism tests and a corresponding GNN framework, $r$-$\ell$-MPNN while showing the ability to count homomorphisms of cactus graphs.

**Strengths:**

The strengths of the paper are:
* The ability to count homomorphisms of cactus graphs without any additional explicit substructure counts.
* Scalability towards large datasets, especially when the graphs are sparse for these datasets.
* The paper is well-written and easy to understand.

**Weaknesses:**

The weaknesses of the paper is that while the paper is well written, there are some terms undefined, or unexplained such as HASH; see the first question in the following section.

**Questions:**

1) What is the HASH function? do you have an example of such a function?
2) Can you add tables concerning the times of your method compared to other methods?
3)  What is the motivation behind using $r=5$ in the experiments? would any $r>5$ result in worse results or simply better results by small significance on the expense of might higher time?

**Limitations:**

The authors have adequately addressed the limitations.

---

> ### Author Rebuttal · Authors · 2024-08-06
>
> We thank the reviewer for their thorough review, acknowledging our contributions, and voting to accept our paper. We address each point individually below. “W/Q” numbers the weakness or question, followed by our response.
>
> ---
> > **W**: “The weaknesses of the paper is that while the paper is well written, there are some terms undefined, or unexplained such as HASH; see the first question in the following section.”
>
> **A**: Thank you for your feedback. We have carefully revised our manuscript to ensure that all important terms, including the HASH function (see below), are clearly defined.
>
> > **Q1**: “What is the HASH function? do you have an example of such a function?”
>
> **A1**: The term HASH function is standard in color refinement algorithms such as $r$-$\ell{}$WL. It refers to any arbitrary injective function on multisets. In practice, one can use SHA-256 of the sorted multiset as an example. For the neural variant, $r$-$\ell{}$GIN, the HASH function can be realized by summation followed by an MLP. We have added this clarification to the paper.
>
> > **Q2**: “Can you add tables concerning the times of your method compared to other methods?”
>
> **A2**: We have included tables comparing our methods to other state-of-the-art methods in terms of memory and runtime. Please refer to Table 9 for these comparisons. Please let us know if you are interested in any other specific comparison. We are happy to add more comparisons.
>
> > **Q3**: “What is the motivation behind using $r=5$ in the experiments? would any result in worse results or simply better results by small significance on the expense of might higher time?”
>
> **A3**: We chose $r=5$ because it matches the cycle-counting power of 3-WL. For the second question, please refer to Table 8, which compares predictive performance for ZINC12K when varying $r$. This table demonstrates that $r=5$ provides a good balance between accuracy and computational efficiency. Increasing $r$ enhances the representation power and complexity of the model, which can negatively impact generalization performance. Therefore, it is crucial to strike a balance between expressivity, computational cost, and generalization abilities.
> To address the Reviewer's question more concretely, we tested a $12$-$\ell{}$GIN on ZINC12K. This configuration resulted in a test MAE of $0.075\pm0.003$, which is within the standard deviation of the performance of a $5$-$\ell{}$GIN. The runtime increased by approximately 20%, which remains more efficient than other higher-order GNNs, such as those based on the $3$-WL test. We will include the results for $r=6,\ldots,12$ in Tables 8 and 9.
>
> ---
> Thank you again for your valuable suggestions, which have helped improve the clarity and contribution of our work. Please let us know if you have any more questions or suggestions!

---

> > ### Comment · Reviewer_rfpS · 2024-08-09
> > **Good rebuttal**
> >
> > The authors satisfactorily addressed my questions. However, I have kept my score, increasing my confidence to 4 in light of the authors' answers.

---

### Official Review · Reviewer_zkXv · 2024-07-12

**Soundness:** 3
**Presentation:** 3
**Contribution:** 3
**Rating:** 7
**Confidence:** 4

**Summary:**

In this paper, the authors propose a loopy version of the Weisfeiler-Lehman (WL) algorithm. This version utilizes an extended notion of neighborhood, incorporating paths between standard neighboring vertices to update vertex coloring. By parameterizing the length $r$ of these paths, we obtain $r$-$l$WL. The results include a hierarchy for $r$-$l$WL based on $r$ and a comparison with $k$-WL. The most technical result is that $r$-$l$WL is expressive enough to determine homomorphism counts of $(r+2)$-cactus graphs. Experiments show that $r$-$l$WL is competitive in detecting substructures.

**Strengths:**

**S1 Interesting Idea:**
The inclusion of paths in the neighborhood is an elegant and innovative way of extracting local information around a vertex. This approach allows for rigorous theoretical analysis and captures cactus graphs, which is a significant advantage.

**S2 Experiments:**
The authors empirically demonstrate that the neural version of $r$-$l$WL captures crucial information for graph learning tasks.

**S3 Capturing Cacti:** The most technically involved proof shows that cactus graphs, or at least their homomorphism counts, can be captured. This is a noteworthy result.

**Weaknesses:**

**W1 Capturing Cacti:**
The motivation for focusing on cactus graphs is unclear. The authors should better justify why this class of graphs is interesting and relevant, and provide examples in the main paper.

**W2 Insufficient Comparison with Subgraph GNNs:**
The paper mentions that subgraph GNNs are bounded by $3$-WL, but there are subgraph GNNs beyond $3$-WL, such as $k$-OSAN s (ordered subgraph aggregation networks, Qian et al.), which can capture features that $k$-WL cannot and are bounded by $(k+1)$-WL. The paper overlooks this line of work and lacks comparison with $k$-OSANs, both empirically and theoretically. For example, specifying paths of length $r$ and a central node requires special subgraphs of size $r+1$ (the path plus central node), suggesting that $r$-$l$WL may be included in $(r+1)$-OSAN? This could imply that some results stem from $k$-OSAN properties. A more detailed comparison is needed.

**W3 Experimental Comparison:**
The experiments do not seem to include any subgraph GNNs that capture properties beyond 3-WL. The authors should make more clear what are the capabilities of the methods with which they compare.

**W4 Unlabeled Graphs:**
The paper does not address vertex labels. Can the approach be generalized to vertex labels? Additionally, $c^{(0)}$ in section 3.2 is undefined.

Minor Comments: The authors sometimes use obscure references. For instance:
- line 71: Do you mean Tinhofer or Dvorak?
- line 96: Why refer to Dimitrov 2023 for the notion of graph invariant, which has existed for ages?

**Questions:**

Please comment on **W1**, **W2** and **W3**.

**Limitations:**

This has been addressed in a satisfactory way by the authors.

---

> ### Author Rebuttal · Authors · 2024-08-06
>
> We thank you for your thorough review and valuable suggestions. We include a theoretical and experimental comparison with $k$-OSAN in an updated manuscript and believe the points below address the Reviewers' questions adequately.
>
> ---
> > **W1**: “The motivation for focusing on cactus graphs is unclear…”
>
> **A1:**  Our primary focus was on developing a scalable and expressive method. Our analysis revealed a close connection to cactus graphs, a significant class between trees and tree-width 2 graphs. While the class of tree-width 2 graphs is larger, no less than cubic time GNN is known that can count all tree-width 2 graphs. Hence, our method can provably capture a smaller class than 3-WL, but is more scalable and local. This is a trade-off we make. Note that this shows that our method, while being local, is not less expressive than other non-local variants, e.g., k-OSAN. See the next answer.
>
> Moreover, many chemical datasets contain cactus graphs; $58.77\\%$ of ZINC250K are cactus graphs. Consider, for instance, rings with hydrogen atoms attached or any other structure (e.g., carboxyl groups). The presence of such structures, while being cactus graphs and not mere cycle graphs, can significantly alter the molecular properties of a more complex graph. The practical significance of being able to count cactus graphs is also shown by the improved predictive performance of our model when applied to chemical datasets.
>
> We are happy to follow the Reviewer's suggestion and include this discussion on the relevance of cactus graphs in our manuscript.
>
> > **W2**: “Insufficient Comparison with Subgraph GNNs: …”
>
> **A2**: Thank you for highlighting this important direction. We will address this in detail in a potential camera-ready version, referencing Qian et al. (2022). They introduce k-OSAN and vertex-selected k-OSAN (k-VSAN), both bounded by $k+1$-WL but incomparable to $k$-WL. We can make the following simple observation: For every $k$, there exists an $r$ such that $r$-$\ell{}$WL is not less powerful than $k$-OSAN and $k$-VSAN, following from the fact that both are less expressive than $(k+1)$-WL and Corollary 1 in our manuscript.
>
> Moreover, as a corollary of our Theorem 2, we show that for every $k \geq 1$, there exist infinitely many graphs that $1$-$\ell{}$WL can separate but $k$-VSAN cannot distinguish. This shows that the Reviewer’s conjecture that “r-$\ell{}$WL may be included in k-OSAN” does not hold.
>
> The proof of this result parallels the proof of Corollary 2 ii) in our manuscript: (B. Zhang et al., 2024) characterized the class of patterns $\mathcal{F}^{\mathrm{sub}(k)}$ that $k$-VSAN can homomorphism-count. There exist infinitely many cactus graphs in $\mathcal{M}^3$ that are not in $\mathcal{F}^{\mathrm{sub}(k)}$. A combination of Theorem 3.8 in (B. Zhang et al., 2024) and Theorem 2 in our manuscript shows that there are infinitely many graphs that $1$-$\ell{}$WL can separate but $k$-VSAN cannot distinguish. We will present this result in the main paper and a detailed proof in the appendix.
>
> This result demonstrates that even our least expressive algorithm is not less powerful than any $k$-VSAN algorithm, which may be of independent interest, given that the expressive power of $k$-VSAN increases with $k$, along with its computational complexity.
>
> Since there are pairs of graphs that $1$-$\ell{}$WL cannot distinguish but $k$-VSAN can, this proves that $1$-$\ell{}$WL and $k$-VSAN are incomparable for any $k$.
>
> > **W3**: “Experimental Comparison: …”
>
> **A3**: We are happy to update our manuscript and include OSAN in the BREC dataset-baseline. To go beyond $3$-WL with $k$-OSAN, we would need to consider at least $2$-OSAN, which has a complexity of $n^2m$, where $n$ is the number of nodes and $m$ the number of edges. This complexity is too high for our computational resources, which is why we did not use $2$-OSAN or higher-order variants as baselines, consistent with other research in this area.
>
> However, if the Reviewer can provide a reference with $2$-OSAN baseline results, we are happy to include those in our paper!
>
> We will also clarify the capabilities of other methods:
>
> Table 1: PPGN has $3$-WL expressivity and can count up to 7-cycles and homomorphism-count all graphs of tree-width 2. Nested GNNs are strictly between $1$-WL and $3$-WL. GSNs include explicit subgraph counts. While GSN is more powerful than $1$-WL, its exact expressive power depends on the chosen pattern to be counted.
>
> Table 2: We have a strict hierarchy in the expressive power of the baselines we chose: MPNN ≤ Subgraph GNN ≤ local $2$-GNN ≤ local $2$-FGNN. These variants, apart from MPNNs, are more expressive than $1$-WL and can subgraph-count up to 7-cycles in theory. Their homomorphism-expressivity is fully characterized in (B. Zhang et al., 2024).
>
> We will add these explanations in more detail and details for other real-world experiments to our appendix.
>
> > **W4**: “Unlabeled Graphs: …“
>
> **A4**: Yes, we can easily account for graphs with vertex labels, both for the patterns we aim to subgraph- or homomorphism-count and for the graphs themselves. Note that in our empirical evaluation, all GNNs use node attributes, if available in the datasets. Our proofs require minimal modifications to accommodate vertex labels. We will follow your suggestion and update our manuscript to include these modifications. Thank you!
>
> Minor Comments:
>
> Line 71: We referred to Tinhofer, who proved that two graphs are fractionally isomorphic if and only if $1$-WL does not distinguish them. This result was used by Dell et al. (2018) to prove that $1$-WL is equivalent to the homomorphism-counts of all trees. We would ask the Reviewer kindly to specify the Dvorak references so that we can add a citation if applicable.
>
> Line 96: We will remove the Dimitrov 2023 reference from this line.
>
> ---
>
> Thank you again for your valuable feedback, which led to new insights and results. Please let us know if you have any more suggestions or questions!

---

> > ### Comment · Reviewer_zkXv · 2024-08-08
> > **Good rebuttal**
> >
> > I have read the rebuttal and I am pleased with the responses. I would indeed appreciate if the case for cactus graphs is made stronger in the paper, as is explained here in the rebuttal. Moreover, the connection with subgraph GNNs, whether OSANs or other, should be explored or at least discussed briefly in related work. Based on the rebuttal and I am happy to accept the paper and to raise my score. The paper I referred to is "On recognizing graphs by numbers of homomorphisms by Zdeněk Dvořák.

---

### Official Review · Reviewer_pJ5G · 2024-07-17

**Soundness:** 3
**Presentation:** 3
**Contribution:** 3
**Rating:** 7
**Confidence:** 4

**Summary:**

This introduce $r$-loopy Weisfeiler-Leman, a new hierarchy of graph isomorphism test and a corresponding GNN framework. It achieves good cycle counting power and surpuss $k$-WL in some cases. The power of r-lWL is examined in various synthetic and real-world datasets.

**Strengths:**

1. Strong theoretic results with concrete proof.
2. The algorithm is local with good expressivity.

**Weaknesses:**

1. More datasets [1, 2] can be included to evaluate expressivity, especially long range expressivity.


[1] Vijay Prakash Dwivedi, Ladislav Rampásek, Michael Galkin, Ali Parviz, Guy Wolf, Anh Tuan Luu, Dominique Beaini: Long Range Graph Benchmark. NeurIPS 2022.
[2] Yanbo Wang, Muhan Zhang. Towards Better Evaluation of GNN Expressiveness with BREC Dataset. arxiv/abs/2304.07702

**Questions:**

1. For any pair of non-isomorphic graph, can $r$-lwl differentiate them with large enough $r$? (similar that $k$-WL can solve graph isomorphism problem for graph of node less than $k$).

**Limitations:**

Yes

---

> ### Author Rebuttal · Authors · 2024-08-06
>
> We thank the reviewer for their thorough review, acknowledging our contributions, and voting to accept our paper. We address each point individually below. “W/Q” numbers the weakness or question, followed by our response.
>
> ---
> > **W**: “More datasets [1, 2] can be included to evaluate expressivity, especially long range expressivity.”
>
> **A**:  Thank you for your suggestion. We have already included the BREC dataset in our manuscript, as detailed in Table 4.
> Regarding [1], we do not expect more expressive local GNNs, such as our proposed $r$-$\ell$GIN, to outperform state-of-the-art methods on LRGB, as the predictive performance appears to correlate with the GNNs' ability to capture long-range dependencies.  However, we are enthusiastic about testing our algorithm on these tasks. Our first preliminary results without any sweeping or positional encodings suggest improved performance over standard baseline:
> |  | Peptides Structural (MAE $\downarrow$) | Peptides Functional (AP $\uparrow$) |
> |---------------|-----------------|---------------------|
> | GCN	| 0.3496 ± 0.0013 | 59.30 ± 0.23 |
> | GINE	| 0.3547 ± 0.004 | 54.98 ± 0.79 |
> | GatedGCN	| 0.3420 ± 0.0013 | 58.64 ± 0.77 |
> |$r$-$\ell{}$GIN	| 0.2513 ± 0.0021 | 65.70 ± 0.60 |
>
> We are happy to follow Reviewer pJ5G’s suggestion and update our manuscript with these experiments.
>
> > **Q**: “For any pair of non-isomorphic graph, can $r$-lwl differentiate them with large enough $r$? (similar that $k$-WL can solve graph isomorphism problem for graph of node less than $k$).”
>
> **A**: Thank you for your insightful question. It is indeed an open question and part of our ongoing research. While it is established that for every $k$, there exists an $r$ such that $r$-$\ell$WL is not less powerful than $k$-WL (see Corollary 1 in our manuscript), this does not prove that increasing $r$ will enable $r$-$\ell$WL to solve graph isomorphism universally. We conjecture that $k$-WL and $r$-$\ell$WL are incomparable, where $r$ is chosen as described in Corollary 1. We acknowledge the complexity of this topic and appreciate the opportunity to explore it further.
>
> ---
> Thank you again for your insightful review and positive comments, especially regarding our work's strong theoretical contributions. Please let us know if you have any further questions!

---

### Decision · Program_Chairs · 2024-09-25

**Decision:**

Accept (oral)

**Comment:**

This paper suggests a new hierarchy of graph isomorphism tests, alternative to the standard kWL heirarchy. This is done by appending information on paths between neighborhing nodes the the standard MPNN procedure. The paper shows strong theoretical results, and in practice good separation abilities and competitive performance on standard benchmarks. The reviewers all recommended acceptance.